# SUBGRAPH DIFFUSION FOR 3D MOLECULAR REPRESENTATION LEARNING: COMBINING CONTINUOUS AND DISCRETE

## ABSTRACT

Molecular representation learning has shown great success in AI-based discovery. The 3D geometric structure contains crucial information about the underlying energy function, related to the physical and chemical properties. Recently, denoising diffusion probabilistic models have achieved impressive results in molecular conformation generation. However, the knowledge of pre-trained diffusion models has not been fully exploited in molecular representation learning. In this paper, we study the ability of representation learning inherent in the diffusion model for conformation generation. We introduce a new general diffusion model framework called MaskedDiff for molecular representation learning. Instead of adding noise to atoms like conventional diffusion models, MaskedDiff uses a discrete distribution to select a subset of the atoms to add continuous Gaussian noise at each step during the forward process. Further, we develop a novel subgraph diffusion model termed SUBGDIFF for enhancing the perception of molecular substructure in the denoising network (noise predictor), by incorporating auxiliary subgraph predictors during training. Experiments on molecular conformation generation and 3D molecular property predictions demonstrate the superior performance of our approach.

## 1 INTRODUCTION

Molecular representation learning (MRL) has attracted tremendous attention due to its significant role in learning from limited labeled data for applications like AI-based drug discovery (Shen & Nicolaou, 2019) and material science (Pollice et al., 2021). From a physical chemistry perspective, the 3D molecular conformation can largely determine the properties of molecules and the activities of drugs (Cruz-Cabeza & Bernstein, 2014). Thus, numerous geometric neural network architectures and self-supervised learning strategies have been proposed to explore 3D molecular structures to improve performance on downstream molecular property prediction tasks (Schütt et al., 2017; Zaidi et al., 2023; Liu et al., 2023a).

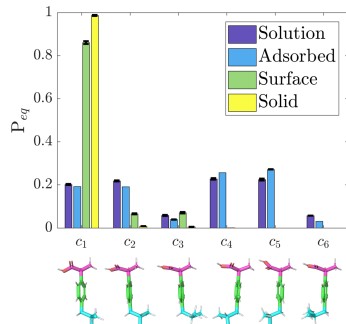

Figure 1: Equilibrium probability of the six ibuprofen conformers c1–c6 in four different conditions. The 3D substructure is a significant characteristic of a molecule.

Meanwhile, diffusion probabilistic models (DPM) have shown remarkable power to generate realistic samples, especially in synthesizing high-quality images and videos (Sohl-Dickstein et al., 2015; Ho et al., 2020). By modeling the generation as a reverse diffusion process, DPMs transform a random noise into a sample in the target distribution. Recently, diffusion models have demonstrated strong capabilities of molecular 3D conformation generation (Xu et al., 2022; Jing et al., 2022). The training process of a DPM for conformation generation can be viewed as the reconstruction of the original conformation from a noisy version, where the noise is modulated by different time steps. Consequently, the denoising objective in the diffusion model can naturally be used as a self-supervised representation learning technique (Pan et al., 2023). Inspired by this intuition, several works have used this technique for molecule pretraining (Liu et al., 2023b; Zaidi et al., 2023). Despite con-

siderable progress, the potential of DPMs for molecular representation learning has not been fully exploited. In this paper, we intend to explore the potential of generative DPM for MRL. To this aim, we raise the question: *Can we effectively enhance the perception of 3D molecular structures with the denoising network (noise predictor) of DPM? If yes, how to achieve it?*

To answer this question, we first analyze the gap between the current DPMs and the characteristics of molecular structures. Most diffusion models on molecules propose to independently inject continuous Gaussian noise into the every node feature (Hoogeboom et al., 2022) or atomic coordinates of 3D molecular geometry (Xu et al., 2022; Zaidi et al., 2023). This however implicitly models each atom as a separate particle, neglecting the substructure in the molecules which plays a significant role in molecular representation learning (Yu & Gao, 2022; Wang et al., 2022a).As shown in Figure 1[1], the 3D geometric substructure contains crucial information about the properties, such as the equilibrium distribution, crystallization and solubility (Marinova et al., 2018). As a result, uniformly adding same-scale Gaussian noise to all atoms makes it difficult for the denoising network to capture the properties of the 3D molecules related to the substructure. So here we try to answer the previous question by training a DPM with the knowledge of substructures.

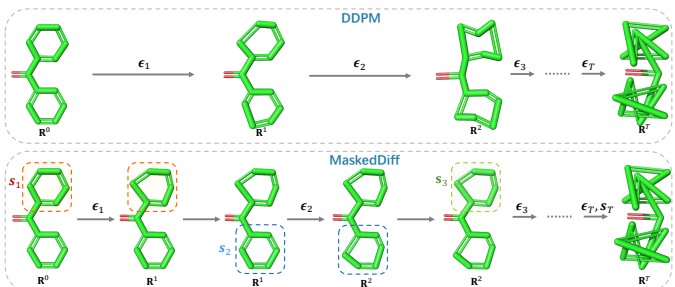

Figure 2: Comparison of forward process between DDPM (Ho et al., 2020) and MaskedDiff. For each step, DDPM adds noise into all atomic coordinates, while MaksedDIff selects a subset of the atoms to diffuse.

Toward this goal, we first propose a general masked diffusion framework named MaskedDiff, adding different Gaussian noise to 3D molecular conformation. Specifically, instead of adding the same Gaussian noise to every atomic coordinate, MaskedDiff introduces a discrete binary distribution to the diffusion process, where a mask vector sampling from the distribution can be used to select a subset of the atoms to determine which substructure the noise should be added to at the current time step (Figure 2). MaskedDiff can unify many masked-related diffusion models in other domain (Alcaraz & Strodthoff, 2022; Lei et al., 2023). Despite the fact that MaskedDiff can be directly used for self-supervised learning, it cannot be employed for generative tasks due to the difficulty of determining the mask vector during generation.

In order to make MaskedDiff usable for both molecular conformation generation and self-supervised representation learning, we design a novel subgraph diffusion model termed SUBGDIFF which incorporates a mask predictor (akin to a node classifier) in MaskedDiff during training that explicitly imposes the denoising network to capture the substructure information from the molecules. In SUBGDIFF, the substructure is concretized as the subgraph of the molecular graph. The mask predictor can also be used to generate the mask vector during molecule generation, thereby giving the generative ability to SUBGDIFF. With the ability to capture the substructure information from the noisy 3D molecular, the denoising networks tend to gain more representation power. It is made possible by the discrete distribution involved in the diffusion model, which, in contrast to conventional same-scale Gaussian models, captures the subgraph in the noisy graphs. These improvements enhance the performance of SUBGDIFF on molecular conformation generation and 3D molecular representation learning tasks. The experiments on molecular conformation generation and 3D molecular property prediction demonstrate the superior performance of our approach.

The key contributions of this paper are as follows: (1) The paper proposes a novel general mask diffusion model framework MaskedDiff. This framework combines the continuous and discrete characteristics, thereby being capable of recovering many typical diffusion models. (2) A new diffusion model SUBGDIFF is designed to enhance the representation power of the DPM for molecular conformation generation via equipping the subgraph constraint in the diffusion process. SUBGDIFF can be used for molecular conformation generation and self-supervised representation learning. (3) The proposed method achieves superior performance on molecular conformation generation and 3D molecular property protection tasks compared to the typical continuous diffusion models.

---

[1]Adapted with permission from Marinova et al. (2018). Copyright 2018 American Chemical Society.

## 2 RELATED WORK

**Diffusion models on graphs.** The diffusion models on graphs can be mainly divided into two categories: continuous diffusion and discrete diffusion. Continuous diffusion applies a Gaussian noise process on each node or edge (Ingraham et al., 2019; Niu et al., 2020), including GeoDiff (Xu et al., 2022), EDM (Hoogeboom et al., 2022). Meanwhile, discrete diffusion constructs the Markov chain on discrete space, including Digress (Haefeli et al., 2022) and GraphARM (Kong et al., 2023a). However, it remains open to exploring fusing the discrete characteristic into the continuous Gaussian on graph learning, although a closely related work has been proposed for images and cannot be used for generation (Pan et al., 2023). Our work, SUBGDIFF, is the first masked diffusion model for graphs, combining discrete characteristics and the continuous Gaussian.

**Conformation generation.** Various deep generative models have been proposed for conformation generation, including CVGAE (Mansimov et al., 2019), GRAPHDG (Simm & Hernandez-Lobato, 2020), CGCF (Xu et al., 2021a), CONFVAE (Xu et al., 2021b), CONFGF (Shi et al., 2021) and GEOMOL (Ganea et al., 2021). Recently, diffusion-based methods have shown competitive performance. Torsional Diffusion (Jing et al., 2022) raises a diffusion process on the hypertorus defined by torsion angles. However, it is not suitable as a self-supervised learning technique due to the lack of local information (length and angle of bonds). GEODIFF (Xu et al., 2022) generates molecular conformation by doing a conventional diffusion model on atomic coordinates. However, these methods view the atoms as separate particles, without considering the critical dependence between atoms, especially the substructure.

**SSL for 3D molecular property prediction.** There exist several works leveraging the 3D molecular conformation to boost the representation learning, including GeoSSL (Liu et al., 2023b), the denoising pretraining approach raised by Zaidi et al. (2023) and MoleculeSDE (Liu et al., 2023a), etc. However, those studies have not considered the molecular substructure in the pertaining. In this paper, we concentrate on how to boost the perception of molecular substructure in the denoising networks through the diffusion model.

## 3 PRELIMINARIES

**Notations.** We use $\mathbf{I}$ to denote the identity matrix with dimensionality implied by context. $\odot$ represents the element product and $\text{diag}(\mathbf{s})$ denotes the diagonal matrix with diagonal elements of the vector $\mathbf{s}$. The topological molecular graph can be denoted as $\mathcal{G}(\mathcal{V}, \mathcal{E}, \mathbf{X})$ where $\mathcal{V}$ is the set of nodes, $\mathcal{E}$ is the set of edges, $\mathbf{X}$ is the node feature matrix, and its corresponding 3D Conformational Molecular Graph is represented as $G_{3D}(\mathcal{G}, \mathbf{R})$, where $\mathbf{R} = [R_1, \cdots, R_{|\mathcal{V}|}] \in \mathbb{R}^{|\mathcal{V}| \times 3}$ is the set of 3D coordinates of atoms.

**DDPM.** Denoising diffusion probabilistic models (DDPM) (Ho et al., 2020) is a typical diffusion model (Sohl-Dickstein et al., 2015) which consists of a diffusion (aka forward) and a reverse process. In the setting of molecular conformation generation, the diffusion model adds noise on the 3D molecular coordinates $\mathbf{R}$ (Xu et al., 2022).

**Forward Process.** Given the fixed variance schedule $\beta_1, \beta_2, \cdots, \beta_T$, the posterior distribution $q(R^{1:T}|R^0)$ that is fixed to a Markov chain can be written as

$$q(\mathbf{R}^{1:T}|\mathbf{R}^0) = \prod_{t=1}^{T} q(\mathbf{R}^t|\mathbf{R}^{t-1}), \quad q(\mathbf{R}^t|\mathbf{R}^{t-1}) = \mathcal{N}(\mathbf{R}^t, \sqrt{1-\beta_t}\mathbf{R}^{t-1}, \beta_t\mathbf{I}). \quad (1)$$

To simplify notation, we consider the diffusion on single atom coordinate $R_v$ and omit the subscript $v$ to get the general notion $R$ throughout the paper. Let $\alpha_t = 1 - \beta_t$, $\bar{\alpha}_t = \prod_{i=1}^{t}(1 - \beta_t)$, and then the sampling of $R^t$ at any time step $t$ has the closed form: $q(R^t|R^0) = \mathcal{N}(R^t, \sqrt{\bar{\alpha}_t}R^0, (1 - \bar{\alpha}_t)\mathbf{I})$.

**Reverse Process and Training.** The reverse process is defined as a Markov chain starting from a Gaussian distribution $p(R^T) = \mathcal{N}(R^T; \mathbf{0}, \mathbf{I})$:

$$p_\theta(R_{0:T}) = p(R^T)\prod_{t=1}^{T} p_\theta(R^{t-1}|R^t); \quad p_\theta(R^{t-1}|R^t) = \mathcal{N}(R^{t-1}; \mu_\theta(R^t, t), \sigma_t), \quad (2)$$

where $\sigma_t = \frac{1-\bar{\alpha}_{t-1}}{1-\bar{\alpha}_t}\beta_t$ denote time-dependent constant. In DDPM, $\mu_\theta(R^t, t)$ is parameterized as $\mu_\theta(R^t, t) = \frac{1}{\bar{\alpha}_t}(R^t - \frac{\beta_t}{\sqrt{1-\bar{\alpha}_t}}\epsilon_\theta(R^t, t))$ and $\epsilon_\theta$, i.e., the *denoising network*, is parameterized by a

neural network where the inputs are $R^t$ and time step $t$. The training objective of DDPM is:

$$\mathcal{L}_{simple}(\theta) = \mathbb{E}_{t,R^0,\epsilon}[\|\|\epsilon - \epsilon_\theta(\sqrt{\bar{\alpha}_t}R^0 + \sqrt{1-\bar{\alpha}_t}\epsilon, t)\|^2], \quad \epsilon \sim \mathcal{N}(\mathbf{0}, \mathbf{I}).$$ (3)

**Sampling.** After training, samples are generated through the reverse process $p_\theta(R^{0:T})$. Specifically, $R^T$ is first sampled from $\mathcal{N}(\mathbf{0}, \mathbf{I})$, and $R^t$ in each step is predicted as follows,

$$R^{t-1} = \frac{1}{\sqrt{\alpha_t}}(R^t - \frac{1-\alpha_t}{\sqrt{1-\bar{\alpha}_t}}\epsilon_\theta(R^t, t)) + \sigma_t z, \quad z \sim \mathcal{N}(\mathbf{0}, \mathbf{I}).$$ (4)

## 4 METHODOLOGY

Directly using the typical diffusion model on atomic coordinates of 3D molecular means each atom is viewed as an independent single data point. However, the subgraph plays an important role in the molecular generation (Jin et al., 2020) and representation learning (Zang et al., 2023). Therefore, ignoring connections between nodes may hurt the denoising network's ability to capture molecular substructure. Here, we propose to involve a mask operation in each diffusion step, driving a new masked diffusion for 3D molecular representation learning. Further, we also include the mask predictor and reset the state of the Markov Chain to be an expectation of mask distribution, leading to a new diffusion model SUBGDIFF for molecular generation and representation.

### 4.1 AN IMPORTANT LEMMA FOR DIFFUSION MODEL

According to (Sohl-Dickstein et al., 2015; Ho et al., 2020), the diffusion model is trained by optimizing the variational bound on the negative log-likelihood $-\log p_\theta(R^0)$, in which the tricky terms are $L_{t-1} = D_{KL}(q(R^{t-1}|R^t, R^0)\|p_\theta(R^{t-1}|R^t)))$, $T \geq t > 1$. Here we provide a lemma that tells us the posterior distribution $q(R^{t-1}|R^t, R^0)$ used in the training and sampling algorithms of the diffusion model can be determined by $q(R^t|R^{t-1}, R^0)$, $q(R^{t-1}|R^0)$. Formally, we have

**Lemma 4.1** *Assume the forward and reverse processes of the diffusion model are both Markov chains. Given the forward Gaussian distribution $q(R^t|R^{t-1}, R^0) = \mathcal{N}(R^t; \mu_1 R^{t-1}, \sigma_1^2 I)$, $q(R^{t-1}|R^0) = \mathcal{N}(R^{t-1}; \mu_2 R^0, \sigma_2^2 I)$ and $\epsilon_0 \sim \mathcal{N}(\mathbf{0}, \mathbf{I})$, the distribution $q(R^{t-1}|R^t, R^0)$ is*

$$q(R^{t-1}|R^t, R^0) \propto \mathcal{N}(R^{t-1}; \frac{1}{\mu_1}(R^t - \frac{\sigma_1^2}{\sqrt{\mu_1^2\sigma_2^2 + \sigma_1^2}}\epsilon_0), \frac{\sigma_1^2\sigma_2^2}{\mu_1^2\sigma_2^2 + \sigma_1^2}I).$$ (5)

*Parameterizing $p_\theta(R^{t-1}|R^t)$ in the reverse process as $\mathcal{N}(R^{t-1}; \frac{1}{\mu_1}(R^t - \frac{\sigma_1^2}{\sqrt{\mu_1^2\sigma_2^2 + \sigma_1^2}}\epsilon_\theta(R^t, t)), \frac{\sigma_1^2\sigma_2^2}{\mu_1^2\sigma_2^2 + \sigma_1^2}I)$, the training objective of the DPM can be written as*

$$\mathcal{L}(\theta) = \mathbb{E}_{t,R^0,\epsilon}\left[\frac{\sigma_1^2}{2\mu_1^2\sigma_2^2}\|\epsilon - \epsilon_\theta(\mu_1\mu_2 R^0 + \sqrt{\mu_1^2\sigma_2^2 + \sigma_1^2}\epsilon, t)\|^2\right],$$ (6)

*and the sampling (reverse) process is*

$$R^{t-1} = \frac{1}{\mu_1}\left(R^t - \frac{\sigma_1^2}{\sqrt{\mu_1^2\sigma_2^2 + \sigma_1^2}}\epsilon_\theta(R^t, t)\right) + \frac{\sigma_1\sigma_2}{\sqrt{\mu_1^2\sigma_2^2 + \sigma_1^2}}z, \quad z \sim \mathcal{N}(\mathbf{0}, \mathbf{I})$$ (7)

The proof of the lemma can be found in the Appendix. Once we get the variables $(\mu_1, \sigma_1, \mu_2, \sigma_2)$, we can directly obtain the training objective and sampling process via lemma 4.1, which will help the design of new diffusion models.

### 4.2 MASKED DIFFUSION MODEL

Let us focus on the typical DDPM. Using reparameterization trick, we have $R_v^t = \sqrt{1-\beta_t}R_v^{t-1} + \sqrt{\beta_t}\epsilon_{t-1}, \forall v \in \mathcal{V}$, in which the Gaussian noise $\epsilon_{t-1}$ is injected to every atom. Moreover, the training objective in equation 3 shows that the denoising networks would always predict a Gaussian noise for all atoms. Neither the diffusion nor denoising process of DDPM does not take into account the substructure of the molecule. Instead, we propose **MaskedDiff**, where a mask vector $\mathbf{s}_t = [s_{t_1}, \cdots, s_{t_{|\mathcal{V}|}}]^\top \in \{0, 1\}^{|\mathcal{V}|}$ is sampled from a discrete distribution $p_{\mathbf{s}_t}(\mathcal{S})$ to select a subset of the atoms to determine which atoms will be added noise at step $t$. In molecular graphs, the discrete mask distribution $p_{\mathbf{s}_t}(\mathcal{S})$ is equivalent to

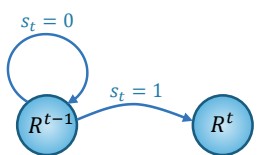

Figure 3: The Markov Chain of MaskedDiff is a lazy Markov Chain.

the subgraph distribution, defined over a predefined sample space $\chi = \{G^i_{\text{sub}}\}^N_{i=1}$, where each sample is a connected subgraph extracted from $G$. Further, the pre-defined distribution $p_{\mathbf{s}_t}(\mathcal{S})$ should keep the selected connected subgraph to cohere with the molecular substructures. Here, we adopt a Torsional-based decomposition methods (Jing et al., 2022)(subsec. 4.3.3 ). Thus, the state transition of MaskedDiff can be formulated as (Figure 3): $R^t_v = \sqrt{1-\beta_t}R^{t-1}_v + \sqrt{\beta_t}\epsilon_{t-1}$ if $s_{t_v} = 1$, otherwise $R^t_v = R^{t-1}_v$, which can be rewritten as $R^t_v = \sqrt{1 - s_{t_v}\beta_t}R^{t-1}_v + \sqrt{s_{t_v}\beta_t}\epsilon_{t-1}$. The posterior distribution $q(R^{1:T}|R^0)$ can be expressed as matrix form:

$$q(\mathbf{R}^{1:T}|\mathbf{R}^0) = \prod_{t=1} q(\mathbf{R}^t|\mathbf{R}^{t-1}); \quad q(\mathbf{R}^t|\mathbf{R}^{t-1}) = \mathcal{N}(\mathbf{R}^t, \sqrt{1-\beta_t\text{diag}(\mathbf{s}_t)}\mathbf{R}^{t-1}, \beta_t\text{diag}(\mathbf{s}_t)\mathbf{I}). \quad (8)$$

To simplify the notation, we consider the diffusion on single node $v \in G_{3D}$ and omit the subscript of coordinate $R^t_v$ and $s_{t_v}$ to get the notion $R^t$ and $s_t$. By defining $\gamma_t = 1 - s_t\beta_t, \bar{\gamma}_t = \prod^t_{i=1}(1 - s_t\beta_t)$, the closed form of sampling $R^t$ from $R^0$ is $q(R^t|R^0) = \mathcal{N}(R^t, \sqrt{\bar{\gamma}_t}R^0, (1 - \bar{\gamma}_t)\mathbf{I})$. By Lemma 4.1, with $\mu_1 = \sqrt{1 - s_t\beta_t}, \sigma_1 = \sqrt{s_t\beta_t}, \mu_2 = \sqrt{\bar{\gamma}_{t-1}}, \sigma_2 = \sqrt{1 - \bar{\gamma}_{t-1}}$, the training objective of MaskedDiff is:

$$\mathcal{L}(\theta) = \mathbb{E}_{t,R^0,\epsilon}\left[\frac{s_t\beta_t}{2(1 - s_t\beta_t)(1 - \bar{\gamma}_{t-1})}\|\epsilon - \epsilon_\theta(\sqrt{\bar{\gamma}_t}R^0 + \sqrt{(1 - \bar{\gamma}_t)}\epsilon, t, \mathcal{G})\|^2\right]. \quad (9)$$

It is clear that if $s_t = 0, \mathcal{L}(\theta) = 0$, which means that this node $v$ will not be trained in time step $t$. Therefore, the $\mathbf{s}_t$ also determines the substructure selected in time step $t$. One drawback of MaskedDiff is that it cannot be directly employed in sampling since it is unable to obtain $(\mathbf{s}_1, \mathbf{s}_2, \cdots, \mathbf{s}_T)$ to derive the $\sigma_1$ and $\sigma_2$ in equation 7. The discussion with related work (MDM (Pan et al., 2023), MDSM (Lei et al., 2023) and SSSD (Alcaraz & Strodthoff, 2022) ) is deferred to Appendix A.

### 4.3 SUBGDIFF: A DIFFUSION MODEL FOR REPRESENTATION LEARNING AND CONFORMATION GENERATION

In this section, we propose a novel diffusion model called SUBGDIFF for self-supervised representation learning and conformation generation. Inheriting from MaskedDiff, SUBGDIFF adopts the mask vector to embed the substructure into the denoising network $\epsilon_\theta$. However, the main problem is that MaskedDiff cannot be used for generations. To solve the problem, SUBGDIFF applies multiple techniques to make it better for generation.

#### 4.3.1 MASK ESTIMATION

Recall the forward process in MaskedDiff, only a subgraph (substructure) in the molecular graph is chosen to diffuse at each time step. Correspondingly, during the reverse process, the mask is used to determine which subgraph needs to be denoised. This means that the sampling process will prioritize the subgraphs that are selected by the mask, which is also reflected by equation 7 ($\sigma_1$ and $\sigma_2$). However, the mask series $(\mathbf{s}_1, \mathbf{s}_2, \cdots, \mathbf{s}_T)$ cannot be accessed during sampling. This uncertainty will bring the tribulation to make the denoising network capture the substructure. To estimate the mask series, SUBGDIFF uses a mask predictor to infer $\mathbf{s}_t$ and adapt an expectation state to eliminate the effect of $(\mathbf{s}_1, \cdots, \mathbf{s}_{t-1})$.

**Mask Predictor.** Given the current time step $t$ in sampling, we need to infer the pivotal mask (subgraph) $\mathbf{s}_t$ to highlight the subgraph of $G_{3D}(\mathbf{X}, \mathbf{R}^t)$ that will be denoised to recover $\mathbf{R}^{t-1}$. Thus, we first introduce a mask predictor to estimate the mask vector $\mathbf{s}_t$ during training (the theoretical motivation can be seen in Appendix). Consequently, the **training objective** of SUBGDIFF is:

$$\mathcal{L}_{simple}(\theta, \vartheta) = \mathbb{E}_{t,\mathbf{R}^0,\mathbf{s}_t,\epsilon}[\|\text{diag}(\mathbf{s}_t)(\epsilon - \epsilon_\theta(\mathcal{G}, \mathbf{R}^t, t))\|^2 + \lambda\text{BCE}(\mathbf{s}_t, s_\vartheta(\mathcal{G}, \mathbf{R}^t, t))], \quad (10)$$

where $\text{BCE}(s_{t_i}, s_{\vartheta_i}) = s_{t_i}\log s_\vartheta(\mathcal{G}, \mathbf{R}^t, t)_i + (1 - s_{t_i})\log(1 - s_\vartheta(\mathcal{G}, \mathbf{R}^t, t)_i)$ is the Binary Cross Entropy loss and $\lambda$ is the weight used for the trade-off. The mask predictor $s_\vartheta$ is implemented as a node classifier with $G_{3D}(\mathcal{G}, \mathbf{R}^t)$ as input and shares a molecule encoder with $\epsilon_\theta$, thereby explicitly imposing the denoising network to capture the substructure information from molecules. Eventually, the $s_\vartheta$ can be used to infer the mask vector $\hat{\mathbf{s}}_t = s_\vartheta(\mathcal{G}, \hat{\mathbf{R}}^t, t)$ during sampling. More importantly, this BCE loss explicitly imposes the denoising network to capture the substructure information from the molecules.

**Expectation State Diffusion.** As mentioned above, the MaskedDiff cannot be used for sampling due to the unknown mask series $(\mathbf{s}_1, \mathbf{s}_2, \cdots, \mathbf{s}_t)$. We have designed a mask predictor to infer $\mathbf{s}_t$.

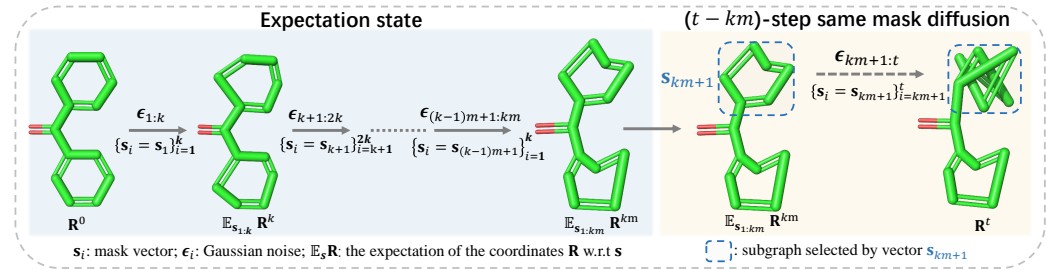

$s_i$: mask vector; $\epsilon_i$: Gaussian noise; $\mathbb{E}_s R$: the expectation of the coordinates $R$ w.r.t $s$    [ ]: subgraph selected by vector $s_{km+1}$

Figure 4: The forward process of SUBGDIFF. The state 0 to $km$ uses the expectation state, and the state $km+1$ to $t$ applies the same-mask diffusion.

However, using another predictor to infer $(s_1, \cdots, s_{t-1})$ solely from $R^t$ becomes challenging due to the intricate modulation of noise introduced in $R^t$ through multi-step Gaussian processes. This complex modulation of noise in $R^t$ also heightens the challenge of predicting $s_t$, a critical factor in enhancing the denoising network's ability to discern substructures during self-supervised learning. Recall the forward process of MaskedDiff. The state $R^{t-1} = \sqrt{\bar{\gamma}_{t-1}}R^0 + \sqrt{(1-\bar{\gamma}_{t-1})}\epsilon_0 \propto (s_1, \cdots, s_{t-1}, \epsilon_0)$. To eliminate the effect of mask series, we use the mean state $\mathbb{E}_{s_{1:t-1}}R^{t-1}$ to estimate the state $R^{t-1}$. Assume each node $v \in \mathcal{V}$, $s_{t_v} \sim Bern(p)$ (i.i.d. w.r.t. $t$), the $\mathbb{E}_{s_{1:t-1}}R^{t-1}$ can be formulated as:

$$\mathbb{E}_{s_{1:t-1}}R^{t-1} = \sqrt{\bar{\alpha}_i}R^0 + p(\sum_{i=1}^{t}\frac{\bar{\alpha}_t}{\bar{\alpha}_i}\beta_i)^{1/2}\epsilon_0, \tag{11}$$

where $\alpha_i := (p\sqrt{1-\beta_i}+1-p)^2$ and $\bar{\alpha}_t := \prod_{i=1}^{t}\alpha_i$ are general form of $\alpha_j$ and $\bar{\alpha}_j$ in DDPM ($p = 1$), respectively. This estimation is reasonable since the expectation $\mathbb{E}_{s_{1:t-1}}R^{t-1}$ is like a cluster center of $R^{t-1}$, which can represent the $R^{t-1}$ properly. Meanwhile, using expectation is beneficial to reduce the complexity of $R^t$ for predicting the mask $s_t$ during training. This will improve the denoising network to perceive the substructure when we use the diffusion model for pretraining. Eventually, we get a new forward process, in which, state 0 to state $t-1$ use the $\mathbb{E}_{s_{1:t-1}}R^{t-1}$ and state $t$ remains as MaskedDiff. Formally, we have $q(R^t|R^{t-1}) = \mathcal{N}(R^t; \sqrt{1-s_t\beta_t}R^{t-1}, (s_t\beta_t)\mathbf{I})$ and $q(R^{t-1}|R^0) = q(\mathbb{E}R^{t-1}|R^0) = \mathcal{N}(\mathbb{E}R^{t-1}; \prod_{i=1}^{t-1}\sqrt{\alpha_i}R^0, p^2\sum_{i=1}^{t-1}\prod_{j=i+1}^{t-1}\alpha_j\beta_i I)$. From Lemma 4.1, the training objective we can use equation 10 and the sampling process is:

$$R^{t-1} = \frac{1}{\sqrt{1-\hat{s}_t\beta_t}}R^t - \frac{\hat{s}_t\beta_t}{\sqrt{1-\hat{s}_t\beta_t}\sqrt{\hat{s}_t\beta_t + (1-s_t\beta_t)p^2\sum_{i=1}^{t-1}\frac{\bar{\alpha}_{t-1}}{\bar{\alpha}_i}\beta_i}}\epsilon_\theta(R^t, t) + \sigma_t z, \tag{12}$$

where $\hat{s}_t = s_\vartheta(R^t, t)$ and $\sigma_t = \hat{s}_t\beta_t p^2 \sum_{i=1}^{t-1}\frac{\bar{\alpha}_{t-1}}{\bar{\alpha}_i}\beta_i / (\hat{s}_t\beta_t + p^2(1-\hat{s}_t\beta_t)\sum_{i=1}^{t-1}\frac{\bar{\alpha}_{t-1}}{\bar{\alpha}_i}\beta_i)$.

### 4.3.2 $k$-STEP SAME-MASK DIFFUSION.

Although we can successfully use MaskedDiff for sampling with Exceptional state and mask predictor, optimizing the mask predictor with equation 10 is still not trivial. To be specific, the mask predictor should be capable of perceiving the sensible noise change between time steps $t-1$ and $t$. However, the noise scale $\beta_t$ is relatively small when $t$ is small, especially if the diffusion step is larger than a thousand. As a result, it is difficult to precisely predict

---

**Algorithm 1:** Training SUBGDIFF

**Input:** A molecular graph $G_{3D}$, $k$ for same mask diffusion
Sample $t \sim \mathcal{U}(1, ..., T)$, $\epsilon \sim \mathcal{N}(\mathbf{0}, \mathbf{I})$
Sample $\mathbf{s}^t \sim p_{s_t}(\mathcal{S})$
$\mathbf{R}^t \leftarrow q(\mathbf{R}^t|\mathbf{R}^0)$     ▷ Eq.14
$\mathcal{L}_1 = \text{BCE}(\mathbf{s}_t, s_\vartheta(\mathcal{G}, \mathbf{R}^t, t))$
$\mathcal{L}_2 = \|\text{diag}(\mathbf{s}_t)(\epsilon - \epsilon_\theta(\mathcal{G}, \mathbf{R}^t, t))\|^2$
optimizer.step($\lambda\mathcal{L}_1 + \mathcal{L}_2$)

---

the mask. To reduce the complexity of the mask series $(\mathbf{s}_1, \mathbf{s}_2, \cdots, \mathbf{s}_T)$ and accumulate more noise on the same subgraph, SUBGDIFF generalizes the one-step mask sampling to $k$-step mask sampling (Figure 5 in Appendix), in which the selected subgraph will be continuously diffused $k$ steps. After that, the difference between the selected and unselected parts will be distinct enough to help the mask predictor perceive it. The forward process of $k$-step step Same-mask diffusion can be written as ($t > k, k \in \mathbb{N}$):

$$q(R^t|R^{t-k}) = \mathcal{N}\left(R^t, \sqrt{\prod_{i=t-k+1}^{t}(1-s_{t-k+1}\beta_i)}R^{t-k}, (1-\prod_{i=t-k}^{t}(1-s_{t-k+1}\beta_i)\mathbf{I})\right). \tag{13}$$

### 4.3.3 SUBGDIFF

With $k$-step same mask and mask estimation techniques, we propose a novel diffusion model called SUBGDIFF. SUBGDIFF divides the entire diffusion step $T$ into $T/k$ diffusion intervals. In each interval $[ki, k(i+1)]$, the mask vectors $\{\mathbf{s}_j\}_{j=ki+2}^{k(i+1)}$ are equal to $\mathbf{s}_{ki+1}$.

To eliminate the effect of $\{\mathbf{s}_{ik+1}|i = 1, 2, \ldots\}$ and obtain the generative ability, SUBGDIFF also adopts the expectation state at the split time step $\{ik|i = 1, 2, \cdots\}$, that is, gets the expectation of $\mathbb{E}\mathbf{R}^{ik}$ at step $ik$ w.r.t. $\mathbf{s}_{ik+1}$. We therefore propose a new two-phase diffusion process. In the first phase, the state 1 to state $k\lfloor t/k \rfloor$ use the expectation state diffusion, while in the second phase, state $k(\lfloor t/k \rfloor) + 1$ to state $t$ use the $k$-step same mask diffusion. The state transition refers to Figure 4. With $m := \lfloor t/k \rfloor$, the two phases can be formulated as follows,

**Phase I**: Step $0 \to k\lfloor t/k \rfloor$: $\mathbb{E}_{s_{1:km}} R^{km} = \sqrt{\bar{\alpha}_m} R^0 + p\sqrt{\sum_{l=1}^{m} \frac{\bar{\alpha}_m}{\bar{\alpha}_l}(1 - \prod_{i=(l-1)k+1}^{kl}(1 - \beta_i))}\epsilon_0$,

where $\alpha_j = (p\sqrt{\prod_{i=(j-1)k+1}^{kj}(1 - \beta_i)} + 1 - p)^2$ is a general forms of $\alpha_j$ in equation 11 (in which case $k = 1$) and $\bar{\alpha}_t = \prod_{i=1}^{t} \alpha_i$. In the rest of the paper, $\alpha_j$ denotes the general version without a special statement. Actually, the $\mathbb{E}_{s_{1:km}} R^{km}$ only calculate the expectation of random variable $\{\mathbf{s}_{ik+1}|i = 1, 2, \cdots\}$.

**Phase II**: Step $k\lfloor t/k \rfloor + 1 \to t$: The phase is a $(t - km)$-step same mask diffusion. $R^t = \sqrt{\prod_{i=km+1}^{t}(1 - \beta_i s_{km+1})}\mathbb{E}_{s_{1:km}} R^{km} + \sqrt{1 - \prod_{i=km+1}^{t}(1 - \beta_i s_{km+1})}\epsilon_{km}$. Let $\gamma_i = 1 - \beta_i s_{km+1}$, $\bar{\gamma}_t = \prod_{i=1}^{t} \gamma_i$, and $\bar{\beta}_t = \prod_{i=1}^{t}(1 - \beta_i)$, we can drive the single-step state transition: $q(R^t|R^{t-1}) = \mathcal{N}(R^t; \sqrt{\gamma_t}R^{t-1}, (1 - \gamma_t)\mathbf{I})$ and

$$q(R^{t-1}|R^0) = \mathcal{N}(R^{t-1}; \sqrt{\frac{\bar{\gamma}_{t-1}\bar{\alpha}_m}{\bar{\gamma}_{km}}}R^0, \left(\frac{\bar{\gamma}_{t-1}}{\bar{\gamma}_{km}}p^2 \sum_{l=1}^{m} \frac{\bar{\alpha}_m}{\bar{\alpha}_l}(1 - \frac{\bar{\beta}_{kl}}{\bar{\beta}_{(l-1)k}}) + 1 - \frac{\bar{\gamma}_{t-1}}{\bar{\gamma}_{km}}\right)I). \tag{14}$$

Then we can obtain $\mu_1, \sigma_1, \mu_2, \sigma_2$ in Lemma 4.1. Thus, the **training objective** of SUBGDIFF is:

$$\mathcal{L}_{simple}(\theta, \vartheta) = \mathbb{E}_{t,\mathbf{R}^0,\mathbf{s}_t,\epsilon}[\|\text{diag}(\mathbf{s}_t)(\epsilon - \epsilon_\theta(\mathcal{G}, \mathbf{R}^t, t))\|^2 - \lambda\text{BCE}(\mathbf{s}_t, s_\vartheta(\mathcal{G}, \mathbf{R}^t, t))], \tag{15}$$

where $\mathbf{R}^t$ can be calculated by equation 14. Because $s_\vartheta$ shares the encoder with $\epsilon_\theta$, when using this objective for pretraining, $\epsilon_\vartheta$ can be effectively trained to capture the substructure information.

**Sampling.** The sampling process does not fully correspond to the forward process. Although the forward process uses the expectation state w.r.t $\mathbf{s}$, we can only update the mask $\hat{\mathbf{s}}_t$ when $t = ik, i = 1, 2, \cdots$. Eventually, the sampling process is shown below,

$$R^{t-1} = \frac{1}{\sqrt{\gamma_t}}(R^t - \frac{\hat{s}_{k\lfloor t/k \rfloor+1}\beta_t}{\sqrt{\gamma_t(\frac{\bar{\gamma}_{t-1}}{\bar{\gamma}_{km}}p^2 \sum_{l=1}^{m} \frac{\bar{\alpha}_m}{\bar{\alpha}_l}(1 - \frac{\bar{\beta}_{kl}}{\bar{\beta}_{(l-1)k}}) + 1 - \frac{\bar{\gamma}_{t-1}}{\bar{\gamma}_{km}}) + \hat{s}_{k\lfloor t/k \rfloor+1}\beta_t}}\epsilon_\theta(R^t, t))$$

$$+ \frac{\sqrt{\hat{s}_{k\lfloor t/k \rfloor+1}\beta_t}\sqrt{\frac{\bar{\gamma}_{t-1}}{\bar{\gamma}_{km}}p^2 \sum_{l=1}^{m} \frac{\bar{\alpha}_m}{\bar{\alpha}_l}(1 - \frac{\bar{\beta}_{kl}}{\bar{\beta}_{(l-1)k}}) + 1 - \frac{\bar{\gamma}_{t-1}}{\bar{\gamma}_{km}}}}{\sqrt{\gamma_t(\frac{\bar{\gamma}_{t-1}}{\bar{\gamma}_{km}}p^2 \sum_{l=1}^{m} \frac{\bar{\alpha}_m}{\bar{\alpha}_l}(1 - \frac{\bar{\beta}_{kl}}{\bar{\beta}_{(l-1)k}}) + 1 - \frac{\bar{\gamma}_{t-1}}{\bar{\gamma}_{km}}) + \hat{s}_{k\lfloor t/k \rfloor+1}\beta_t}}z, \tag{16}$$

where $z \sim \mathcal{N}(\mathbf{0}, \mathbf{I})$, $m = \lfloor t/k \rfloor$ and $\hat{s}_{k\lfloor t/k \rfloor+1} = s_\vartheta(\mathcal{G}, R^{km+1}, km + 1)$. It is clear that the subgraph selected by $\hat{\mathbf{s}}_{km+1}$ will be generated preferentially. The mask predictor can be viewed as a discriminator of important subgraphs, indicating the optimal subgraph should be recovered in the next $k$ steps. After

---

**Algorithm 2:** Sampling from SUBGDIFF

Sample $\mathbf{R}^T \sim \mathcal{N}(\mathbf{0}, \mathbf{I})$
**for** $t = T$ to $1$ **do**
  $\mathbf{z} \sim \mathcal{N}(\mathbf{0}, \mathbf{I})$ if $t > 1$, else $\mathbf{z} = \mathbf{0}$
  **If** $t\%k == 0$ or $t == T$: $\hat{\mathbf{s}} \leftarrow s_\vartheta(\mathcal{G}, \mathbf{R}^t, t)$
  $\hat{\epsilon} \leftarrow \epsilon_\theta(\mathcal{G}, \mathbf{R}^t, t)$     ▷ Posterior
  $\mathbf{R}^{t-1} \leftarrow$ equation 16     ▷ sampling
**end**
**return** $\mathbf{R}^0$

---

the key subgraph (substructure) is generated properly, the model can gently fine-tune the rest atoms (cf. the video in supplementary material). This subgraph diffusion would intuitively increase the robustness and generalization of the generation process, which is also verified by the experiments in sec. 5.2. While the DDPM (or GEODIFF) generates the atomic coordinates altogether, which is sub-optimal since some parts of the molecule shouldn't be revised after well-generated. The training and sampling algorithms of SUBGDIFF are summarized in Alg. 1 and Alg. 2.

## 4.4 MASK DISTRIBUTION

As mentioned in Subsection 4.2, the subgraphs (mask vectors) sampled from the mask distribution should be connected. In this paper, we pre-define the mask distribution to be a discrete distribution, with sample space $\chi = \{G_{sub}^i\}_{i=1}^{N}$, and $p_t(\mathcal{S} = G_{sub}^i) = 1/N, t > 1$, where $G_{sub}^i$ is the subgraph split by the Torsional-based decomposition methods (Jing et al., 2022). The decomposition approach will cut off one torsional edge in a 3D molecule to make the molecule into two components, each of which contains at least two atoms. The two components are represented as two complementary mask vectors (i.e. $\mathbf{s}' + \mathbf{s} = \mathbf{1}$). Thus $n$ torsional edges in $G_{3D}^i$ will generate $2n$ subgraphs. Finally, for each atom $v$, the $s_{t_v} \sim Bern(0.5)$, i.e. $p = 0.5$ in SUBGDIFF.

## 5 EXPERIMENTS

We conducted experiments to address the following two questions: 1) Can substructure improve the representation ability of the denoising network during self-supervised learning? 2) Can the SUBGDIFF outperform the conventional diffusion model in conformation generation? For the first question, we employ SUBGDIFF as a denoising pretraining task. For the second question, we compare SUBGDIFF with GEODIFF. We will pay more attention to the first question.

Table 1: Results on 12 quantum mechanics prediction tasks from QM9. We take 110K for training, 10K for validation, and 11K for testing. The evaluation is mean absolute error (MAE), and the best and the second best results are marked in bold and underlined, respectively. The backbone is **SchNet**.

| Pretraining | Alpha ↓ | Gap ↓ | HOMO↓ | LUMO↓ | Mu ↓ | Cv ↓ | G298 ↓ | H298 ↓ | R2 ↓ | U298 ↓ | U0 ↓ | Zpve ↓ |
|---|---|---|---|---|---|---|---|---|---|---|---|---|
| Random init | 0.070 | 50.59 | 32.53 | 26.33 | 0.029 | 0.032 | 14.68 | 14.85 | 0.122 | 14.70 | 14.44 | 1.698 |
| Supervised | 0.070 | 51.34 | 32.62 | 27.61 | 0.030 | 0.032 | 14.08 | 14.09 | 0.141 | 14.13 | 13.25 | 1.727 |
| Type Prediction | 0.084 | 56.07 | 34.55 | 30.65 | 0.040 | 0.034 | 18.79 | 19.39 | 0.201 | 19.29 | 18.86 | 2.001 |
| Angle Prediction | 0.084 | 57.01 | 37.51 | 30.92 | 0.037 | 0.034 | 15.81 | 15.89 | 0.149 | 16.41 | 15.76 | 1.850 |
| 3D InfoGraph | 0.076 | 53.33 | 33.92 | 28.55 | 0.030 | 0.032 | 15.97 | 16.28 | 0.117 | 16.17 | 15.96 | 1.666 |
| GeossL-RR | 0.073 | 52.57 | 34.44 | 28.41 | 0.033 | 0.038 | 15.74 | 16.11 | 0.194 | 15.58 | 14.76 | 1.804 |
| GeossL-InfoNCE | 0.075 | 53.00 | 34.29 | 27.03 | 0.029 | 0.033 | 15.67 | 15.53 | 0.125 | 15.79 | 14.94 | 1.675 |
| GeossL-EBM-NCE | 0.073 | 52.86 | 33.74 | 28.07 | 0.031 | 0.032 | 14.02 | 13.65 | 0.121 | 13.70 | 13.45 | 1.677 |
| MoleculeSDE | 0.062 | 47.74 | 28.02 | 24.60 | 0.028 | 0.029 | 13.25 | 12.70 | 0.120 | 12.68 | 12.93 | 1.643 |
| **Ours** | **0.054** | **44.88** | **25.45** | **23.75** | **0.027** | **0.028** | **12.03** | **11.46** | **0.110** | **11.32** | **11.25** | **1.568** |

### 5.1 MOLECULAR PROPERTY PREDICTION

This experiment aims to verify whether the introduced mask in the diffusion can enhance the denoising network to perceive the structure of 3D molecules.

Table 2: Results for 2D molecular property prediction tasks (with 2D topology only). We report the mean (and standard deviation) ROC-AUC of three random seeds with scaffold splitting for each downstream task. The backbone is GIN. The best and second best results are marked bold and underlined, respectively.

| Pre-training | BBBP ↑ | Tox21 ↑ | ToxCast ↑ | Sider ↑ | ClinTox ↑ | MUV ↑ | HIV ↑ | Bace ↑ | Avg ↑ |
|---|---|---|---|---|---|---|---|---|---|
| – (random init) | 68.1±0.59 | 75.3±0.22 | 62.1±0.19 | 57.0±1.33 | 83.7±2.93 | 74.6±2.35 | 75.2±0.70 | 76.7±2.51 | 71.60 |
| AttrMask | 65.0±2.36 | 74.8±0.25 | 62.9±0.11 | 61.2±0.12 | 87.7±1.19 | 73.4±2.02 | 76.8±0.53 | 79.7±0.33 | 72.68 |
| ContextPred | 65.7±0.62 | 74.2±0.06 | 62.5±0.31 | 62.2±0.59 | 77.2±0.88 | 75.3±1.57 | 77.1±0.86 | 76.0±2.08 | 71.28 |
| InfoGraph | 67.5±0.11 | 73.2±0.43 | 63.7±0.50 | 59.9±0.30 | 76.5±1.07 | 74.1±0.74 | 75.1±0.99 | 77.8±0.88 | 70.96 |
| MolCLR | 66.6±1.89 | 73.0±0.16 | 62.9±0.38 | 57.5±1.77 | 86.1±0.95 | 72.5±2.38 | 76.2±1.51 | 71.5±3.17 | 70.79 |
| 3D InfoMax | 68.3±1.12 | 76.1±0.18 | 64.8±0.25 | 60.6±0.78 | 79.9±3.49 | 74.4±2.45 | 75.9±0.59 | 79.7±1.54 | 72.47 |
| GraphMVP | 69.4±0.21 | 76.2±0.38 | 64.5±0.20 | 60.5±0.25 | 86.5±1.70 | 76.2±2.28 | 76.2±0.81 | 79.8±0.74 | 73.66 |
| MoleculeSDE(VE) | 68.3±0.25 | 76.9±0.23 | 64.7±0.06 | 60.2±0.29 | 80.8±2.53 | 76.8±1.71 | 77.0±1.68 | 79.9±1.76 | 73.15 |
| MoleculeSDE(VP) | 70.1±1.35 | 77.0±0.12 | 64.0±0.07 | 60.8±1.04 | 82.6±3.64 | 76.6±3.25 | 77.3±1.31 | 81.4±0.66 | 73.73 |
| Ours | **70.2±2.23** | **77.2±0.39** | **65.0±0.48** | 62.2±0.974 | **88.2±1.57** | **77.3±1.17** | 77.6±0.51 | **82.1±0.96** | **74.85** |

**Dataset and Settings.** For pretraining, we follow Liu et al. (2023a) and use PCQM4Mv2 (Hu et al., 2020b). It's a sub-dataset of PubChemQC (Nakata & Shimazaki, 2017) with 3.4 million molecules with both the geometric conformations and topological graph. The downstream tasks are various molecular property predictions. Regarding 3D fine-tuning, we take the QM9 dataset and follow the literature (Schütt et al., 2017; 2021; Liu et al., 2023a), using 110K for training, 10K for validation and 11k for testing. For 2D fine-tuning, we use eight 2D molecular property prediction tasks from MoleculeNet (Wu et al., 2017).

**Pretraining framework.** To explore the potential of the proposed method in self-supervised learning tasks, we consider MoleculeSDE (Liu et al., 2023a), a SOTA pretraining framework, to be the training backbone for pertaining 3D molecules, where the $2D \rightarrow 3D$ model we use SUBGDIFF and $3D \rightarrow 2D$ we simply extend the SUBGDIFF to process the node feature and graph adjacency. The details can be found in the Appendix.

**Baselines.** For 3D tasks, we incorporate the three coordinate-MI-unaware SSL methods: (1) Type Prediction; (2) Angle Prediction; (3) 3D InfoGraph (Stärk et al., 2022), and two contrastive baselines: (4) GeoSSL-InfoNCE (Oord et al., 2018) and (5) GeoSSL-EBM-NCE (Liu et al., 2021), Additionally, we include two generative SSL baseline: (6) GeoSSL-RR (RR for Representation Reconstruction) and (7) MoleculeSDE(Liu et al., 2023a) For 2D tasks, we consider AttrMask (Hu et al., 2020a; Liu et al., 2019), ContexPred (Hu et al., 2020a), InfoGraph (Sun et al., 2020), MolCLR (Wang et al., 2022b), 3D InfoMax, vanilla GraphMVP (Liu et al., 2021), and MoleculeSDE. More details see Appendix F.1.

**Results.** The results shown in Table 1 and Table 2 suggest that SUBGDIFF outperforms MoleculeSDE in most downstream tasks, demonstrating the introduced mask vector boosts the perception of molecular substructure in the denoising network during pretraining. Further, SUBGDIFF achieves SOTA performance compared to the baselines. This also reveals that the proposed masked-based denoising objective is promising for molecular representation learning due to the involvement of the prior knowledge concerning substructure during training.

Table 4: Results on **GEOM-QM9** dataset under different diffusion timesteps. DDPM (Ho et al., 2020) is the sampling method used in GeoDiff. Our proposed sampling method (Algorithm 2) can be viewed as a DDPM variant. ▲/▼ denotes SUBGDIFF outperforms/underperforms GEODIFF. The threshold $\delta = 0.5$Å.

| Models | Timesteps | Sampling method | COV-R (%) ↑ | | MAT-R (Å) ↓ | | COV-P (%) ↑ | | MAT-P (Å) ↓ | |
|---|---|---|---|---|---|---|---|---|---|---|
| | | | Mean | Median | Mean | Median | Mean | Median | Mean | Median |
| GEODIFF | 5000 | DDPM | 80.36 | 83.82 | 0.2820 | 0.2799 | 53.66 | 50.85 | 0.6673 | 0.4214 |
| SUBGDIFF | 5000 | DDPM (ours) | 90.91▲ | 95.59▲ | 0.2460▲ | 0.2351▲ | 50.16▼ | 48.01▼ | 0.6114▲ | 0.4791▼ |
| GEODIFF | 500 | DDPM | 80.20 | 83.59 | 0.3617 | 0.3412 | 45.49 | 45.45 | 1.1518 | 0.5087 |
| SUBGDIFF | 500 | DDPM (ours) | 89.78▲ | 94.17▲ | 0.2417▲ | 0.2449▲ | 50.03▲ | 48.31▲ | 0.5571▲ | 0.4921▲ |
| GEODIFF | 200 | DDPM | 69.90 | 72.04 | 0.4222 | 0.4272 | 36.71 | 33.51 | 0.8532 | 0.5554 |
| SUBGDIFF | 200 | DDPM (ours) | 85.53▲ | 88.99▲ | 0.2994▲ | 0.3033▲ | 47.76▲ | 45.89▲ | 0.6971▲ | 0.5118▲ |

## 5.2 CONFORMATION GENERATION

To evaluate the generation efficiency and generation performance, we conduct the experiments with various time steps, including 5000, 500 and 200, to compare SUBGDIFF with GeoDiff.

**Dataset.** Following prior works (Xu et al., 2022; 2021a), we utilize the GEOM-QM9 (Ramakrishnan et al., 2014) and GEOM-Drugs (Axelrod & Gomez-Bombarelli, 2022) datasets. The former dataset comprises small molecules of up to 9 heavy atoms, while the larger drug-like compounds. We reuse the data split provided by Xu et al. (2022). For both datasets, the training dataset comprises $40,000$ molecules, each with $5$ conformations, resulting in $200,000$ conformations in total. The test split includes 200 distinctive molecules, with $14,324$ conformations for Drugs and $22,408$ conformations for QM9.

Table 3: Results on the **GEOM-QM9** dataset. The threshold $\delta = 0.5$Å

| Models | Train data | COV-R (%) ↑ | | MAT-R (Å) ↓ | |
|---|---|---|---|---|---|
| | | Mean | Median | Mean | Median |
| CVGAE | QM9 | 0.09 | 0.00 | 1.6713 | 1.6088 |
| GRAPHDG | QM9 | 73.33 | 84.21 | 0.4245 | 0.3973 |
| CGCF | QM9 | 78.05 | 82.48 | 0.4219 | 0.3900 |
| CONFVAE | QM9 | 77.84 | 88.20 | 0.4154 | 0.3739 |
| GEOMOL | QM9 | 71.26 | 72.00 | 0.3731 | 0.3731 |
| GEODIFF | Drugs | 74.94 | 79.15 | 0.3492 | 0.3392 |
| **SUBGDIFF** | Drugs | **83.50** | **88.70** | **0.3116** | **0.3075** |

**Denoising networks.** Following Xu et al. (2022), we use an equivariant convolutional network GFN as the denoising network for conformation generation and self-supervised learning tasks. The description of evaluation metrics and model architecture are deferred to the Appendix F.

**Results.** The results on the GEOM-QM9 dataset are reported in Table 4. From the results, we get the following observations: **(1):** SUBGDIFF significantly outperforms the baselines on COV-R and MAT-R, indicating the SUBGDIFF tends to explore more possible conformations. **(2):** SUBGDIFF consistently outperforms GEODIFF when adopting 200 and 500 sampling steps, demonstrating the competitive sampling efficiency of our method. Surprisingly, SUBGDIFF with 500 steps achieves much better performance than GEODIFF with 5000 steps on 5 out of 8 metrics, which implies our method can accelerate the sampling efficiency (10x).

**Domain generalization.** We design two cross-domain tasks: (1) Training on QM9 (small molecular with up to 9 heavy atoms) and testing on Drugs (medium-sized organic compounds); (2) Training on Drugs and testing on QM9. The results are depicted in Table 3 and Table 10 (refer to Appendix), respectively. The results suggest that SUBGDIFF consistently outperforms GEODIFF by a large margin, demonstrating the introduced mask effectively enhances the robustness and generalization of the denoising network.

## 6 CONCLUSION

We first present a masked diffusion framework, which involves the subset constraint in the diffusion model by introducing the mask vector to the forward process. The framework is a model-agnostic approach that can used for any diffusion model built on European space. Further, a novel diffusion model SUBGDIFF is developed for molecular conformation generation and self-supervised representation learning. SUBGDIFF is the first diffusion method that fuses the substructure into training and sampling. Benefiting from the substructure, SUBGDIFF effectively boosts the perception of molecular substructure in the denoising network, thereby achieving state-of-the-art performance at conformation generation and 3D property prediction tasks. There are several exciting avenues for future work. The mask distribution is so flexible that we can incorporate chemical prior knowledge into efficient subgraph sampling. Besides, the proposed SUBGDIFF can be generalized to proteins such that the denoising network can learn meaningful secondary structures.

## ETHICS STATEMENT

In this work, we propose a novel diffusion model for molecular conformation and representation learning, where no human subject is related.

## REPRODUCIBILITY STATEMENT

We summarize the efforts made to ensure reproducibility in this work. (1) Datasets: we use the public datasets QM9 where the processing details are included in sec 5 and Appendix F. (2) Model Training: We provide the training details (including hyper-parameters settings) in Appendix F.1 and the procedure of training in Algorithms 1 and the procedure of sampling in Algorithms 2.

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

# Appendix

## CONTENTS

## A  MORE RELATED WORKS

**Conformation generation.**    Recently, various deep generative models have been proposed for conformation generation. CVGAE (Mansimov et al., 2019) first proposed a VAE model to directly generate 3D atomic coordinates. GRAPHDG (Simm & Hernandez-Lobato, 2020) and CGCF (Xu et al., 2021a) proposed to predict the interatomic distance matrix by VAE and Flow respectively, and then solve the geometry through the Distance Geometry (DG) technique (Liberti et al., 2014). CONFVAE further improves this pipeline by designing an end-to-end framework via bilevel optimization (Xu et al., 2021b). CONFGF (Shi et al., 2021; Luo et al., 2021) proposed to learn the gradient of the log-likelihood *w.r.t* coordinates via denoising score matching (DSM) (Song & Ermon, 2019; 2020). GEOMOL (Ganea et al., 2021) proposed to reconstruct the local and global structures of the conformation from a set of geometric quantities (*i.e.* length and angles). Most recently, diffusion-based methods have shown competitive performance. Torsional Diffusion (Jing et al., 2022) raises a diffusion process on the hypertorus defined by torsion angles. However, it is not suitable as a self-supervised learning technique due to the lack of local information (length and angle of bonds). GEODIFF (Xu et al., 2022) and EDM (Hoogeboom et al., 2022) generated 3D molecular by doing a conventional diffusion model (Ho et al., 2020) on atomic coordinates or atom feature. However, these methods view the atoms of molecules as separate particles, without considering the critical dependence between atoms, especially the substructure (Marinova et al., 2018).

**Remark A.1 (discussion with MDM, MDSM  and SSSD )**

*Previous works also share the similar idea of masked diffusion, such as MDM (Pan et al., 2023), MDSM (Lei et al., 2023) and SSSD (Alcaraz & Strodthoff, 2022). However, the difference between our MaskedDiff and them mainly lies in the following two aspects: i) Usage: the mask matrix/vector in SSSD and MDSM is fixed in all training steps, which means some segments of the data (time series or images) will never be diffused. But our method samples the $\mathbf{s}_t \sim p_{\mathbf{s}_t}(\mathcal{S})$ at each time step, hence a suitable discrete distribution $p(\mathcal{S})$ can ensure that almost all nodes can be added noise. ii) Purpose: MDSM and MDM concentrate on self-supervised pre-training, while MaskedDiff serves as a potent generative model (see next subsection) and self-supervised pre-training algorithm. Notably, when $\mathbf{s}_t = \mathbf{s}_0, \forall t,$ MaskedDiff can recover to MDSM.*

**Dissusion with D3FG,DIffPAck and GraphARM**

- Compare to D3FG (Lin et al., 2023): D3FG adopts three different diffusion models (D3PM,DDPM,and SO(3) Diffusion) to generate three different parts of molecules(linkerr types, center atom position, and functional group orientations), respectively. In general, these three parts can also be viewed as three subgraphs(subset). Essentially, D3FG firstly selects the subgraph and only injects noise on the fixed subgraph during the entire diffusion process, while our method is capable of selecting different subgraphs from a mask distribution in each time step during the forward process. Further, our model is actually a single diffusion model that can enhance the denoising network to perceive substructure information by fusing a mask variable in it.

- Compare to DiffPACK(Zhang et al., 2023): DIffPAck is an Autoregressive generative method that predicts the torsional angle $\chi_i(i = 1, 2, .., 4)$ of protein side-chains with the condition $\chi_{1,...,i-1}$, where $\chi_i$ is a predefined subset of atoms. It uses a torsional-based diffusion model to approximate the distribution $p(\chi_i|\chi_{1,...,i-1})$, in which every subset $\chi_i$ needs a separate score network to estimate. Essentially, it can be viewed as selecting a subset first and then only adding noise on the fixed subset during the entire diffusion process. In contrast, our method proposes to randomly sample a subset from mask distribution $p(S)$ in *each time-step* during the forward process, which is more flexible and cost-effective (requires only a score network and a subgraph predictor).

- Compare to GraphARM (Kong et al., 2023b): Kong et al. (2023b) proposes an autoregressive diffusion model GraphARM, which absorbs one node in each time-step by masking it along with its connecting edges during the forward process. Differently from GraphARM[3], our SugGDiff selects a subgraph in each time step to inject the Gaussian noise, which is equivalent to masking several nodes during the forward process. In addition, the number of steps in GraphARM must be the same as the number of nodes due to the usage of the absorbing state, while our method can set any time-step during diffusion theoretically since we use the real-value Gaussian noise.

## B  PROOF OF LAMMA 4.1

**Lemma 4.1** *Assume the forward and reverse processes of the diffusion model are both Markov chains. Given the forward Gaussian distribution $q(R^t|R^{t-1}, R^0) = \mathcal{N}(R^t; \mu_1 R^{t-1}, \sigma_1^2 \boldsymbol{I})$, $q(R^{t-1}|R^0) = \mathcal{N}(R^{t-1}; \mu_2 R^0, \sigma_2^2 \boldsymbol{I})$ and $\epsilon_0 \sim \mathcal{N}(\mathbf{0}, \mathbf{I})$, the distribution $q(R^{t-1}|R^t, R^0)$ is*

$$q(R^{t-1}|R^t, R^0) \propto \mathcal{N}(R^{t-1}; \frac{1}{\mu_1}(R^t - \frac{\sigma_1^2}{\sqrt{\mu_1^2\sigma_2^2 + \sigma_1^2}}\epsilon_0), \frac{\sigma_1^2\sigma_2^2}{\mu_1^2\sigma_2^2 + \sigma_1^2}\boldsymbol{I}). \tag{5}$$

*Parameterizing $p_\theta(R^{t-1}|R^t)$ in the reverse process as $\mathcal{N}(R^{t-1}; \frac{1}{\mu_1}(R^t - \frac{\sigma_1^2}{\sqrt{\mu_1^2\sigma_2^2 + \sigma_1^2}}\epsilon_\theta(R^t, t)), \frac{\sigma_1^2\sigma_2^2}{\mu_1^2\sigma_2^2 + \sigma_1^2}\boldsymbol{I})$, the training objective of the DPM can be written as*

$$\mathcal{L}(\theta) = \mathbb{E}_{t,R^0,\epsilon}\Big[\frac{\sigma_1^2}{2\mu_1^2\sigma_2^2}\|\epsilon - \epsilon_\theta(\mu_1\mu_2 R^0 + \sqrt{\mu_1^2\sigma_2^2 + \sigma_1^2}\epsilon, t)\|^2\Big], \tag{6}$$

*and the sampling (reverse) process is*

$$R^{t-1} = \frac{1}{\mu_1}\left(R^t - \frac{\sigma_1^2}{\sqrt{\mu_1^2\sigma_2^2 + \sigma_1^2}}\epsilon_\theta(R^t, t)\right) + \frac{\sigma_1\sigma_2}{\sqrt{\mu_1^2\sigma_2^2 + \sigma_1^2}}z, \quad z \sim \mathcal{N}(\mathbf{0}, \mathbf{I}) \tag{7}$$

**Proof:** Given the forward Gaussian distribution $q(R^t|R^{t-1}, R^0) = \mathcal{N}(R^t; \mu_1 R^{t-1}, \sigma_1^2 I)$ and $q(R^{t-1}|R^0) = \mathcal{N}(R^{t-1}; \mu_2 R^0, \sigma_2^2 I)$, we have

$$q(R^t|R^0) = q(R^t|R^{t-1}, R^0)q(R^{t-1}|R^0) = \mathcal{N}(R^t; \mu_1\mu_2 R^0, (\sigma_1^2 + \mu_1^2\sigma_2^2)I) \tag{17}$$

From the DDPM, we know training a diffusion model should optimize the ELBO of the data

$$\log p(\mathbf{R}) \geq \mathbb{E}_{q(\mathbf{R}^{1:T}|\mathbf{R}^0)} \left[ \log \frac{p(\mathbf{R}^{0:T})}{q(\mathbf{R}^{1:T}|\mathbf{R}^0)} \right] \tag{18}$$

$$= \underbrace{\mathbb{E}_{q(\mathbf{R}^1|\mathbf{R}^0)} \left[ \log p_{\boldsymbol{\theta}}(\mathbf{R}^0|\mathbf{R}^1) \right]}_{\text{reconstruction term}} - \underbrace{D_{\text{KL}}(q(\mathbf{R}^T|\mathbf{R}^0) \parallel p(\mathbf{R}^T))}_{\text{prior matching term}} - \sum_{t=2}^{T} \underbrace{\mathbb{E}_{q(\mathbf{R}^t|\mathbf{R}^0)} \left[ D_{\text{KL}}(q(\mathbf{R}^{t-1}|\mathbf{R}^t,\mathbf{R}^0) \parallel p_{\boldsymbol{\theta}}(\mathbf{R}^{t-1}|\mathbf{R}^t)) \right]}_{\text{denoising matching term}} \tag{19}$$

To compute the KL divergence $D_{\text{KL}}(q(\mathbf{R}^{t-1}|\mathbf{R}^t, \mathbf{R}^0) \parallel p_{\boldsymbol{\theta}}(\mathbf{R}^{t-1}|\mathbf{R}^t))$, we first rewrite $q(\mathbf{R}^{t-1}|\mathbf{R}^t, \mathbf{R}^0)$ by Bayes rule

$$q(R^{t-1}|R^t, R^0) = \frac{q(R^t|R^{t-1}, R^0)q(R^{t-1}|R^0)}{q(R^t|R^0)} \tag{20}$$

$$= \frac{\mathcal{N}(R^t; \mu_1 R^{t-1}, \sigma_1^2 \mathbf{I})\mathcal{N}(R^{t-1}; \mu_2 R^0, \sigma_2^2 \mathbf{I})}{\mathcal{N}(R^t; \mu_1 \mu_2 R^0, (\sigma_1^2 + \mu_1^2 \sigma_2^2)\mathbf{I})} \tag{21}$$

$$\propto \exp \left\{ - \left[ \frac{(R^t - \mu_1 R^{t-1})^2}{2\sigma_1^2} + \frac{(R^{t-1} - \mu_2 R^0)^2}{2\sigma_2^2} - \frac{(R^t - \mu_1 \mu_2 R^0)^2}{2(\sigma_1^2 + \mu_1^2 \sigma_2^2)} \right] \right\} \tag{22}$$

$$= \exp \left\{ -\frac{1}{2} \left[ \frac{(R^t - \mu_1 R^{t-1})^2}{\sigma_1^2} + \frac{(R^{t-1} - \mu_2 R^0)^2}{\sigma_2^2} - \frac{(R^t - \mu_1 \mu_2 R^0)^2}{\sigma_1^2 + \mu_1^2 \sigma_2^2} \right] \right\} \tag{23}$$

$$= \exp \left\{ -\frac{1}{2} \left[ \frac{(-2\mu_1 R^t R^{t-1} + \mu_1^2 (R^{t-1})^2)}{\sigma_1^2} + \frac{((R^{t-1})^2 - 2\mu_2 R^{t-1} R^0)}{\sigma_2^2} + C(R^t, R^0) \right] \right\} \tag{24}$$

$$\propto \exp \left\{ -\frac{1}{2} \left[ -\frac{2\mu_1 R^t R^{t-1}}{\sigma_1^2} + \frac{\mu_1^2 (R^{t-1})^2}{\sigma_1^2} + \frac{(R^{t-1})^2}{\sigma_2^2} - \frac{2\mu_2 R^{t-1} R^0}{\sigma_2^2} \right] \right\} \tag{25}$$

$$= \exp \left\{ -\frac{1}{2} \left[ (\frac{\mu_1^2}{\sigma_1^2} + \frac{1}{\sigma_2^2})(R^{t-1})^2 - 2 \left( \frac{\mu_1 R^t}{\sigma_1^2} + \frac{\mu_2 R^0}{\sigma_2^2} \right) R^{t-1} \right] \right\} \tag{26}$$

$$= \exp \left\{ -\frac{1}{2} \left[ \frac{\sigma_1^2 + \mu_1^2 \sigma_2^2}{\sigma_1^2 \sigma_2^2}(R^{t-1})^2 - 2 \left( \frac{\mu_1 R^t}{\sigma_1^2} + \frac{\mu_2 R^0}{\sigma_2^2} \right) R^{t-1} \right] \right\} \tag{27}$$

$$= \exp \left\{ -\frac{1}{2} \left( \frac{\sigma_1^2 + \mu_1^2 \sigma_2^2}{\sigma_1^2 \sigma_2^2} \right) \left[ (R^{t-1})^2 - 2\frac{\left( \frac{\mu_1 R^t}{\sigma_1^2} + \frac{\mu_2 R^0}{\sigma_2^2} \right)}{\frac{\sigma_1^2 + \mu_1^2 \sigma_2^2}{\sigma_1^2 \sigma_2^2}} R^{t-1} \right] \right\} \tag{28}$$

$$= \exp \left\{ -\frac{1}{2} \left( \frac{\sigma_1^2 + \mu_1^2 \sigma_2^2}{\sigma_1^2 \sigma_2^2} \right) \left[ (R^{t-1})^2 - 2\frac{\left( \frac{\mu_1 R^t}{\sigma_1^2} + \frac{\mu_2 R^0}{\sigma_2^2} \right) \sigma_1^2 \sigma_2^2}{\sigma_1^2 + \mu_1^2 \sigma_2^2} R^{t-1} \right] \right\} \tag{29}$$

$$= \exp \left\{ -\frac{1}{2} \left( \frac{1}{\frac{\sigma_1^2 \sigma_2^2}{\sigma_1^2 + \mu_1^2 \sigma_2^2}} \right) \left[ (R^{t-1})^2 - 2\frac{\mu_1 \sigma_2^2 R^t + \mu_2 \sigma_1^2 R^0}{\sigma_1^2 + \mu_1^2 \sigma_2^2} R^{t-1} \right] \right\} \tag{30}$$

$$\propto \mathcal{N}(R^{t-1}; \underbrace{\frac{\mu_1 \sigma_2^2 R^t + \mu_2 \sigma_1^2 R^0}{\sigma_1^2 + \mu_1^2 \sigma_2^2}}_{\mu_q(R^t, R^0)}, \underbrace{\frac{\sigma_1^2 \sigma_2^2}{\sigma_1^2 + \mu_1^2 \sigma_2^2}}_{\boldsymbol{\Sigma}_q(t)}\mathbf{I}) \tag{31}$$

We can rewrite our variance equation as $\boldsymbol{\Sigma}_q(t) = \sigma_q^2(t)\mathbf{I}$, where:

$$\sigma_q^2(t) = \frac{\sigma_1^2 \sigma_2^2}{\sigma_1^2 + \mu_1^2 \sigma_2^2} \tag{32}$$

From equation 17, we have the relationship between $R^t$ and $R^0$:

$$R^0 = \frac{R^t - \sqrt{\sigma_1^2 + \mu_1^2 \sigma_2^2}\epsilon}{\mu_1 \mu_2} \tag{33}$$

Substituting this into $\mu_q(R^t, R^0)$, we can get

$$\mu_q(R^t, R^0) = \frac{\mu_1\sigma_2^2 R^t + \mu_2\sigma_1^2 R^0}{\sigma_1^2 + \mu_1^2\sigma_2^2} \tag{34}$$

$$= \frac{\mu_1\sigma_2^2 R^t + \mu_2\sigma_1^2 \frac{R^t - \sqrt{\sigma_1^2 + \mu_1^2\sigma_2^2}\epsilon}{\mu_1\mu_2}}{\sigma_1^2 + \mu_1^2\sigma_2^2} \tag{35}$$

$$= \frac{\mu_1\sigma_2^2 R^t + \frac{\sigma_1^2 R^2}{\mu_1} - \frac{\sigma_1^2\sqrt{\sigma_1^2 + \mu_1^2\sigma_2^2}\epsilon}{\mu_1}}{\sigma_1^2 + \mu_1^2\sigma_2^2} \tag{36}$$

$$= \frac{1}{\mu_1} R^t - \frac{\sigma_1^2}{\mu_1\sqrt{\sigma_1^2 + \mu_1^2\sigma_2^2}}\epsilon \tag{37}$$

Thus,

$$q(R^{t-1}|R^t, R^0) \propto \mathcal{N}(R^{t-1}; \underbrace{\frac{1}{\mu_1}(R^t - \frac{\sigma_1^2}{\sqrt{\sigma_1^2 + \mu_1^2\sigma_2^2}}\epsilon)}_{\mu_q(R^t, t)}, \underbrace{\frac{\sigma_1^2\sigma_2^2}{\sigma_1^2 + \mu_1^2\sigma_2^2}\mathbf{I}}_{\Sigma_q(t)}) \tag{38}$$

Parameterizing $p_\theta(R^{t-1}|R^t)$ in the reverse process as $\mathcal{N}(R^{t-1}; \frac{1}{\mu_1}(R^t - \frac{\sigma_1^2}{\sqrt{\mu_1^2\sigma_2^2 + \sigma_1^2}}\epsilon_\theta(R^t, t)), \frac{\sigma_1^2\sigma_2^2}{\mu_1^2\sigma_2^2 + \sigma_1^2}\mathbf{I})$, and the corresponding optimization problem becomes:

$$\arg\min_\theta D_{\text{KL}}(q(R^{t-1}|R^t, R^0) \| p_\theta(R^{t-1}|R^t))$$

$$= \arg\min_\theta D_{\text{KL}}(\mathcal{N}(R^{t-1}; \mu_q, \Sigma_q(t)) \| \mathcal{N}(R^{t-1}; \mu_\theta, \Sigma_q(t))) \tag{39}$$

$$= \arg\min_\theta \frac{1}{2\sigma_q^2(t)}\left[\left\|\frac{\sigma_1^2}{\mu_1\sqrt{\sigma_1^2 + \mu_1^2\sigma_2^2}}\epsilon_0 - \frac{\sigma_1^2}{\mu_1\sqrt{\sigma_1^2 + \mu_1^2\sigma_2^2}}\epsilon_\theta(R^t, t)\right\|_2^2\right] \tag{40}$$

$$= \arg\min_\theta \frac{1}{2\sigma_q^2(t)}\left[\left\|\frac{\sigma_1^2}{\mu_1\sqrt{\sigma_1^2 + \mu_1^2\sigma_2^2}}(\epsilon_0 - \hat\epsilon_\theta(R^t, t))\right\|_2^2\right] \tag{41}$$

$$= \arg\min_\theta \frac{1}{2\sigma_q^2(t)}\left(\frac{\sigma_1^2}{\mu_1\sqrt{\sigma_1^2 + \mu_1^2\sigma_2^2}}\right)^2\left[\left\|\epsilon_0 - \hat\epsilon_\theta(R^t, t)\right\|_2^2\right] \tag{42}$$

$$= \arg\min_\theta \frac{\sigma_1^2}{2\sigma_2^2\mu_1^2}\left[\left\|\epsilon_0 - \hat\epsilon_\theta(R^t, t)\right\|_2^2\right] \tag{43}$$

Therefore, the training objective of the DPM can be written as

$$\mathcal{L}(\theta) = \mathbb{E}_{t, R^0, \epsilon}[\frac{\sigma_1^2}{2\mu_1^2\sigma_2^2}\|\epsilon - \epsilon_\theta(\mu_1\mu_2 R^0 + \sqrt{\mu_1^2\sigma_2^2 + \sigma_1^2}\epsilon, t)\|^2], \tag{44}$$

During the reverse process, we sample $R^{t-1} \sim p_\theta(R^{t-1}|R^t)$. Formally, the sampling (reverse) process is

$$R^{t-1} = \frac{1}{\mu_1}\left(R^t - \frac{\sigma_1^2}{\sqrt{\mu_1^2\sigma_2^2 + \sigma_1^2}}\epsilon_\theta(R^t, t)\right) + \frac{\sigma_1\sigma_2}{\sqrt{\mu_1^2\sigma_2^2 + \sigma_1^2}}z, \quad z \sim \mathcal{N}(\mathbf{0}, \mathbf{I}) \tag{45}$$

## C  DERIVATIONS OF TRAINING OBJECTIVES

### C.1  MASKEDDIFF

Here, we utilize the binary characteristic of the mask vector to derive the ELBO for MaskedDiff, and we also provide a general proof in sec. C.2:

$$\log p(R^0) \geq \mathbb{E}_{q(R^{1:T}, s_{1:T}|R^0)}\left[\log \frac{p(R^{0:T}, s_{1:T})}{q(R^{1:T}|R^0, s_{1:T})q(s_{1:T})}\right] \tag{46}$$

$$= \mathbb{E}_{q(R^{1:T}, s_{1:T}|R^0)} \left[ \log \frac{p(R^T) \prod_{t=1}^{T} p_{\boldsymbol{\theta}}(R^{t-1}, s_t|R^t)}{\prod_{t=1}^{T} q(R^t|R^{t-1}, s_t)q(s_t)} \right] \tag{47}$$

$$= \mathbb{E}_{q(R^{1:T}, s_{1:T}|R^0)} \left[ \log \frac{p(R^T) \prod_{t=1}^{T} p_{\boldsymbol{\theta}}(R^{t-1}|R^t)p_{\theta}(s_t|R^t)}{\prod_{t=1}^{T} q(R^t|R^{t-1}, s_t)q(s_t)} \right] \tag{48}$$

$$= \mathbb{E}_{q(R^{1:T}, s_{1:T}|R^0)} \left[ \log \frac{\prod_{t=1}^{T} p_{\theta}(s_t|R^t)}{\prod_{t=1}^{T} q(s_t)} + \log \frac{p(R^T) \prod_{t=1}^{T} p_{\boldsymbol{\theta}}(R^{t-1}|R^t)}{\prod_{t=1}^{T} q(R^t|R^{t-1}, s_t)} \right] \tag{49}$$

$$= \underbrace{\mathbb{E}_{q(R^{1:T}, s_{1:T}|R^0)} \left[ \sum_{t=1}^{T} \log \frac{p_{\theta}(s_t|R^t)}{q(s_t)} \right]}_{\text{mask prediction term}} + \mathbb{E}_{q(R^{1:T}, s_{1:T}|R^0)} \left[ \log \frac{p(R^T) \prod_{t=1}^{T} p_{\boldsymbol{\theta}}(R^{t-1}|R^t)}{\prod_{t=1}^{T} q(R^t|R^{t-1}, s_t)} \right]$$

$$\tag{50}$$

$$\tag{51}$$

The first term is mask prediction while the second term is similar to the ELBO of the classical diffusion model. The only difference is the $s_t$ in $q(R^t|R^{t-1}, s_t)$. According to Bayes rule, we can rewrite each transition as:

$$q(R^t|R^{t-1}, R^0, s_t) = \begin{cases} \frac{q(R^{t-1}|R^t, R^0)q(R^t|R^0)}{q(R^{t-1}|R^0)}, & \text{if } s_t = 1 \\ \delta_{R_{t-1}}(R_t). & \text{if } s_t = 0 \end{cases} \tag{52}$$

where $\delta_a(x) := \delta(x - a)$ is Dirac delta function, that is, $\delta_a(x) = 0$ if $x \neq a$ and $\int_{-\infty}^{\infty} \delta_a(x)dx = 1$. Without loss of generality, assume that $s_1$ and $s_T$ both equal 1. Armed with this new equation, we drive the second term:

$$\mathbb{E}_{q(R^{1:T}, s_{1:T}|R^0)} \left[ \log \frac{p(R^T) \prod_{t=1}^{T} p_{\boldsymbol{\theta}}(R^{t-1}|R^t)}{\prod_{t=1}^{T} q(R^t|R^{t-1}, s_t)} \right] \tag{53}$$

$$= \mathbb{E}_{q(R^{1:T}, s_{1:T}|R^0)} \left[ \log \frac{p(R^T)p_{\boldsymbol{\theta}}(R^0|R^1) \prod_{t=2}^{T} p_{\boldsymbol{\theta}}(R^{t-1}|R^t)}{q(R^1|R^0) \prod_{t=2}^{T} q(R^t|R^{t-1}, s_t)} \right] \tag{54}$$

$$= \mathbb{E}_{q(R^{1:T}, s_{1:T}|R^0)} \left[ \log \frac{p(R^T)p_{\boldsymbol{\theta}}(R^0|R^1) \prod_{t=2}^{T} p_{\boldsymbol{\theta}}(R^{t-1}|R^t)}{q(R^1|R^0) \prod_{t=2}^{T} q(R^t|R^{t-1}, R^0, s_t)} \right] \tag{55}$$

$$= \mathbb{E}_{q(R^{1:T}, s_{1:T}|R^0)} \left[ \log \frac{p_{\boldsymbol{\theta}}(R^T)p_{\boldsymbol{\theta}}(R^0|R^1)}{q(R^1|R^0)} + \log \prod_{t=2}^{T} \frac{p_{\boldsymbol{\theta}}(R^{t-1}|R^t)}{q(R^t|R^{t-1}, R^0, s_t)} \right] \tag{56}$$

$$= \mathbb{E}_{q(R^{1:T}, s_{1:T}|R^0)} \left[ \log \frac{p(R^T)p_{\boldsymbol{\theta}}(R^0|R^1)}{q(R^1|R^0)} + \log \prod_{t \in \{t|s_t=1\}} \frac{p_{\boldsymbol{\theta}}(R^{t-1}|R^t)}{\frac{q(R^{t-1}|R^t, R^0)q(R^t|R^0)}{q(R^{t-1}|R^0, s_1)}} + \log \prod_{t \in \{t|s_t=0\}} \frac{p_{\boldsymbol{\theta}}(R^{t-1}|R^t)}{\delta_{R^{t-1}}(R^t)} \right] \tag{57}$$

$$= \mathbb{E}_{q(R^{1:T}|R^0)} \left[ \log \frac{p(R^T)p_{\boldsymbol{\theta}}(R^0|R^1)}{q(R^1|R^0)} + \log \prod_{t \in \{t|s_t=0\}} \frac{p_{\boldsymbol{\theta}}(R^{t-1}|R^t)}{\delta_{R^{t-1}}(R^t)} + \log \prod_{t \in \{t|s_t=1\}} \frac{p_{\boldsymbol{\theta}}(R^{t-1}|R^t)}{\frac{q(R^{t-1}|R^t, R^0)\cancel{q(R^t|R^0)}}{\cancel{q(R^{t-1}|R^0)}}} \right] \tag{58}$$

$$= \mathbb{E}_{q(R^{1:T}|R^0)} \left[ \log \prod_{t \in \{t|s_t=0\}} \frac{p_{\boldsymbol{\theta}}(R^{t-1}|R^t)}{\delta_{R^{t-1}}(R^t)} + \log \frac{p(R^T)p_{\boldsymbol{\theta}}(R^0|R^1)}{\cancel{q(R^1|R^0)}} + \log \frac{\cancel{q(R^1|R^0)}}{q(R^T|R^0)} + \log \prod_{t \in \{t|s_t=1\}} \frac{p_{\boldsymbol{\theta}}(R^{t-1}|R^t)}{q(R^{t-1}|R^t, R^0)} \right] \tag{59}$$

$$= \mathbb{E}_{q(R^{1:T}|R^0)} \left[ \sum_{t \in \{t|s_t=0\}} \log \frac{p_{\boldsymbol{\theta}}(R^{t-1}|R^t)}{\delta_{R^{t-1}}(R^t)} + \log \frac{p(R^T)p_{\boldsymbol{\theta}}(R^0|R^1)}{q(R^T|R^0)} + \sum_{t \in \{t|s_t=1\}} \log \frac{p_{\boldsymbol{\theta}}(R^{t-1}|R^t)}{q(R^{t-1}|R^t, R^0)} \right] \tag{60}$$

$$= \sum_{t \in \{t|s_t=0\}} \mathbb{E}_{q(R^{1:T}|R^0)} \left[ \log \frac{p_{\boldsymbol{\theta}}(R^{t-1}|R^t)}{\delta_{R^{t-1}}(R^t)} \right] + \mathbb{E}_{q(R^{1:T}|R^0)} \left[ \log p_{\boldsymbol{\theta}}(R^0|R^1) \right] \tag{61}$$

$$+ \mathbb{E}_{q(R^{1:T}|R^0)} \left[ \log \frac{p(R^T)}{q(R^T|R^0)} \right] + \sum_{t \in \{t|s_t=1\}} \mathbb{E}_{q(R^{1:T}|R^0)} \left[ \log \frac{p_{\boldsymbol{\theta}}(R^{t-1}|R^t)}{q(R^{t-1}|R^t, R^0)} \right] \tag{62}$$

$$= \sum_{t\in\{t|s_t=0\}} \mathbb{E}_{q(R^{1:T}|R^0)}\left[\log\frac{p_{\boldsymbol{\theta}}(R^{t-1}|R^t)}{\delta_{R^{t-1}}(R^t)}\right] + \mathbb{E}_{q(R^1|R^0)}\left[\log p_{\boldsymbol{\theta}}(R^0|R^1)\right] \tag{63}$$

$$+\mathbb{E}_{q(R^T|R^0)}\left[\log\frac{p(R^T)}{q(R^T|R^0)}\right] + \sum_{t\in\{t|s_t=1\}} \mathbb{E}_{q(R^t,R^{t-1}|R^0)}\left[\log\frac{p_{\boldsymbol{\theta}}(R^{t-1}|R^t)}{q(R^{t-1}|R^t,R^0)}\right] \tag{64}$$

$$= \underbrace{\sum_{t\in\{t|s_t=0\}} \mathbb{E}_{q(R^{1:T}|R^0)}\left[\log\frac{p_{\boldsymbol{\theta}}(R^{t-1}|R^t)}{\delta_{R^{t-1}}(R^t)}\right]}_{\textbf{decay term}} + \underbrace{\mathbb{E}_{q(R^1|R^0)}\left[\log p_{\boldsymbol{\theta}}(R^0|R^1)\right]}_{\text{reconstruction term}} \tag{65}$$

$$- \underbrace{D_{\text{KL}}(q(R^T|R^0)\parallel p(R^T))}_{\text{prior matching term}} - \sum_{t\in\{t|s_t=1\}} \underbrace{\mathbb{E}_{q(R^t|R^0)}\left[D_{\text{KL}}(q(R^{t-1}|R^t,R^0)\parallel p_{\boldsymbol{\theta}}(R^{t-1}|R^t))\right]}_{\text{denoising matching term}} \tag{66}$$

Here, the *decay term* represents the terms with $s_t=0$, which are unnecessary to minimize when we set $p_{\boldsymbol{\theta}}(R^{t-1}|R^t):=\delta_{R^{t-1}}(R^t)$. Eventually, the ELOB can be rewritten as follows:

$$\log p(R^0) \geq \sum_{t=1}^T \underbrace{\mathbb{E}_{q(R^{1:T}|R^0)}\left[\log\frac{p_{\vartheta}(s_t|R^t)}{q(s_t)}\right]}_{\text{mask prediction term}} + \underbrace{\mathbb{E}_{q(R^1|R^0)}\left[\log p_{\boldsymbol{\theta}}(R^0|R^1)\right]}_{\text{reconstruction term}}$$

$$- \underbrace{D_{\text{KL}}(q(R^T|R^0)\parallel p(R^T))}_{\text{prior matching term}} - \sum_{t\in\{t|s_t=1\}} \underbrace{\mathbb{E}_{q(R^t|R^0)}\left[D_{\text{KL}}(q(R^{t-1}|R^t,R^0)\parallel p_{\boldsymbol{\theta}}(R^{t-1}|R^t))\right]}_{\text{denoising matching term}} \tag{67}$$

The mask prediction term can be implemented by a node classifier and the denoising matching term can be calculated via Lemma 4.1. In detail,

$$q(R^t|R^{t-1},R^0) = \mathcal{N}(R^{t-1},\sqrt{1-\beta_t s_t}R^{t-1},(\beta_t s_t)\mathbf{I}), \tag{68}$$

$$q(R^{t-1}|R^0) = \mathcal{N}(R^{t-1},\sqrt{\bar{\gamma}_{t-1}}R^0,(1-\bar{\gamma}_{t-1})\mathbf{I}). \tag{69}$$

Thus, the training objective of MaskedDiff is:

$$\mathcal{L}(\theta,\vartheta) = \mathbb{E}_{t,R^0,\epsilon}\left[\frac{s_t\beta_t}{2(1-s_t\beta_t)(1-\bar{\gamma}_{t-1})}\|\epsilon - \epsilon_\theta(\sqrt{\bar{\gamma}_t}R^0 + \sqrt{(1-\bar{\gamma}_t)}\epsilon,t,\mathcal{G})\|^2 + \lambda\text{BCE}(\mathbf{s}_t,s_\vartheta(\mathcal{G},\mathbf{R}^t,t))\right] \tag{70}$$

In order to recover the existing work, we omit the mask prediction term (i.e. Let $p_\theta(s_t|R^t):=q(s_t)$) of MaskedDiff in the main text.

## C.2 ELBO

Here, we can derive the ELBO for SUBGDIFF:

$$\log p(R^0) = \log \int\int p(R^{0:T},s_{1:T})dR^{1:T}ds_{1:T} \tag{71}$$

$$= \log \int\int \frac{p(R^{0:T},s_{1:T})q(R^{1:T},s_{1:T}|R^0)}{q(R^{1:T},s_{1:T}|R^0)}dR^{1:T}ds_{1:T} \tag{72}$$

$$= \log \int\int \left[\frac{p(R^{0:T},s_{1:T})q(R^{1:T}|R^0,s_{1:T})q(s_{1:T})}{q(R^{1:T},s_{1:T}|R^0)}\right]dR^{1:T}ds_{1:T} \tag{73}$$

$$= \log \mathbb{E}_{q(s_{1:T})}\mathbb{E}_{q(R^{1:T}|R^0,s_{1:T})}\left[\frac{p(R^{0:T},s_{1:T}))}{q(R^{1:T},s_{1:T}|R^0)}\right] \tag{74}$$

$$\geq \mathbb{E}_{q(R^{1:T}|R^0,s_{1:T})}\left[\log\mathbb{E}_{q(s_{1:T})}\frac{p(R^{0:T},s_{1:T})}{q(R^{1:T}|R^0,s_{1:T})q(s_{1:T})}\right] \tag{75}$$

$$\geq \mathbb{E}_{q(R^{1:T},s_{1:T}|R^0)}\left[\log\frac{p(R^T)\prod_{t=1}^T p_{\boldsymbol{\theta}}(R^{t-1},s_t|R^t)}{\prod_{t=1}^T q(R^t|R^{t-1},s_t)q(s_t)}\right] \tag{76}$$

$$= \mathbb{E}_{q(R^{1:T},s_{1:T}|R^0)}\left[\log\frac{p(R^T)\prod_{t=1}^T p_{\boldsymbol{\theta}}(R^{t-1}|R^t)p_\theta(s_t|R^t)}{\prod_{t=1}^T q(R^t|R^{t-1},s_t)q(s_t)}\right] \tag{77}$$

$$= \mathbb{E}_{q(R^{1:T}, s_{1:T}|R^0)} \left[ \log \frac{\prod_{t=1}^{T} p_\theta(s_t|R^t)}{\prod_{t=1}^{T} q(s_t)} + \log \frac{p(R^T) \prod_{t=1}^{T} p_\theta(R^{t-1}|R^t)}{\prod_{t=1}^{T} q(R^t|R^{t-1}, s_t)} \right] \tag{78}$$

$$= \underbrace{\mathbb{E}_{q(R^{1:T}, s_{1:T}|R^0)} \left[ \sum_{t=1}^{T} \log \frac{p_\theta(s_t|R^t)}{q(s_t)} \right]}_{\text{mask prediction term}} + \mathbb{E}_{q(R^{1:T}, s_{1:T}|R^0)} \left[ \log \frac{p(R^T) \prod_{t=1}^{T} p_\theta(R^{t-1}|R^t)}{\prod_{t=1}^{T} q(R^t|R^{t-1}, s_t)} \right] \tag{79}$$

$$\tag{80}$$

According to Bayes rule, we can rewrite each transition as:

$$q(R^t|R^{t-1}, R^0, s_{1:t}) = \frac{q(R^{t-1}|R^t, R^0, s_{1:t}) q(R^t|R^0, s_{1:t})}{q(R^{t-1}|R^0, s_{1:t-1})}, \tag{81}$$

Armed with this new equation, we drive the second term:

$$\mathbb{E}_{q(R^{1:T}, s_{1:T}|R^0)} \left[ \log \frac{p(R^T) \prod_{t=1}^{T} p_\theta(R^{t-1}|R^t)}{\prod_{t=1}^{T} q(R^t|R^{t-1}, s_t)} \right] \tag{82}$$

$$= \mathbb{E}_{q(R^{1:T}, s_{1:T}|R^0)} \left[ \log \frac{p(R^T) p_\theta(R^0|R^1) \prod_{t=2}^{T} p_\theta(R^{t-1}|R^t)}{q(R^1|R^0, s_1) \prod_{t=2}^{T} q(R^t|R^{t-1}, s_t)} \right] \tag{83}$$

$$= \mathbb{E}_{q(R^{1:T}, s_{1:T}|R^0)} \left[ \log \frac{p(R^T) p_\theta(R^0|R^1) \prod_{t=2}^{T} p_\theta(R^{t-1}|R^t)}{q(R^1|R^0, s_1) \prod_{t=2}^{T} q(R^t|R^{t-1}, R^0, s_{1:t})} \right] \tag{84}$$

$$= \mathbb{E}_{q(R^{1:T}, s_{1:T}|R^0)} \left[ \log \frac{p_\theta(R^T) p_\theta(R^0|R^1)}{q(R^1|R^0, s_1)} + \log \prod_{t=2}^{T} \frac{p_\theta(R^{t-1}|R^t)}{q(R^t|R^{t-1}, R^0, s_{1:t})} \right] \tag{85}$$

$$= \mathbb{E}_{q(R^{1:T}, s_{1:T}|R^0)} \left[ \log \frac{p(R^T) p_\theta(R^0|R^1)}{q(R^1|R^0, s_1)} + \log \prod_{t=2}^{T} \frac{p_\theta(R^{t-1}|R^t)}{\frac{q(R^{t-1}|R^t, R^0, s_{1:t}) q(R^t|R^0, s_{1:t})}{q(R^{t-1}|R^0, s_{1:t-1})}} \right] \tag{86}$$

$$= \mathbb{E}_{q(R^{1:T}, s_{1:t}|R^0)} \left[ \log \frac{p(R^T) p_\theta(R^0|R^1)}{q(R^1|R^0, s_1)} + \log \prod_{t=2}^{T} \frac{p_\theta(R^{t-1}|R^t)}{\frac{q(R^{t-1}|R^t, R^0, s_{1:t}) q(\cancel{R^t|R^0, s_{1:t}})}{q(\cancel{R^{t-1}|R^0, s_{1:t-1}})}} \right] \tag{87}$$

$$= \mathbb{E}_{q(R^{1:T}, s_{1:t}|R^0)} \left[ \log \frac{p(R^T) p_\theta(R^0|R^1)}{\cancel{q(R^1|R^0, s_1)}} + \log \frac{\cancel{q(R^1|R^0, s_1)}}{q(R^T|R^0, s_{1:T})} + \log \prod_{t=2}^{T} \frac{p_\theta(R^{t-1}|R^t)}{q(R^{t-1}|R^t, R^0, s_{1:t})} \right] \tag{88}$$

$$= \mathbb{E}_{q(R^{1:T}, s_{1:t}|R^0)} \left[ \log \frac{p(R^T) p_\theta(R^0|R^1)}{q(R^T|R^0, s_{1:T})} + \sum_{t=2}^{T} \log \frac{p_\theta(R^{t-1}|R^t)}{q(R^{t-1}|R^t, R^0, s_{1:t})} \right] \tag{89}$$

$$= \mathbb{E}_{q(R^{1:T}, s_{1:t}|R^0)} \left[ \log p_\theta(R^0|R^1) \right] \tag{90}$$

$$+ \mathbb{E}_{q(R^{1:T}, s_{1:t}|R^0)} \left[ \log \frac{p(R^T)}{q(R^T|R^0, s_{1:T})} \right] + \sum_{t=2}^{T} \mathbb{E}_{q(R^{1:T}, s_{1:t}|R^0)} \left[ \log \frac{p_\theta(R^{t-1}|R^t)}{q(R^{t-1}|R^t, R^0, s_{1:t})} \right] \tag{91}$$

$$= \mathbb{E}_{q(R^1, s_1|R^0)} \left[ \log p_\theta(R^0|R^1) \right] \tag{92}$$

$$+ \mathbb{E}_{q(R^T|R^0, s_{1:T}) q(s_{1:T})} \left[ \log \frac{p(R^T)}{q(R^T|R^0, s_{1:T})} \right] + \sum_{t=2}^{T} \mathbb{E}_{q(R^t, R^{t-1}, s_{1:t}|R^0)} \left[ \log \frac{p_\theta(R^{t-1}|R^t)}{q(R^{t-1}|R^t, R^0, s_{1:t})} \right] \tag{93}$$

$$= \underbrace{\mathbb{E}_{q(R^1, s_1|R^0)} \left[ \log p_\theta(R^0|R^1) \right]}_{\text{reconstruction term}} \tag{94}$$

$$\underbrace{- \mathbb{E}_{q(s_{1:t})} D_{\text{KL}}(q(R^T|R^0, s_{1:T}) \| p(R^T))}_{\text{prior matching term}} - \sum_{t=2}^{T} \underbrace{\mathbb{E}_{q(R^t, s_{1:t}|R^0)} \left[ D_{\text{KL}}(q(R^{t-1}|R^t, R^0, s_{1:t}) \| p_\theta(R^{t-1}|R^t)) \right]}_{\text{denoising matching term}} \tag{95}$$

Eventually, the ELOB can be rewritten as follows:

$$\log p(R^0) \geq \sum_{t=1}^{T} \underbrace{\mathbb{E}_{q(R^t, s_t | R^0)} \left[ \log \frac{p_\vartheta(s_t | R^t)}{q(s_t)} \right]}_{\text{mask prediction term}} + \underbrace{\mathbb{E}_{q(R^1, s_1 | R^0)} \left[ \log p_{\boldsymbol{\theta}}(R^0 | R^1) \right]}_{\text{reconstruction term}} \tag{96}$$

$$- \underbrace{\mathbb{E}_{q(s_{1:t})} D_{\text{KL}}(q(R^T | R^0, s_{1:T}) \parallel p(R^T))}_{\text{prior matching term}} - \sum_{t=2}^{T} \underbrace{\mathbb{E}_{q(R^t, s_{1:t} | R^0)} \left[ D_{\text{KL}}(q(R^{t-1} | R^t, R^0, s_{1:t}) \parallel p_{\boldsymbol{\theta}}(R^{t-1} | R^t)) \right]}_{\text{denoising matching term}} \tag{97}$$

The mask prediction term can be implemented by a node classifier $s_\vartheta$. For the denoising matching term, by Bayes rule, the $q(R^{t-1} | R^t, R^0, s_{1:t})$ can be written as:

$$q(R^{t-1} | R^t, R^0, s_{1:t}) = \frac{q(R^t | R^{t-1}, R^0, s_{1:t}) q(R^{t-1} | R^0, s_{1:t-1})}{q(R^t | R^0, s_{1:t})}, \tag{98}$$

In maskedDiff, we have

$$q(R^t | R^{t-1}, R^0, s_{1:t}) := \mathcal{N}(R^{t-1}, \sqrt{1 - \beta_t s_t} R^{t-1}, (\beta_t s_t)\mathbf{I}), \tag{99}$$

$$q(R^{t-1} | R^0, s_{1:t-1}) := \mathcal{N}(R^{t-1}, \sqrt{\bar{\gamma}_{t-1}} R^0, (1 - \bar{\gamma}_{t-1})\mathbf{I}). \tag{100}$$

Then the denoising matching term can also be calculated via Lemma 4.1 (let $q(R^t | R^{t-1}, R^0) := q(R^t | R^{t-1}, R^0, s_{1:t})$ and $q(R^{t-1} | R^0) := q(R^{t-1} | R^0, s_{1:t-1})$). Thus, the training objective of MaskedDiff is:

$$\mathcal{L}(\theta, \vartheta) = \mathbb{E}_{t, R^0, \epsilon} \left[ \frac{s_t \beta_t}{2(1 - s_t \beta_t)(1 - \bar{\gamma}_{t-1})} \| \epsilon - \epsilon_\theta(\sqrt{\bar{\gamma}_t} R^0 + \sqrt{(1 - \bar{\gamma}_t)} \epsilon, t, \mathcal{G}) \|^2 + \lambda \text{BCE}(\mathbf{s}_t, s_\vartheta(\mathcal{G}, \mathbf{R}^t, t)) \right] \tag{101}$$

### C.2.1 EXPECTATION OF $s_{1:T}$

The denoising matching term in equation 97 can be calculated by only sampling $(R^t, s_t)$ instead of $(R^t, s_{1:t})$. Specifically, we substitute equation 98 into the denoising matching term:

$$\mathbb{E}_{q(R^t, R^{t-1}, s_{1:t}|R^0)} \left[ \log \frac{p_{\boldsymbol{\theta}}(R^{t-1}|R^t)}{q(R^{t-1}|R^t, R^0, s_{1:t})} \right] \tag{102}$$

$$= \mathbb{E}_{q(R^t, R^{t-1}, s_{1:t}|R^0)} \left[ \log \frac{p_{\boldsymbol{\theta}}(R^{t-1}|R^t)}{\frac{q(R^t|R^{t-1}, R^0, s_{1:t})q(R^{t-1}|R^0, s_{1:t-1})}{q(R^t|R^0, s_{1:t})}} \right] \tag{103}$$

$$= \mathbb{E}_{q(R^t, R^{t-1}, s_{1:t}|R^0)} \left[ \log \frac{p_{\boldsymbol{\theta}}(R^{t-1}|R^t)}{\frac{q(R^t|R^{t-1}, R^0, s_t)}{q(R^t|R^0, s_{1:t})}} - \log q(R^{t-1}|R^0, s_{1:t-1}) \right] \tag{104}$$

$$\geq \mathbb{E}_{q(R^t, R^{t-1}, |R^0, s_{1:t})} \left[ \mathbb{E}_{q(s_{1:t})} \log \frac{p_{\boldsymbol{\theta}}(R^{t-1}|R^t)}{\frac{q(R^t|R^{t-1}, R^0, s_{1:t})}{q(R^t|R^0, s_{1:t})}} \right] \tag{105}$$

$$- \mathbb{E}_{q(s_t)} \left[ \log \underbrace{\mathbb{E}_{q(s_{1:t-1})} q(R^{t-1}|R^0, s_{1:t-1})}_{:= q(\mathbb{E}_s R^{t-1}|R^0)} + \log \underbrace{\mathbb{E}_{q(s_{1:t-1})} q(R^t|R^{t-1}, R^0, s_{1:t})}_{:= q(R^t|\mathbb{E}_s R^{t-1}, R^0, s_t)} \right] \tag{106}$$

$$= \mathbb{E}_{q(R^t, R^{t-1}, |R^0, s_{1:t})} \left[ \mathbb{E}_{q(s_{1:t})} \log \frac{p_{\boldsymbol{\theta}}(R^{t-1}|R^t)}{\frac{q(R^t|R^{t-1}, R^0, s_t)}{q(R^t|R^0, s_{1:t})}} - \mathbb{E}_{q(s_t)} \log q(\mathbb{E}_s R^{t-1}|R^0) - \mathbb{E}_{q(s_t)} \log q(R^t|\mathbb{E}_s R^{t-1}, R^0, s_t) \right] \tag{107}$$

$$= \mathbb{E}_{q(R^t, R^{t-1}, |R^0, s_{1:t})} \left[ \mathbb{E}_{q(s_{1:t})} \log \frac{p_{\boldsymbol{\theta}}(R^{t-1}|R^t)}{\frac{q(R^t|\mathbb{E}_s R^{t-1}, R^0, s_t)q(\mathbb{E}_s R^{t-1}|R^0)}{q(R^t|R^0, s_{1:t})}} \right] \tag{108}$$

$$= \underbrace{\mathbb{E}_{q(R^t, R^{t-1}, s_{1:t}|R^0)} \left[ \log \frac{p_{\boldsymbol{\theta}}(R^{t-1}|R^t)}{\frac{q(R^t|\mathbb{E}_s R^{t-1}, R^0, s_t)q(\mathbb{E}_s R^{t-1}|R^0)}{q(R^t|R^0, s_{1:t})}} \right]}_{\text{denoising matching term}} \tag{109}$$

$$= \mathbb{E}_{q(R^t, R^{t-1}, s_{1:t}|R^0)} \left[ \log \frac{p_{\boldsymbol{\theta}}(R^{t-1}|R^t)}{\hat{q}(R^{t-1}|R^t, R^0, s_{1:t})} \right] \tag{110}$$

$$= \underbrace{\mathbb{E}_{q(R^t, s_{1:t}|R^0)} \left[ D_{\text{KL}}(\hat{q}(R^{t-1}|R^t, R^0, s_{1:t}) \parallel p_{\boldsymbol{\theta}}(R^{t-1}|R^t)) \right]}_{\text{denoising matching term}} \tag{111}$$

Thus, we should focus on calculating the distribution of

$$\hat{q}(R^{t-1}|R^t, R^0, s_{1:t}) := \frac{q(R^t|\mathbb{E}_s R^{t-1}, R^0, s_t)q(\mathbb{E}_s R^{t-1}|R^0)}{q(R^t|R^0, s_{1:t})} \tag{112}$$

By lemma 4.1, if we can gain the expression of $q(R^t|\mathbb{E}_s R^{t-1}, R^0, s_t)$ and $q(\mathbb{E}_s R^{t-1}|R^0)$, we can get the training objective and sampling process.

### C.3 DIFFUSION PROCESS WITH SINGLE STEP MASK.

#### C.3.1 TRAINING

**I: Step $0$ to Step $t-1$ ($R^0 \to R^{t-1}$):** The state space of the mask diffusion should be the mean of the random state.

$$\mathbb{E}_s R^t \sim \mathcal{N}(\mathbb{E}_s R^t; \sqrt{1 - \beta_t} \mathbb{E}_s R^{t-1}, \beta_t I) \tag{113}$$

$$q(R^t|R^0, s_{1:t}) = \mathcal{N}(R^t, \sqrt{\bar{\gamma}_t} R^0, (1 - \bar{\gamma}_t)\mathbf{I}). \tag{114}$$

Form equation 114, we have:

$$R^t = \sqrt{1 - s_t\beta_t}R^{t-1} + \sqrt{s_t\beta_t}\epsilon_{t-1} \tag{115}$$

$$\mathbb{E}R^t = (p\sqrt{1-\beta_t} + 1 - p)\mathbb{E}R^{t-1} + p\sqrt{\beta_t}\epsilon_{t-1} \tag{116}$$

$$= (p\sqrt{1-\beta_t} + 1 - p)(p\sqrt{1-\beta_{t-1}} + 1 - p)\mathbb{E}R^{t-2} + (p\sqrt{1-\beta_t} + 1 - p)p\sqrt{\beta_{t-1}}\epsilon_{t-2} + p\sqrt{\beta_t}\epsilon_{t-1} \tag{117}$$

$$= (p\sqrt{1-\beta_t} + 1 - p)(p\sqrt{1-\beta_{t-1}} + 1 - p)\mathbb{E}R^{t-2} + \sqrt{[(p\sqrt{1-\beta_t} + 1 - p)p\sqrt{\beta_{t-1}}]^2 + [p\sqrt{\beta_t}]^2}\epsilon_{t-2} \tag{118}$$

$$= .... \tag{119}$$

$$= \prod_{i=1}^{t}(p\sqrt{1-\beta_i} + 1 - p)R^0 + \sqrt{[\prod_{j=2}^{t}(p\sqrt{1-\beta_j} + 1 - p)p\sqrt{\beta_1}]^2 + [\prod_{j=3}^{t}(p\sqrt{1-\beta_j} + 1 - p)p\sqrt{\beta_2}]^2 + ... + \epsilon_0} \tag{120}$$

$$= \prod_{i=1}^{t}(p\sqrt{1-\beta_i} + 1 - p)R^0 + \sqrt{\sum_{i=1}^{t}[\prod_{j=i+1}^{t}(p\sqrt{1-\beta_j} + 1 - p)p\sqrt{\beta_i}]^2} \tag{121}$$

$$= \prod_{i=1}^{t}\sqrt{\alpha_i}R^0 + \sqrt{\sum_{i=1}^{t}[\prod_{j=i+1}^{t}\sqrt{\alpha_i}p\sqrt{\beta_i}]^2\epsilon_0} \tag{122}$$

$$= \prod_{i=1}^{t}\sqrt{\alpha_i}R^0 + p\sqrt{\sum_{i=1}^{t}\prod_{j=i+1}^{t}\alpha_j\beta_i\epsilon_0} \tag{123}$$

$$= \sqrt{\bar{\alpha}_t}R^0 + p\sqrt{\sum_{i=1}^{t}\frac{\bar{\alpha}_t}{\bar{\alpha}_i}\beta_i\epsilon_0} \tag{124}$$

$$\tag{125}$$

where $\alpha_i := (p\sqrt{1-\beta_i} + 1 - p)^2$ and $\bar{\alpha}_t = \prod_{i=1}^{t}\alpha_i$.

$$q(\mathbb{E}R^t|R^0) = \mathcal{N}(R^t; \sqrt{\bar{\alpha}_t}R^0, p^2\sum_{i=1}^{t}\frac{\bar{\alpha}_t}{\bar{\alpha}_i}\beta_i I) \tag{126}$$

**II: Step $t-1$ to Step $t$ ($R^{t-1} \to R^t$):** We build the step $t-1 \to t$ is a discrete transition from $q(\mathbf{R}^{t-1}|\mathbf{R}^0)$, with

$$q(\mathbb{E}_s R^{t-1}|R^0) = \mathcal{N}(R^{t-1}; \prod_{i=1}^{t-1}\sqrt{\alpha_i}R^0, p^2\sum_{i=1}^{t-1}\prod_{j=i+1}^{t-1}\alpha_j\beta_i I) \tag{127}$$

$$q(R^t|\mathbb{E}_s R^{t-1}, s_t) = \mathcal{N}(R^t; \sqrt{1 - s_t\beta_t}\mathbb{E}R^{t-1}, s_t\beta_t I) \tag{128}$$

$$R^t = \sqrt{1 - s_t\beta_t}\mathbb{E}R^{t-1} + \sqrt{s_t\beta_t}\epsilon_{t-1} \tag{129}$$

$$= \sqrt{1 - s_t\beta_t}\left(\sqrt{\bar{\alpha}_{t-1}}R^0 + p\sqrt{\sum_{i=1}^{t-1}\frac{\bar{\alpha}_{t-1}}{\bar{\alpha}_i}\beta_i\epsilon_0}\right) + \sqrt{s_t\beta_t}\epsilon_{t-1} \tag{130}$$

$$= \sqrt{1 - s_t\beta_t}\sqrt{\bar{\alpha}_{t-1}}R^0 + p\sqrt{1 - s_t\beta_t}\sqrt{\sum_{i=1}^{t-1}\frac{\bar{\alpha}_{t-1}}{\bar{\alpha}_i}\beta_i\epsilon_0} + \sqrt{s_t\beta_t}\epsilon_{t-1} \tag{131}$$

$$= \sqrt{1 - s_t\beta_t}\sqrt{\bar{\alpha}_{t-1}}R^0 + \sqrt{p^2(1 - s_t\beta_t)\sum_{i=1}^{t-1}\frac{\bar{\alpha}_{t-1}}{\bar{\alpha}_i}\beta_i + s_t\beta_t\epsilon_0} \tag{132}$$

**Step** $0$ **to Step** $t$ $(R^0 \to R^t)$**:**

$$q(R^t|R^0) = \int q(R^t|\mathbb{E}R^{t-1})q(\mathbb{E}R^{t-1}|R^0)d\mathbb{E}R^{t-1} \tag{133}$$

$$= \mathcal{N}(R^t; \sqrt{1 - s_t\beta_t}\sqrt{\bar{\alpha}_i}R^0, (p^2(1 - s_t\beta_t)\sum_{i=1}^{t-1}\frac{\bar{\alpha}_{t-1}}{\bar{\alpha}_i}\beta_i + s_t\beta_t)I) \tag{134}$$

Thus, from subsection C.2.1, the **training objective** of 1-step SUBGDIFF is:

$$\mathcal{L}_{simple}(\theta, \vartheta) = \mathbb{E}_{t,R^0,s_t,\epsilon}[s_t\|\epsilon - \epsilon_\theta(R^t, t)\|^2 - \mathcal{BCE}(s_t, s_\vartheta(R^t, t))] \tag{135}$$

where $\mathcal{BCE}(s_t, s_\vartheta) = s_t \log s_\vartheta(R^t, t) + (1 - s_t)\log(1 - s_\vartheta(R^t, t))$ is Binary Cross Entropy loss. However, training the MaskedDiff is not trivial. The challenges come from two aspects: 1) the mask predictor should be capable of perceiving the sensible noise change between $(t-1)$-th and $t$-th step. However, the noise scale $\beta_t$ is relatively small when $t$ is small, especially if the diffusion step is larger than a thousand, thereby mask predictor cannot precisely predict. 2) The accumulated noise for each node at $(t-1)$-th step would be mainly affected by the mask sampling from 1 to $t-1$ step, which heavily increases the difficulty of predicting the noise added between $(t-1)$-step to $t$-step.

### C.3.2 SAMPLING

Finally, the sampling can be written as:

$$R^{t-1} = \frac{\left((1 - s_t\beta_t)p^2\sum_{i=1}^{t-1}\frac{\bar{\alpha}_{t-1}}{\bar{\alpha}_i}\beta_i + s_t\beta_t\right)R^t - \left(s_t\beta_t\sqrt{p^2(1 - s_t\beta_t)\sum_{i=1}^{t-1}\frac{\bar{\alpha}_{t-1}}{\bar{\alpha}_i}\beta_i + s_t\beta_t}\right)\epsilon_\theta(R^t, t)}{\sqrt{1 - s_t\beta_t}(s_t\beta_t + (1 - s_t\beta_t)p^2\sum_{i=1}^{t-1}\frac{\bar{\alpha}_{t-1}}{\bar{\alpha}_i}\beta_i)} + \sigma_t z \tag{136}$$

$$= \frac{1}{\sqrt{1 - s_t\beta_t}}R^t - \frac{\left(s_t\beta_t\sqrt{p^2(1 - s_t\beta_t)\sum_{i=1}^{t-1}\frac{\bar{\alpha}_{t-1}}{\bar{\alpha}_i}\beta_i + s_t\beta_t}\right)}{\sqrt{1 - s_t\beta_t}(s_t\beta_t + (1 - s_t\beta_t)p^2\sum_{i=1}^{t-1}\frac{\bar{\alpha}_{t-1}}{\bar{\alpha}_i}\beta_i)}\epsilon_\theta(R^t, t) + \sigma_t z \tag{137}$$

$$= \frac{1}{\sqrt{1 - s_t\beta_t}}R^t - \frac{s_t\beta_t}{\sqrt{1 - s_t\beta_t}\sqrt{s_t\beta_t + (1 - s_t\beta_t)p^2\sum_{i=1}^{t-1}\frac{\bar{\alpha}_{t-1}}{\bar{\alpha}_i}\beta_i}}\epsilon_\theta(R^t, t) + \sigma_t z \tag{138}$$

$$\tag{139}$$

where $s_t = s_\vartheta(R^t, t)$ and

$$\sigma_t = \frac{s_\vartheta(R^t, t)\beta_t p^2\sum_{i=1}^{t-1}\frac{\bar{\alpha}_{t-1}}{\bar{\alpha}_i}\beta_i}{s_\vartheta(R^t, t)\beta_t + p^2(1 - s_\vartheta(R^t, t)\beta_t)\sum_{i=1}^{t-1}\frac{\bar{\alpha}_{t-1}}{\bar{\alpha}_i}\beta_i} \tag{140}$$

Compare with Eq. 4, this sampling process is

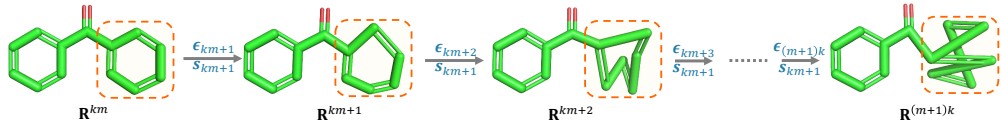

Figure 5: An example of $k$-step same mask diffusion, where the mask vectors are same as $\mathbf{s}_{km+1}$ from step $km$ to $(m+1)k$, $m \in \mathbb{N}^+$ .

## D  MEAN STATE DISTRIBUTION

The state space of the mask diffusion should be the mean of the random state.

$$\mathbb{E}_{s_t} R^t \sim \mathcal{N}(\mathbb{E}R^t; \sqrt{1 - \beta_t}\mathbb{E}_{s_{t-1}}R^{t-1}, \beta_t I) \tag{141}$$

Form equation 114, we have:

$$R^t = \sqrt{1 - s_t\beta_t}R^{t-1} + \sqrt{s_t\beta_t}\epsilon_{t-1} \tag{142}$$

$$\mathbb{E}R^t = (p\sqrt{1 - \beta_t} + 1 - p)\mathbb{E}R^{t-1} + p\sqrt{\beta_t}\epsilon_{t-1} \tag{143}$$

$$= (p\sqrt{1 - \beta_t} + 1 - p)(p\sqrt{1 - \beta_{t-1}} + 1 - p)\mathbb{E}R^{t-2} \tag{144}$$

$$+ (p\sqrt{1 - \beta_t} + 1 - p)p\sqrt{\beta_{t-1}}\epsilon_{t-2} + p\sqrt{\beta_t}\epsilon_{t-1} \tag{145}$$

$$= (p\sqrt{1 - \beta_t} + 1 - p)(p\sqrt{1 - \beta_{t-1}} + 1 - p)\mathbb{E}R^{t-2} \tag{146}$$

$$+ \sqrt{[(p\sqrt{1 - \beta_t} + 1 - p)p\sqrt{\beta_{t-1}}]^2 + [p\sqrt{\beta_t}]^2}\epsilon_{t-2} \tag{147}$$

$$= .... \tag{148}$$

$$= \prod_{i=1}^{t}(p\sqrt{1 - \beta_i} + 1 - p)R^0 \tag{149}$$

$$+ \sqrt{[\prod_{j=2}^{t}(p\sqrt{1 - \beta_j} + 1 - p)p\sqrt{\beta_1}]^2 + [\prod_{j=3}^{t}(p\sqrt{1 - \beta_j} + 1 - p)p\sqrt{\beta_2}]^2 + ... + \epsilon_0} \tag{150}$$

$$= \prod_{i=1}^{t}(p\sqrt{1 - \beta_i} + 1 - p)R^0 + \sqrt{\sum_{i=1}^{t}[\prod_{j=i+1}^{t}(p\sqrt{1 - \beta_j} + 1 - p)p\sqrt{\beta_i}]^2} \tag{151}$$

$$= \prod_{i=1}^{t}\sqrt{\alpha_i}R^0 + \sqrt{\sum_{i=1}^{t}[\prod_{j=i+1}^{t}\sqrt{\alpha_i}p\sqrt{\beta_i}]^2}\epsilon_0 \tag{152}$$

$$= \prod_{i=1}^{t}\sqrt{\alpha_i}R^0 + p\sqrt{\sum_{i=1}^{t}\prod_{j=i+1}^{t}\alpha_j\beta_i}\epsilon_0 \tag{153}$$

$$= \sqrt{\bar{\alpha_t}}R^0 + p\sqrt{\sum_{i=1}^{t}\frac{\bar{\alpha}_t}{\bar{\alpha}_i}\beta_i}\epsilon_0 \tag{154}$$

$$\tag{155}$$

where $\alpha_i := (p\sqrt{1 - \beta_i} + 1 - p)^2$ and $\bar{\alpha}_t = \prod_{i=1}^{t}\alpha_i$.

Finally, the Mean state distribution is:

$$q(\mathbb{E}R^t|R^0) = \mathcal{N}(\mathbb{E}R^t; \prod_{i=1}^{t}\sqrt{\alpha_i}R^0, p^2\sum_{i=1}^{t}\prod_{j=i+1}^{t}\alpha_j\beta_i I) \tag{156}$$

## E  THE DERIVATION OF SUBGDIFF

When $t$ is an integer multiple of $k$,

$$\mathbb{E}R^t = \prod_{j=1}^{t/k}(p\sqrt{\prod_{i=(j-1)k+1}^{kj}(1-\beta_i)}+1-p)R^0 \tag{157}$$

$$+ \sqrt{\sum_{l=1}^{t/k}\left[\prod_{j=l+1}^{t/k}(p\sqrt{\prod_{i=(j-1)k+1}^{kj}(1-\beta_i)}+1-p)p\sqrt{1-\prod_{i=(l-1)k+1}^{kl}(1-\beta_i)}\right]^2}\epsilon_0 \tag{158}$$

$$= \prod_{j=1}^{t/k}\sqrt{\alpha_j}R^0 + p\sqrt{\sum_{l=1}^{t/k}\prod_{j=l+1}^{t/k}\alpha_j(1-\prod_{i=(l-1)k+1}^{kl}(1-\beta_i))}\epsilon_0 \tag{159}$$

$$= \sqrt{\bar{\alpha}_{t/k}}R^0 + p\sqrt{\sum_{l=1}^{t/k}\frac{\bar{\alpha}_{t/k}}{\bar{\alpha}_l}(1-\prod_{i=(l-1)k+1}^{kl}(1-\beta_i))}\epsilon_0 \tag{160}$$

where $\alpha_j = (p\sqrt{\prod_{i=(j-1)k+1}^{kj}(1-\beta_i)}+1-p)^2$.

When $t \in \mathbb{N}$, we have

$$R^t = \sqrt{\prod_{i=k\lfloor t/k\rfloor+1}^{t}(1-\beta_i s_{\lfloor t/k\rfloor})}\mathbb{E}R^{\lfloor t/k\rfloor\times k} + \sqrt{1-\prod_{i=k\lfloor t/k\rfloor+1}^{t}(1-\beta_i s_{\lfloor t/k\rfloor})}\epsilon_{\lfloor t/k\rfloor\times k} \tag{161}$$

$$= \sqrt{\prod_{i=k\lfloor t/k\rfloor+1}^{t}(1-\beta_i s_{\lfloor t/k\rfloor})}\left(\sqrt{\bar{\alpha}_{\lfloor t/k\rfloor}}R^0 + p\sqrt{\sum_{l=1}^{\lfloor t/k\rfloor}\frac{\bar{\alpha}_{\lfloor t/k\rfloor}}{\bar{\alpha}_l}(1-\prod_{i=(l-1)k+1}^{kl}(1-\beta_i))}\epsilon_0\right) \tag{162}$$

$$+ \sqrt{1-\prod_{t=\lfloor t/k\rfloor}^{t}(1-\beta_i s_{\lfloor t/k\rfloor})}\epsilon_{\lfloor t/k\rfloor} \tag{163}$$

$$= \sqrt{\prod_{i=k\lfloor t/k\rfloor+1}^{t}\gamma_i}\sqrt{\bar{\alpha}_{\lfloor t/k\rfloor}}R^0 \tag{164}$$

$$+ \sqrt{\left(\prod_{i=k\lfloor t/k\rfloor+1}^{t}\gamma_i\right)p^2\sum_{l=1}^{\lfloor t/k\rfloor}\frac{\bar{\alpha}_{\lfloor t/k\rfloor}}{\bar{\alpha}_l}(1-\prod_{i=(l-1)k+1}^{kl}(1-\beta_i))+\left(1-\prod_{i=k\lfloor t/k\rfloor+1}^{t}\gamma_i\right)}\epsilon_0 \tag{165}$$

where $\gamma_i = 1-\beta_i s_{\lfloor t/k\rfloor}$.

$$q(R^t|R^0) = \mathcal{N}(R^{k\lfloor t/k\rfloor};\sqrt{\prod_{i=k\lfloor t/k\rfloor+1}^{t}\gamma_i}\sqrt{\bar{\alpha}_{\lfloor t/k\rfloor}}R^0, \tag{166}$$

$$\left(\left(\prod_{i=k\lfloor t/k\rfloor+1}^{t}\gamma_i\right)p^2\sum_{l=1}^{\lfloor t/k\rfloor}\frac{\bar{\alpha}_{\lfloor t/k\rfloor}}{\bar{\alpha}_l}(1-\prod_{i=(l-1)k+1}^{kl}(1-\beta_i))+1-\prod_{i=k\lfloor t/k\rfloor+1}^{t}\gamma_i\right)I) \tag{167}$$

Let $m = \lfloor t/k \rfloor$, $\bar{\gamma}_i = \prod_{t=1}^{i} \gamma_t$, and $\bar{\beta}_t = \prod_{i=1}^{t}(1 - \beta_i)$

$$q(R^t | R^0) = \mathcal{N}(R^{km}; \sqrt{\frac{\bar{\gamma}_t}{\bar{\gamma}_{km}}} \sqrt{\bar{\alpha}_m} R^0, \left( \frac{\bar{\gamma}_t}{\bar{\gamma}_{km}} p^2 \sum_{l=1}^{m} \frac{\bar{\alpha}_m}{\bar{\alpha}_l}(1 - \frac{\bar{\beta}_{kl}}{\bar{\beta}_{(l-1)k}}) + 1 - \frac{\bar{\gamma}_t}{\bar{\gamma}_{km}} \right) I) \quad (168)$$

### E.0.1 SAMPLING

$$\mu_1 = \sqrt{1 - s_{km+1}\beta_t}, \tag{169}$$

$$\sigma_1^2 = s_{km+1}\beta_t \tag{170}$$

$$\mu_2 = \sqrt{\frac{\bar{\gamma}_{t-1}}{\bar{\gamma}_{km}}} \sqrt{\bar{\alpha}_m} \tag{171}$$

$$\sigma_2^2 = \frac{\bar{\gamma}_{t-1}}{\bar{\gamma}_{km}} p^2 \sum_{l=1}^{m} \frac{\bar{\alpha}_m}{\bar{\alpha}_l}(1 - \prod_{i=(l-1)k+1}^{kl}(1 - \beta_i)) + 1 - \frac{\bar{\gamma}_{t-1}}{\bar{\gamma}_{km}} \tag{172}$$

According to the Lemma 4.1, we have

$$R^{t-1} = \frac{1}{\mu_1} \left( R^t - \frac{\sigma_1^2}{\sqrt{\mu_1^2 \sigma_2^2 + \sigma_1^2}} \epsilon_\theta(R^t, t) \right) + \frac{\sigma_1 \sigma_2}{\sqrt{\mu_1^2 \sigma_2^2 + \sigma_1^2}} z \tag{173}$$

$$= \frac{1}{\sqrt{1 - s_{km+1}\beta_t}}(R^t - \tag{174}$$

$$\frac{s_{km+1}\beta_t}{\sqrt{(1 - s_{km+1}\beta_t)(\frac{\bar{\gamma}_{t-1}}{\bar{\gamma}_{km}} p^2 \sum_{l=1}^{m} \frac{\bar{\alpha}_m}{\bar{\alpha}_l}(1 - \prod_{i=(l-1)k+1}^{kl}(1 - \beta_i)) + 1 - \frac{\bar{\gamma}_{t-1}}{\bar{\gamma}_{km}}) + s_{km+1}\beta_t}} \epsilon_\theta(R^t, t))$$
$$\tag{175}$$

$$+ \frac{\sqrt{s_{km+1}\beta_t} \sqrt{\frac{\bar{\gamma}_{t-1}}{\bar{\gamma}_{km}} p^2 \sum_{l=1}^{m} \frac{\bar{\alpha}_m}{\bar{\alpha}_l}(1 - \prod_{i=(l-1)k+1}^{kl}(1 - \beta_i)) + 1 - \frac{\bar{\gamma}_{t-1}}{\bar{\gamma}_{km}}}}{\sqrt{(1 - s_{km+1}\beta_t)(\frac{\bar{\gamma}_{t-1}}{\bar{\gamma}_{km}} p^2 \sum_{l=1}^{m} \frac{\bar{\alpha}_m}{\bar{\alpha}_l}(1 - \prod_{i=(l-1)k+1}^{kl}(1 - \beta_i)) + 1 - \frac{\bar{\gamma}_{t-1}}{\bar{\gamma}_{km}}) + s_{km+1}\beta_t}} z$$
$$\tag{176}$$

**Algorithm 3:** Training SUBGDIFF

---

**Input:** A molecular graph $G_{3D}$, $k$ for same mask diffusion, the

Sample $t \sim \mathcal{U}(1, ..., T)$ , $\epsilon \sim \mathcal{N}(\mathbf{0}, \mathbf{I})$

Sample $\mathbf{s}^t \in p_{s_t}(\mathcal{S})$       ▷ Sample a masked vector (subgraph node-set)

$\mathbf{R}^t \leftarrow q(\mathbf{R}^t | \mathbf{R}^0)$       ▷ equation 14

$\mathcal{L}_1 = \text{BCE}(\mathbf{s}_t, s_\vartheta(\mathcal{G}, \mathbf{R}^t, t))$       ▷ Mask prediction loss

$\mathcal{L}_2 = \|\text{diag}(\mathbf{s}_t)(\epsilon - \epsilon_\theta(\mathcal{G}, \mathbf{R}^t, t))\|^2$       ▷ Denoising loss

optimizer. step$(\lambda \mathcal{L}_1 + \mathcal{L}_2)$       ▷ Optimize parameters $\theta, \vartheta$

---

**Algorithm 4:** Sampling from SUBGDIFF

---

Sample $\mathbf{R}^T \sim \mathcal{N}(\mathbf{0}, \mathbf{I})$       ▷ Random noise initialization

**for** $t$ = $T$ **to** $1$ **do**

     $\mathbf{z} \sim \mathcal{N}(\mathbf{0}, \mathbf{I})$ if $t > 1$, else $\mathbf{z} = \mathbf{0}$       ▷ Random noise

     **If** $t\%k == 0$ or $t == T$: $\hat{\mathbf{s}} \leftarrow s_\vartheta(\mathcal{G}, \mathbf{R}^t, t)$       ▷ Mask vecter prediction

     $\hat{\epsilon} \leftarrow \epsilon_\theta(\mathcal{G}, \mathbf{R}^t, t)$       ▷ Posterior

     $\mathbf{R}^{t-1} \leftarrow$ equation 16       ▷ sampling

**end**

**return** $\mathbf{R}^0$

---

## F  Additional Experiments

Table 5: Additional hyperparameters of our SUBGDIFF.

| Task | $\beta_1$ | $\beta_T$ | $\beta$ scheduler | $T$ | k (k-same mask) | $\tau$ | Batch Size | Train Iter. |
|---|---|---|---|---|---|---|---|---|
| QM9 | 1e-7 | 2e-3 | sigmoid | 5000 | 250 | 10Å | 64 | 2M |
| Drugs | 1e-7 | 2e-3 | sigmoid | 5000 | 250 | 10Å | 32 | 6M |

Table 6: Additional hyperparameters of our SUBGDIFF with different timesteps.

| Task | $\beta_1$ | $\beta_T$ | $\beta$ scheduler | $T$ | k (k-same mask) | $\tau$ | Batch Size | Train Iter. |
|---|---|---|---|---|---|---|---|---|
| 500-step QM9 | 1e-7 | 2e-2 | sigmoid | 500 | 25 | 10Å | 64 | 2M |
| 200-step QM9 | 1e-7 | 5e-2 | sigmoid | 200 | 10 | 10Å | 64 | 2M |
| 500-step Drugs | 1e-7 | 2e-2 | sigmoid | 500 | 25 | 10Å | 32 | 4M |
| 1000-step Drugs | 1e-7 | 9e-3 | sigmoid | 500 | 50 | 10Å | 32 | 4M |

### F.1  Details of settings.

All models are trained with SGD using the ADAM optimizer.

#### F.1.1  Conformation Generation

**Evaluation metrics for conformation generation.** To compare the generated and ground truth conformer ensembles, we employ the same evaluation metrics as in a prior study (Ganea et al., 2021): Average Minimum RMSD (AMR) and Coverage. These metrics enable us to assess the quality of the generated conformers from two perspectives: Recall (R) and Precision (P). Recall measures the extent to which the generated ensemble covers the ground-truth ensemble, while Precision evaluates the accuracy of the generated conformers.

The four metrics built upon root-mean-square deviation (RMSD), which is defined as the normalized Frobenius norm of two atomic coordinates matrices, after alignment by Kabsch algorithm (Kabsch, 1976). Formally, let $S_g$ and $S_r$ denote the sets of generated and reference conformers respectively, then the **Cov**erage and **Mat**ching metrics (Xu et al., 2021a) can be defined as:

$$\text{COV-R}(S_g, S_r) = \frac{1}{|S_r|}\left|\left\{\mathcal{C} \in S_r|\, \text{RMSD}(\mathcal{C}, \hat{\mathcal{C}}) \leq \delta, \hat{\mathcal{C}} \in S_g \right\}\right|, \tag{177}$$

$$\text{MAT-R}(S_g, S_r) = \frac{1}{|S_r|} \sum_{\mathcal{C} \in S_r} \min_{\hat{\mathcal{C}} \in S_g} \text{RMSD}(\mathcal{C}, \hat{\mathcal{C}}), \tag{178}$$

where $\delta$ is a threshold. The other two metrics COV-P and MAT-P can be defined similarly but with the generated sets $S_g$ and reference sets $S_r$ exchanged. In practice, $S_g$ is set as twice of the size of $S_r$ for each molecule.

**Settings**. For GEODIFF (Xu et al., 2022) with 5000 steps, we use the checkpoints released in public GitHub to reproduce the results. For 200 and 500 steps, we retrain it and do the DDPM sampling.

**Comparison with GEODIFF using Langevin Dynamics sampling method.** In order to verify that our proposed diffusion process can bring benefits to other sampling methods, we conduct the experiments to compare our proposed diffusion model with GEODIFF by adopting a typical sampling method Langevin dynamics (LD sampling)(Song & Ermon, 2019) :

$$\mathbf{R}^{t-1} = \mathbf{R}^t + \alpha_t \epsilon_\theta(\mathcal{G}, \mathbf{R}^t, t) + \sqrt{2\alpha_t}\mathbf{z}_{t-1} \tag{179}$$

where $\mathbf{z}_t \sim \mathcal{N}(\mathbf{0}, \mathbf{I})$ and $h\sigma_t^2$. $h$ is the hyper-parameter referring to step size and $\sigma_t$ is the noise schedule in the forward process. We use various time-step to evaluate the generalization and robustness of the proposed method, and the results shown in Table 7 indicate that our method significantly outperforms GEODIFF, especially when the time-step is relatively small (200,500), which implies that our training method can effectively improve the efficiency of denoising.

Table 7: Results on **GEOM-QM9** dataset with different time steps. Langevin dynamics (Song & Ermon, 2019) is a typical sampling method used in DPM. ▲denotes SUBGDIFF outperforms GEODIFF. The threshold $\delta = 0.5$Å.

| Steps | Sampling method | Models | COV-R (%) ↑ Mean | Median | MAT-R (Å) ↓ Mean | Median | COV-P (%) ↑ Mean | Median | MAT-P (Å) ↓ Mean | Median |
|-------|-----------------|--------|------|--------|------|--------|------|--------|------|--------|
| 5000 | Langevin dynamics | GEODIFF | 88.35 | 92.55 | 0.2166 | 0.2154 | 52.67 | 50.00 | 0.4398 | 0.4264 |
| 5000 | Langevin dynamics | SUBGDIFF | 88.76▲ | 94.23▲ | 0.2343↓ | 0.2244↓ | 50.28↓ | 49.62↓ | 0.4728↓ | 0.4549↓ |
| 500 | Langevin dynamics | GEODIFF | 87.80 | 93.66 | 0.3179 | 0.3216 | 46.25 | 45.02 | 0.6173 | 0.5112 |
| 500 | Langevin dynamics | SUBGDIFF | 91.40▲ | 95.39▲ | 0.2543▲ | 0.2601▲ | 51.71▲ | 48.50▲ | 0.5035▲ | 0.4734▲ |
| 200 | Langevin dynamics | GEODIFF | 86.60 | 93.09 | 0.3532 | 0.3574 | 42.98 | 42.60 | 0.5563 | 0.5367 |
| 200 | Langevin dynamics | SUBGDIFF | 90.36▲ | 95.93▲ | 0.3064▲ | 0.3098▲ | 48.56▲ | 46.46▲ | 0.5540▲ | 0.5082▲ |

Table 8: Results on **GEOM-QM9** dataset. The threshold $\delta = 0.5$Å.

| Models | COV-R (%) ↑ Mean | Median | MAT-R (Å) ↓ Mean | Median | COV-P (%) ↑ Mean | Median | MAT-P (Å) ↓ Mean | Median |
|--------|------|--------|------|--------|------|--------|------|--------|
| CVGAE | 0.09 | 0.00 | 1.6713 | 1.6088 | - | - | - | - |
| GRAPHDG | 73.33 | 84.21 | 0.4245 | 0.3973 | 43.90 | 35.33 | 0.5809 | 0.5823 |
| CGCF | 78.05 | 82.48 | 0.4219 | 0.3900 | 36.49 | 33.57 | 0.6615 | 0.6427 |
| CONFVAE | 77.84 | 88.20 | 0.4154 | 0.3739 | 38.02 | 34.67 | 0.6215 | 0.6091 |
| GEOMOL | 71.26 | 72.00 | 0.3731 | 0.3731 | - | - | - | - |
| CONFGF | 88.49 | 94.31 | 0.2673 | 0.2685 | 46.43 | 43.41 | **0.5224** | 0.5124 |
| GEODIFF | 80.36 | 83.82 | 0.2820 | 0.2799 | **53.66** | **50.85** | 0.6673 | **0.4214** |
| **SUBGDIFF** | **90.91** | **95.59** | **0.2460** | **0.2351** | 50.16 | 48.01 | 0.6114 | 0.4791 |

**Comparison with SOTAs.** **i) Baselines:** We compare SUBGDIFF with 7 state-of-the-art baselines: CVGAE (Mansimov et al., 2019), GRAPHDG (Simm & Hernandez-Lobato, 2020), CGCF (Xu et al., 2021a), CONFVAE (Xu et al., 2021b), CONFGF (Shi et al., 2021) and GEODIFF (Xu et al., 2022). For the above baselines, we reuse the experimental results reported by Xu et al. (2022). For GEODIFF (Xu et al., 2022), we use the checkpoints released in public GitHub to reproduce the results. **ii)Results:** The results on the GEOM-QM9 dataset are reported in Table 8. From the results, we get the following observation: SUBGDIFF significantly outperforms the baselines on COV-R, indicating the SUBGDIFF tends to explore more possible conformations. This implicitly demonstrates the subgraph will help fine-tune the generated conformation to be a potential conformation.

Table 9: Results on the **GEOM-Drugs** dataset under different diffusion timesteps. DDPM (Ho et al., 2020) is the sampling method used in GeoDiff and Langevin dynamics (Song & Ermon, 2019) is a typical sampling method used in DPM. Our proposed sampling method (Algorithm 2) can be viewed as a DDPM variant. ▲/▼ denotes SUBGDIFF outperforms/underperforms GEODIFF. The threshold $\delta = 1.25$Å.

| Models | Timesteps | Sampling method | COV-R (%) ↑ | | MAT-R (Å) ↓ | |
|---|---|---|---|---|---|---|
| | | | Mean | Median | Mean | Median |
| GEODIFF | 500 | DDPM | 50.25 | 48.18 | 1.3101 | 1.2967 |
| SUBGDIFF | 500 | DDPM (ours) | 76.16▲ | 86.43▲ | 1.0463▲ | 1.0264▲ |
| GEODIFF | 500 | LD | 64.12 | 75.56 | 1.1444 | 1.1246 |
| SUBGDIFF | 500 | LD (ours) | 74.30▲ | 77.87▲ | 1.0003▲ | 0.9905▲ |

Table 10: Results on the **GEOM-Drugs** dataset. The threshold $\delta = 1.25 \text{Å}$

| Models | Train data | COV-R (%) ↑ Mean | COV-R (%) ↑ Median | MAT-R (Å) ↓ Mean | MAT-R (Å) ↓ Median |
|---|---|---|---|---|---|
| CVGAE | Drugs | 0.00 | 0.00 | 3.0702 | 2.9937 |
| GRAPHDG | Drugs | 8.27 | 0.00 | 1.9722 | 1.9845 |
| GEODIFF | QM9 | 7.99 | 0.00 | 2.7704 | 2.3297 |
| **SUBGDIFF** | QM9 | **24.01** | **9.93** | **1.6128** | **1.5819** |

**Model Architecture.** We adopt the graph field network (GFN) from Xu et al. (2022) as the GNN encoder for extracting the 3D molecular information. In the $l$-th layer, the GFN receives node embeddings $\mathbf{h}^l \in \mathbb{R}^{n \times b}$ (where $b$ represents the feature dimension) and corresponding coordinate embeddings $\mathbf{x}^l \in \mathbb{R}^{n \times 3}$ as input. It then produces the output $\mathbf{h}^{l+1}$ and $\mathbf{x}^{l+1}$ according to the following process:

$$\mathbf{m}_{ij}^l = \Phi_m^l \left( \mathbf{h}_i^l, \mathbf{h}_j^l, \|\mathbf{x}_i^l - \mathbf{x}_j^l\|^2, e_{ij}; \theta_m \right) \tag{180}$$

$$\mathbf{h}_i^{l+1} = \Phi_h^l \left( \mathbf{h}_i^l, \sum_{j \in \mathcal{N}(i)} \mathbf{m}_{ij}^l; \theta_h \right) \tag{181}$$

$$\mathbf{x}_i^{l+1} = \sum_{j \in \mathcal{N}(i)} \frac{1}{d_{ij}} (R_i - R_j) \Phi_x^l \left( \mathbf{m}_{ij}^l; \theta_x \right) \tag{182}$$

where $\Phi$ are implemented as feed-forward networks and $d_{ij}$ denotes interatomic distances. The initial embedding $\mathbf{h}^0$ is composed of atom embedding and time step embedding while $\mathbf{x}^0$ represents atomic coordinates. $\mathcal{N}(i)$ is the neighborhood of $i^{th}$ node, consisting of connected atoms and other ones within a radius threshold $\tau$, helping the model capture long-range interactions explicitly and support disconnected molecular graphs.

Eventually, the Gaussian noise and mask can be predicted as follows (C.f. Figure 6):

$$\hat{\epsilon}_i = \mathbf{x}_i^L \tag{183}$$

$$\hat{s}_i = \text{MLP}(\mathbf{h}_i^L) \tag{184}$$

where $\hat{\epsilon}_i$ is equivalent and $\hat{s}_i$ is invariant.

## F.2 DOMAIN GENERELIZAION

The results of Training on QM9 (small molecular with up to 9 heavy atoms) and testing on Drugs (medium-sized organic compounds) can be found in table 10.

## F.3 SELF-SUPERVISED LEARNING

### F.3.1 MODEL ARCHITECTURE

We use the pretraining framework MoleculeSDE proposed by Liu et al. (2023a) and extend our SUBGDIFF to multi-modality pertaining. The two key components of MoleculeSDE are two SDEs(stochastic differential equations Song et al. (2020)): an SDE from 2D topology to 3D conformation (2D → 3D) and an SDE from 3D conformation to 2D topology (3D → 2D). In practice, these two SDEs can be replaced by discrete diffusion models. In this paper, we use the proposed SUBGDIFF to replace the SDEs.

**2D topological molecular graph.** A topological molecular graph is denoted as $g_{2D} = \mathcal{G}(\mathcal{V}, \mathbf{E}, \mathbf{X})$, where $\mathbf{X}$ is the atom attribute matrix and $\mathbf{X}$ is the bond attribute matrix. The 2D graph representation with graph neural network (GNN) is:

$$\boldsymbol{x} \triangleq \boldsymbol{H}_{2D} = \text{GIN}(g_{2D}) = \text{GIN}(\mathbf{X}, \mathbf{X}), \tag{185}$$

where GIN is the a powerful 2D graph neural network (Xu et al., 2018) and $\boldsymbol{H}_{2D} = [h_{2D}^0, h_{2D}^1, \ldots]$, where $h_{2D}^i$ is the $i$-th node representation.

**3D conformational molecular graph.** The molecular conformation is denoted as $g_{3D} := G_{3D}(\mathcal{G}, \mathbf{R})$. The conformational representations are obtained by a 3D GNN SchNet (Schütt et al., 2017):

$$\boldsymbol{y} \triangleq \boldsymbol{H}_{3D} = \text{SchNet}(g_{3D}) = \text{SchNet}(\mathcal{G}, \mathbf{R}), \tag{186}$$

where $\boldsymbol{H}_{3\text{D}} = [h_{3\text{D}}^0, h_{3\text{D}}^1, \ldots]$, and $h_{3\text{D}}^i$ is the $i$-th node representation.

**An SE(3)-Equivariant Conformation Generation** The first objective is the conditional generation from topology to conformation, $p(\boldsymbol{y}|\boldsymbol{x})$, implemented as SUBGDIFF. The denoising network we adopt is the SE(3)-equivariance network ($S_\theta^{2\text{D}\rightarrow 3\text{D}}$) used in MoleculeSDE. The details of the network architecture refer to Liu et al. (2023a).

Therefore, the training objective from 2D topology graph to 3D confirmation is:

$$
\begin{aligned}
\mathcal{L}_{2\text{D}\rightarrow 3\text{D}} = \mathbb{E}_{\boldsymbol{x},\mathbf{R},t,\mathbf{s}_t} \mathbb{E}_{\mathbf{R}_t|\mathbf{R}} \\
\left[\left\|\text{diag}(\mathbf{s}_t)(\epsilon - S_\theta^{2\text{D}\rightarrow 3\text{D}}(\boldsymbol{x},\mathbf{R}_t,t))\right\|_2^2 + \text{BCE}(\mathbf{s}_t, s_\vartheta^{2\text{D}\rightarrow 3\text{D}}(\boldsymbol{x},\mathbf{R}_t,t))\right],
\end{aligned}
\tag{187}
$$

where $s_\vartheta^{2\text{D}\rightarrow 3\text{D}}(\boldsymbol{x},\mathbf{R}_t,t)$ gets the invariant feature from $S_\theta$ and introduces a mask head (MLP) to read out the mask prediction.

**An SE(3)-Invariant Topology Generation.** The second objective is to reconstruct the 2D topology from 3D conformation, i.e., $p(\boldsymbol{x}|\boldsymbol{y})$. We also use the SE(3)-invariant score network $S_\theta^{3\text{D}\rightarrow 2\text{D}}$ proposed by MoleculeSDE. The details of the network architecture refer to Liu et al. (2023a). For modeling $S_\theta^{3\text{D}\rightarrow 2\text{D}}$, it needs to satisfy the SE(3)-invariance symmetry property. The inputs are 3D conformational representation $\boldsymbol{y}$, the noised 2D information $\boldsymbol{x}_t$ at time $t$, and time $t$. The output of $S_\theta^{3\text{D}\rightarrow 2\text{D}}$ is the Gaussian noise, as $(\epsilon^\mathbf{X}, \epsilon^\mathbf{E})$. The diffused 2D information contains two parts: $\boldsymbol{x}_t = (\mathbf{X}_t, \mathbf{E}_t)$. For node feature $\mathbf{X}$, the training objective is

$$
\mathcal{L}_{3\text{D}\rightarrow 2\text{D}}^\mathbf{X} = \mathbb{E}_{\mathbf{X},\boldsymbol{y}} \mathbb{E}_{t,\mathbf{s}_t} \mathbb{E}_{\mathbf{X}_t|\mathbf{X}}
\tag{188}
$$

$$
\left[\left\|\text{diag}(\mathbf{s}_t)(\epsilon - S_\theta^{3\text{D}\rightarrow 2\text{D}}(\boldsymbol{y},\mathbf{X}_t,t))\right\|_2^2 + \text{BCE}(\mathbf{s}_t, s_\vartheta^{3\text{D}\rightarrow 2\text{D}}(\boldsymbol{y},\mathbf{X}_t,t))\right].
\tag{189}
$$

For edge feature $\mathbf{E}$, we define a mask matrix $\mathbf{S}$ from mask vector $\mathbf{s}$: $\mathbf{S}_{ij} = 1$ if $\mathbf{s}_i = 1$ or $\mathbf{s}_j = 1$, otherwise, $\mathbf{S}_{ij} = 0$. Eventually, the ojective can be written as:

$$
\mathcal{L}_{3\text{D}\rightarrow 2\text{D}}^\mathbf{E} = \mathbb{E}_{\mathbf{E},\boldsymbol{y}} \mathbb{E}_{t,\mathbf{s}_t} \mathbb{E}_{\mathbf{E}_t|\mathbf{E}}
\tag{190}
$$

$$
\left[\left\|\mathbf{S}_t \odot (\epsilon - S_\theta^{3\text{D}\rightarrow 2\text{D}}(\boldsymbol{y},\mathbf{E}_t,t))\right\|_2^2 + \text{BCE}(\mathbf{s}_t, s_\vartheta^{3\text{D}\rightarrow 2\text{D}}(\boldsymbol{y},\mathbf{E}_t,t))\right],
\tag{191}
$$

Then the score network $S_\theta^{3\text{D}\rightarrow 2\text{D}}$ is also decomposed into two parts for the atoms and bonds: $S_\theta^{\mathbf{X}_t}(\boldsymbol{x}_t)$ and $S_\theta^{\mathbf{E}_t}(\boldsymbol{x}_t)$. Similarly, the mask predictor $s_\vartheta^{3\text{D}\rightarrow 2\text{D}}$ is also decomposed into two parts for the atoms and bonds: $s_\vartheta^{\mathbf{X}_t}(\boldsymbol{x}_t)$ and $s_\vartheta^{\mathbf{E}_t}(\boldsymbol{x}_t)$.

Similar to the topology to conformation generation procedure, the $s_\vartheta^{3\text{D}\rightarrow 2\text{D}}(\boldsymbol{x},\mathbf{R}_t,t)$ gets the invariant feature from $S_\theta^{3\text{D}\rightarrow 2\text{D}}$ and introduces a mask head (MLP) to read out the mask prediction.

**Learning.** Following MoleculeSDE, we incorporate a contrastive loss called EBM-NCE Liu et al. (2021). EBM-NCE provides an alternative approach to estimate the mutual information $I(X;Y)$ and is anticipated to complement the generative self-supervised learning (SSL) method. As a result, the ultimate objective is:

$$
\mathcal{L}_{\text{overall}} = \alpha_1 \mathcal{L}_{\text{Contrastive}} + \alpha_2 \mathcal{L}_{2\text{D}\rightarrow 3\text{D}} + \alpha_3 (\mathcal{L}_{3\text{D}\rightarrow 2\text{D}}^\mathbf{X} + \mathcal{L}_{3\text{D}\rightarrow 2\text{D}}^\mathbf{E}),
\tag{192}
$$

where $\alpha_1, \alpha_2, \alpha_3$ are three coefficient hyperparameters.

### F.3.2 DATASET AND SETTINGS

**Dataset.** For pretraining, following MoleculeSDE, we use PCQM4Mv2 (Hu et al., 2020b). It's a sub-dataset of PubChemQC (Nakata & Shimazaki, 2017) with 3.4 million molecules with both the topological graph and geometric conformations.

**Baselines for 3D property prediction** We begin by incorporating three coordinate-MI-unaware SSL methods: (1) Type Prediction, which aims to predict the atom type of masked atoms; (2) Angle Prediction, which focuses on predicting the angle among triplet atoms, specifically the bond angle prediction; (3) 3D InfoGraph, which adopts the contrastive learning paradigm by considering the node-graph pair from the same molecule geometry as positive and negative otherwise. Next, in

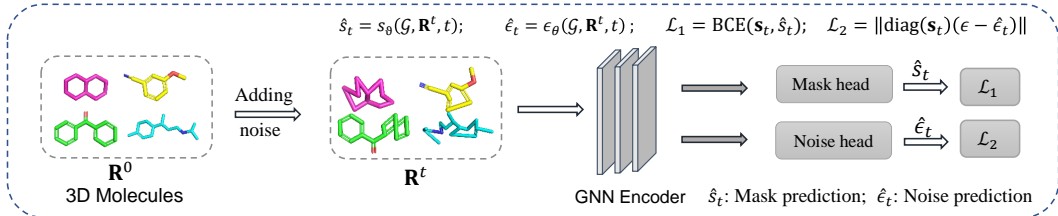

$$\hat{s}_t = s_\theta(\mathcal{G}, \mathbf{R}^t, t); \quad \hat{\epsilon}_t = \epsilon_\theta(\mathcal{G}, \mathbf{R}^t, t); \quad \mathcal{L}_1 = \text{BCE}(\mathbf{s}_t, \hat{s}_t); \quad \mathcal{L}_2 = \|\text{diag}(\mathbf{s}_t)(\epsilon - \hat{\epsilon}_t)\|$$

Figure 6: The model architecture for denoising SUBGDIFF.

accordance with the work of (Liu et al., 2023b), we include two contrastive baselines: (4) GeoSSL-InfoNCE (Oord et al., 2018) and (5) GeoSSL-EBM-NCE (Liu et al., 2021). We also incorporate a generative SSL baseline named (6) GeoSSL-RR (RR for Representation Reconstruction). The above baselines are pre-trained on a subset of 1M molecules with 3D geometries from Molecule3D (Xu et al., 2021c) and we reuse the results reported by Liu et al. (2023b) with SchNet as backbone.

**Baselines for 2D topology pretraining.** We pick up the most promising ones as follows. Attr-Mask (Hu et al., 2020a; Liu et al., 2019), ContexPred (Hu et al., 2020a), InfoGraph (Sun et al., 2020), and MolCLR (Wang et al., 2022b).

**Baselines for 2D and 3D multi-modality pretraining.** We include MoleculeSDE(Liu et al., 2023a)(Variance Exploding (VE) and Variance Preserving (VP)) as a crucial baseline to verify the effectiveness of our methods due to the same pertaining framework. We reproduce the results from the released (Code).

**Compared with GEODIFF.** We directly reuse the pre-trained model of the molecular conformation generation in sec. 5.2 for fine-tuning, to compare our method with GEODIFF from naive denoising pretraining perspective (Zaidi et al., 2023). The results are shown in Table 11.

**Results on MD17.** Regarding 3D fine-tuning on MD17, we follow the literature (Schütt et al., 2017; 2021; Liu et al., 2023b) of using 1K for training and 1K for validation, while the test set (from 48K to 991K) is much larger. The results can be seen in Table 12.

Table 11: Results on 12 quantum mechanics prediction tasks from QM9. We take 110K for training, 10K for validation, and 11K for testing. The evaluation is mean absolute error (MAE), and the best and the second best results are marked in bold and underlined, respectively. The backbone is **SchNet**.

| Pretraining | Alpha ↓ | Gap ↓ | HOMO ↓ | LUMO ↓ | Mu ↓ | Cv ↓ | G298 ↓ | H298 ↓ | R2 ↓ | U298 ↓ | U0 ↓ | Zpve ↓ |
|---|---|---|---|---|---|---|---|---|---|---|---|---|
| GEODIFF | 0.078 | 51.84 | 30.88 | 28.29 | 0.028 | 0.035 | 15.35 | 11.37 | 0.132 | 15.76 | 15.24 | 1.869 |
| SUBGDIFF | 0.076▲ | 50.80▲ | 31.15▼ | 26.62▲ | 0.025▲ | 0.032▲ | 14.92▲ | 12.86▲ | 0.129▲ | 14.74▲ | 14.53▲ | 1.710▲ |

Table 12: Results on eight **force** prediction tasks from MD17. We take 1K for training, 1K for validation, and 48K to 991K molecules for the test concerning different tasks. The evaluation is mean absolute error, and the best results are marked in bold and underlined, respectively.

| Pretraining | Aspirin ↓ | Benzene ↓ | Ethanol ↓ | Malonaldehyde ↓ | Naphthalene ↓ | Salicylic ↓ | Toluene ↓ | Uracil ↓ |
|---|---|---|---|---|---|---|---|---|
| – (random init) | 1.203 | 0.380 | 0.386 | 0.794 | 0.587 | 0.826 | 0.568 | 0.773 |
| Type Prediction | 1.383 | 0.402 | 0.450 | 0.879 | 0.622 | 1.028 | 0.662 | 0.840 |
| Distance Prediction | 1.427 | 0.396 | 0.434 | 0.818 | 0.793 | 0.952 | 0.509 | 1.567 |
| Angle Prediction | 1.542 | 0.447 | 0.669 | 1.022 | 0.680 | 1.032 | 0.623 | 0.768 |
| 3D InfoGraph | 1.610 | 0.415 | 0.560 | 0.900 | 0.788 | 1.278 | 0.768 | 1.110 |
| RR | 1.215 | 0.393 | 0.514 | 1.092 | 0.596 | 0.847 | 0.570 | 0.711 |
| InfoNCE | 1.132 | 0.395 | 0.466 | 0.888 | 0.542 | 0.831 | 0.554 | 0.664 |
| EBM-NCE | 1.251 | 0.373 | 0.457 | 0.829 | 0.512 | 0.990 | 0.560 | 0.742 |
| 3D InfoMax | 1.142 | 0.388 | 0.469 | 0.731 | 0.785 | 0.798 | 0.516 | 0.640 |
| GraphMVP | 1.126 | 0.377 | 0.430 | 0.726 | 0.498 | 0.740 | 0.508 | 0.620 |
| GeoSSL-1L | 1.364 | 0.391 | 0.432 | 0.830 | 0.599 | 0.817 | 0.628 | 0.607 |
| GeoSSL | 1.107 | 0.360 | 0.357 | 0.737 | 0.568 | 0.902 | 0.484 | 0.502 |
| MoleculeSDE (VE) | 1.112 | 0.304 | 0.282 | 0.520 | 0.455 | 0.725 | 0.515 | 0.447 |
| MoleculeSDE (VP) | 1.244 | 0.315 | 0.338 | **0.488** | 0.432 | 0.712 | 0.478 | 0.468 |
| Ours | **0.880** | **0.252** | **0.258** | 0.491 | **0.325** | 0.572 | **0.362** | **0.420** |

