# OpenReview forum: "Subgraph Diffusion for 3D Molecular Representation Learning: Combining Continuous and Discrete"
_ICLR.cc/2024/Conference — Submitted to ICLR 2024_

### Official Review · Reviewer_aJi2 · 2023-10-29

**Soundness:** 3 good
**Presentation:** 2 fair
**Contribution:** 3 good
**Rating:** 6
**Confidence:** 4

**Summary:**

The work proposes a mask diffusion model for 3D molecular representation learning and the conformation generation task. The proposed diffusion model selects a subset of atoms and adds noise from a Gaussian distribution to the atom's 3D coordinates. In the reverse process, a mask predictor is used to recover the 3D information of the noisy subgraph. Finally, the proposed diffusion model is validated using the QM9 conformation generation and quantum property prediction tasks.

**Strengths:**

First, the study uses a diffusion model for molecular representation learning based on 3D coordinates. The topic is attractive and holds promise in many important applications for drug discoveries.

Second, the proposed mask diffusion model on subgraphs is novel. The experiment results look good.

Third, the proposed method has a good theoretical motivation.

**Weaknesses:**

Weakness 1: The presentation of the work could be improved. The use of mathematical symbols is burdensome and difficult to follow. Explanatory figures, such as Figure 4, are not self-explanatory due to numerous unexplained symbols. Algorithm 1 is helpful for readers trying to understand the method's training, but it lacks many details. For instance, is the mask predictor different from the noise predictor? Which neural network architectures are used?

Weakness 2: The proposed method has only been validated on QM9, which encompasses four types of atoms. Given that the periodic table consists of 118 elements, the paper doesn't address the extent to which the model can generalize in real-world scenarios.

**Questions:**

Q1: Can the diffusion model be adapted to accommodate node features and graph structures? How dependent is the model on the graph structure?

Q2: Could the model be used for other important tasks such as the datasets from MoleculeNet [1] or OGBG [2]?

Q3: Can the model be compared against other graph representation learning methods for molecular graphs that do not process 3D coordinates? Such a comparison could more effectively underscore the significance of explicitly incorporating 3D coordinates in practical applications.


Ref.

1. MoleculeNet: a benchmark for molecular machine learning. Chemical Science.

2. Open Graph Benchmark: Datasets for Machine Learning on Graphs. NeurIPS 2020.

---

> ### Author Response · Authors · 2023-11-20
> **Reply to Reviewer aJi2 (1/3)**
>
> We thank the reviewer for the positive feedback and constructive comments. We provide pointwise responses below.
>
> > **Weakness 1**: The presentation of the work could be improved. The use of mathematical symbols is burdensome and difficult to follow. Explanatory figures, such as Figure 4, are not self-explanatory due to numerous unexplained symbols. Algorithm 1 is helpful for readers trying to understand the method's training, but it lacks many details. For instance, is the mask predictor different from the noise predictor? Which neural network architectures are used?
>
> **Response 1:** Thanks so much for your helpful suggestions! We will carefully improve the presentation and clarify the mathematical notations. We have updated Figure 4 to make it easier to follow. For Algorithm 1, **1)** The mask predictor and the noise predictor are different. **2)** Our network consists of a GNN encoder (e.g. SchNet) for extracting the 3D graph information, a noise predictor (e.g. an MLP), and a mask predictor (e.g. an MLP). Notably, these two predictors share the same GNN encoder. **3)** In our experiments, we adopt the graph field network (GFN) from GeoDiff as the GNN encoder to extract 3D molecular information. Specifically, in the $l$-th layer, the GFN receives node embeddings $\mathbf{h}^l \in \mathbb{R}^{n \times b}$ (where $b$ represents the feature dimension) and corresponding coordinate embeddings $\mathbf{x}^l \in \mathbb{R}^{n \times 3}$ as input. It then produces the output $\mathbf{h}^{l+1}$ and $\mathbf{x}^{l+1}$ according to the following process:
>
> $$
>          \mathbf{m} _{ij}^l =\Phi^l _{m}\left(\mathbf{h} _{i}^{l}, \mathbf{h} _{j}^{l},\|\mathbf{x} _{i}^{l}-\mathbf{x} _{j}^{l}\|^{2}, e _{ij}; \theta_m \right)  $$
>
> $$\mathbf{h} _{i}^{l+1} =\Phi^l _{h}(\mathbf{h} _{i}^l, \sum _{j \in \mathcal{N}(i)} \mathbf{m} _{ij}^l; \theta _h) $$
>      $$\mathbf{x} _{i}^{l+1} = \sum _{j \in \mathcal{N}(i)}\frac{1}{d _{ij}}(R _{i} - R _{j}) \Phi^l _{x}(\mathbf{m} _{ij}^l ; \theta _x )$$
>
> where $\Phi$ are implemented as feed-forward networks and $d_{ij}$ denotes interatomic distances. The initial embedding $\mathbf{h}^0$ is composed of atom embedding and time step embedding while $\mathbf{x}^0$ represents atomic coordinates. $\mathcal N(i)$ is the neighborhood of $i^{th}$ node.
>
> Eventually, the Gaussian noise and mask can be predicted as follows:
> $$ \hat{\epsilon}_i = \mathbf{x}_i^L; \quad
>      \hat{s}_i = \text{MLP}(\mathbf{h}_i^L) $$
>
> where $\hat{\epsilon}_i$ is equivalent and $\hat{s}_i$ is invariant. We added a schematic diagram (Figure 6) in the appendix of the revised version.
> We will add these clarifications in the revised version.
>
> ---
> > **Weakness 2**: The proposed method has only been validated on QM9, which encompasses four types of atoms. Given that the periodic table consists of 118 elements, the paper doesn't address the extent to which the model can be generalized in real-world scenarios.
>
> **Response 2:** Thanks for your insightful comments. We invite you to check our results on domain generation. We train the model with QM9 and test it on the Drugs dataset, which contains larger molecules and has more types of atoms than QM9. The results are posted in the manuscript (sec. 5.2), and we also provide them below (Table 1 and Table 2). The results concerning training on QM9 and testing on Drugs suggest that SubGDiff consistently outperforms GeoDiff by a large margin, suggesting the proposed method can generalize well in real-world scenarios. Table 2 further demonstrates the generalization capability of our method from Drugs to QM9.
>
> **Table 1: Results on the {GEOM-Drugs} dataset. The threshold $\delta=1.25A$**
> | Models    |  Train data  |Test data|    CCOV-R (%)  Mean $\uparrow$  |  COV-R (%) Median $\uparrow$ | MAT-R (A)  Mean $\downarrow$   | MAT-R (A) Median  $\downarrow$ |
> |-----------|-------|-----------|--------|------|--------|---------|
> | CVGAE    | Drugs | Drugs | 0.00   | 0.00   | 3.0702 | 2.9937  |
> | GraphDG  | Drugs | Drugs |8.27   | 0.00   | 1.9722 | 1.9845  |
> | GeoDiff  | *QM9*   |Drugs | 7.99   | 0.00   | 2.7704 | 2.3297  |
> | **Ours**   | *QM9*   |Drugs | **24.01**  | **9.93**   | **1.6128** | **1.5819**  |
>
>
> **Table 2: Results on the {GEOM-QM9} dataset. The threshold $\delta=0.5A$**
> | Models    | Train data |Test data|    CCOV-R (%)  Mean $\uparrow$  |  COV-R (%) Median $\uparrow$ | MAT-R (A)  Mean $\downarrow$   | MAT-R (A) Median  $\downarrow$ |
> |---|-------|-------|------|--------|------|------|
> | CVGAE    | QM9      | QM9  | 0.09   | 0.00   | 1.6713  | 1.6088  |
> | GraphDG  | QM9      | QM9  | 73.33  | 84.21  | 0.4245  | 0.3973  |
> | CGCF     | QM9      | QM9  | 78.05  | 82.48  | 0.4219  | 0.3900  |
> | ConfVAE  | QM9      | QM9  | 77.84  | 88.20  | 0.4154  | 0.3739  |
> | GeoMol   | QM9      | QM9 | 71.26  | 72.00  | 0.3731  | 0.3731  |
> | GeoDiff  | *Drugs*    | QM9  | 74.94  | 79.15  | 0.3492  | 0.3392  |
> | **Ours**   | *Drugs*  | QM9    | **83.50**  | **88.70**  | **0.3116**  | **0.3075**  |

---

> ### Author Response · Authors · 2023-11-20
> **Reply to Reviewer aJi2 (2/3)**
>
> > **Q1**: Can the diffusion model be adapted to accommodate node features and graph structures? How dependent is the model on the graph structure?
>
> **A1:** **1)** The proposed diffusion model can be extended to node features and graph structures. For node features, similar to the proposed MaskedDiff, we only need to operate the diffusion model on node feature space instead of Euclidean space (3D coordinate). Regarding graph structure, for example in Digress[1], the transition matrices can be designed as $Q_m=diag(Q_n,I_{m-n})$, which means adding noise to the edges of the selected subgraph of size $n$.  **2)** In our model, the graph structure provides the initial graph topology for subgraph decomposition; in other words, the mask distribution $p(S)$ depends on the graph topology. Besides, the graph structure serves as a condition for the conformational distribution $p(R|\mathcal{G}(A,X))$ and is therefore input to the denoising network to assist in predicting Gaussian noise.
>
> **Reference:**
>
> *[1] DiGress: Discrete Denoising diffusion for graph generation, ICLR 2023*
>
>
> ---
> > **Q2**: Could the model be used for other important tasks such as the datasets from MoleculeNet [1] or OGBG [2]?
>
> **A2:** We conduct additional experiments to evaluate our method on eight classification tasks from MoleculeNet[1]. Utilizing our MaskedDiff to replace the SDE in MoleculeSDE[3], a pretraining framework, we established a MaskedDiff-based pretraining framework.  Following the procedure of MoleculeSDE[3], we pretrain the framework on the PCQM4Mv2 dataset and then fine-tune it on the MoleculeNet[1] dataset. The results can be found in Table 3. It is clear that our method consistently outperforms other methods across all tasks.
>
> **References:**
>
> *[1]MoleculeNet: a benchmark for molecular machine learning. Chemical Science.*
>
> *[2]Open Graph Benchmark: Datasets for Machine Learning on Graphs. NeurIPS 2020.*
>
> *[3] Liu S, Du W, Ma Z M, et al. A group symmetric stochastic differential equation model for molecule multi-modal pretraining//International Conference on Machine Learning. PMLR, 2023: 21497-21526.*
>
>
> **Table 3: Results for molecular property prediction tasks (with 2D topology only). The backbone is GIN.**
> | Pre-training      | BBBP $\uparrow$ | Tox21 $\uparrow$ | ToxCast $\uparrow$ | Sider $\uparrow$ | ClinTox $\uparrow$ | MUV $\uparrow$ | HIV $\uparrow$ | Bace $\uparrow$ | Avg $\uparrow$ |
> |-------------------|-----------------|------------------|--------------------|------------------|--------------------|----------------|----------------|-----------------|----------------|
> | -- (random init)  | 68.1 $\pm$ 0.59   | 75.3 $\pm$ 0.22    | 62.1 $\pm$ 0.19      | 57.0 $\pm$ 1.33    | 83.7 $\pm$ 2.93      | 74.6 $\pm$ 2.35  | 75.2 $\pm$ 0.70  | 76.7  $\pm$  2.51   | 71.60          |
> | AttrMask          | 65.0 $\pm$ 2.36   | 74.8 $\pm$ 0.25    | 62.9 $\pm$ 0.11      | 61.2 $\pm$ 0.12    | 87.7 $\pm$ 1.19      | 73.4 $\pm$ 2.02  | 76.8 $\pm$ 0.53  | 79.7 $\pm$ 0.33   | 72.68          |
> | ContextPred       | 65.7 $\pm$ 0.62   | 74.2 $\pm$ 0.06    | 62.5 $\pm$ 0.31      | 62.2 $\pm$ 0.59    | 77.2 $\pm$ 0.88      | 75.3 $\pm$ 1.57  | 77.1 $\pm$ 0.86  | 76.0 $\pm$ 2.08   | 71.28          |
> | InfoGraph         | 67.5 $\pm$ 0.11   | 73.2 $\pm$ 0.43    | 63.7 $\pm$ 0.50      | 59.9 $\pm$ 0.30    | 76.5 $\pm$ 1.07      | 74.1 $\pm$ 0.74  | 75.1 $\pm$ 0.99  | 77.8 $\pm$ 0.88   | 70.96          |
> | MolCLR            | 66.6 $\pm$ 1.89   | 73.0 $\pm$ 0.16    | 62.9 $\pm$ 0.38      | 57.5 $\pm$ 1.77    | 86.1 $\pm$ 0.95      | 72.5 $\pm$ 2.38  | 76.2 $\pm$ 1.51  | 71.5 $\pm$ 3.17   | 70.79          |
> | 3D InfoMax        | 68.3 $\pm$ 1.12   | 76.1 $\pm$ 0.18    | 64.8 $\pm$ 0.25      | 60.6 $\pm$ 0.78    | 79.9 $\pm$ 3.49      | 74.4 $\pm$ 2.45  | 75.9 $\pm$ 0.59  | 79.7 $\pm$ 1.54   | 72.47          |
> | GraphMVP          | 69.4 $\pm$ 0.21   | 76.2 $\pm$ 0.38    | 64.5 $\pm$ 0.20      | 60.5 $\pm$ 0.25    | 86.5 $\pm$ 1.70      | 76.2 $\pm$ 2.28  | 76.2 $\pm$ 0.81  | 79.8 $\pm$ 0.74   | 73.66          |
> | MoleculeSDE (VE)  | 68.3 $\pm$ 0.25   | 76.9 $\pm$ 0.23    | 64.7 $\pm$ 0.06      | 60.2 $\pm$ 0.29    | 80.8 $\pm$ 2.53      | 76.8 $\pm$ 1.71  | 77.0 $\pm$ 1.68  | 79.9 $\pm$ 1.76   | 73.15          |
> | MoleculeSDE(VP)   | 70.1 $\pm$ 1.35   | 77.0 $\pm$ 0.12    | 64.0 $\pm$ 0.07      | 60.8 $\pm$ 1.04    | 82.6 $\pm$ 3.64      | 76.6 $\pm$ 3.25  | 77.3 $\pm$ 1.31  | 81.4 $\pm$ 0.66   | 73.73          |
> |  **Ours**          | **70.2 $\pm$ 2.23**   | **77.2 $\pm$ 0.39**    | **65.0 $\pm$ 0.48**     | **62.2 $\pm$ 0.974**    | **88.2 $\pm$ 1.57**      | **77.3 $\pm$ 1.17**  | **77.6 $\pm$ 0.5**1  | **82.1 $\pm$ 0.96**  | **74.85**

---

> ### Author Response · Authors · 2023-11-20
> **Reply to Reviewer aJi2 (3/3)**
>
> > **Q3**: Can the model be compared against other graph representation learning methods for molecular graphs that do not process 3D coordinates? Such a comparison could more effectively underscore the significance of explicitly incorporating 3D coordinates in practical applications.
>
> **A3:** Thanks for this constructive suggestion! Following the advice, we compare our method with various graph representation learning methods on 2D molecular property prediction tasks, and the results are presented in Table 3 above. The baselines, including AttrMask[1], ContexPred[1], InfoGraph[2], and MolCLR[3] are all exclusively 2D graph learning methods that don't process 3D coordinates, while other methods all leverage the 3D coordinates information. As indicated in the results in Table 3, methods incorporating 3D coordinates demonstrate a significant performance advantage over those without 3D information. Furthermore, our approach stands out by enabling the simultaneous exploration of 2D and 3D information, leading to state-of-the-art performances.
>
> **References:**
>
> *[1] Hu, W., et al. "Strategies For Pre-training Graph Neural Networks." International Conference on Learning Representations (ICLR). 2020.*
>
> *[2] Sun, Fan-Yun, et al. "InfoGraph: Unsupervised and Semi-supervised Graph-Level Representation Learning via Mutual Information Maximization." International Conference on Learning Representations. 2019.*
>
> *[3]Wang, Yuyang, et al. "Molecular contrastive learning of representations via graph neural networks." Nature Machine Intelligence 4.3 (2022): 279-287.*

---

### Official Review · Reviewer_7f3c · 2023-10-30

**Soundness:** 3 good
**Presentation:** 3 good
**Contribution:** 2 fair
**Rating:** 6
**Confidence:** 3

**Summary:**

The paper proposed a new molecular generation and representation learning method. The paper first proposed MaskedDiff for better representation learning for molecular graphs, and then further propose SubgDiff to make the model capable for molecule generation. Experiments demonstrate the effectiveness of the method for both generation and property prediction.

**Strengths:**

1. Presentation is clear and both method and experiment are well explained.
2. The author provides good mathematical details and analysis for the proposed method.
3. The idea overall is interesting, where the author draw inspiration from subgraph-based representation learning method and adopt it address the generation task.

**Weaknesses:**

The main weakness in my mind is about the performance. The generation quality in Tab1 and Tab5 seems even worse than the baseline GeoDiff, and the property prediction results in Tab2 is also not consistently better than baselines and GeoDiff.

**Questions:**

I may missed but didn't ind it: what's the number of diffusion timesteps T for each experiment's config? This is important for audience to have a better understanding of the method's sampling efficiency.

---

> ### Author Response · Authors · 2023-11-20
> **Reply to Reviewer 7f3c (1/3)**
>
> Thank you for your endorsement and the positive comments! Below please find the responses to some specific comments.
>
> > **W1:** The main weakness in my mind is about the performance. The generation quality in Tab1 and Tab5 seems even worse than the baseline GeoDiff, and the property prediction results in Tab2 are also not consistently better than baselines and GeoDiff.
>
> **Response 1:** Thanks for your careful reviews! **1)** For a more comprehensive comparison with GeoDiff, we include results using 200 and 500 diffusion steps (Table 4 below). The findings suggest that our method significantly outperforms GeoDiff when reducing the number of diffusion steps, indicating an improvement in sampling efficiency.  **2)** Additionally, we conduct extra experiments to support property prediction tasks, and the results are shown in Table 1, Table 2, and Table 3 below. In these experiments, we employ another stronger score network (a pretraining framework provided by MoleculeSDE[1]). The new experimental results suggest that our methods can significantly outperform the baselines consistently under the self-supervised learning context, implying our method can be used as an important technique for improving the pretrained model to capture the substructure information during training.
>
> *[1] Liu S, Du W, Ma Z M, et al. A group symmetric stochastic differential equation model for molecule multi-modal pretraining//International Conference on Machine Learning. PMLR, 2023: 21497-21526.*
>
> **Table 1: Results for molecular property prediction tasks (with 2D topology only)**
> | Pre-training      | BBBP $\uparrow$ | Tox21 $\uparrow$ | ToxCast $\uparrow$ | Sider $\uparrow$ | ClinTox $\uparrow$ | MUV $\uparrow$ | HIV $\uparrow$ | Bace $\uparrow$ | Avg $\uparrow$ |
> |-------------------|-----------------|------------------|--------------------|------------------|--------------------|----------------|----------------|-----------------|----------------|
> | -- (random init)  | 68.1 $\pm$ 0.59   | 75.3 $\pm$ 0.22    | 62.1 $\pm$ 0.19      | 57.0 $\pm$ 1.33    | 83.7 $\pm$ 2.93      | 74.6 $\pm$ 2.35  | 75.2 $\pm$ 0.70  | 76.7  $\pm$  2.51   | 71.60          |
> | AttrMask          | 65.0 $\pm$ 2.36   | 74.8 $\pm$ 0.25    | 62.9 $\pm$ 0.11      | 61.2 $\pm$ 0.12    | 87.7 $\pm$ 1.19      | 73.4 $\pm$ 2.02  | 76.8 $\pm$ 0.53  | 79.7 $\pm$ 0.33   | 72.68          |
> | ContextPred       | 65.7 $\pm$ 0.62   | 74.2 $\pm$ 0.06    | 62.5 $\pm$ 0.31      | 62.2 $\pm$ 0.59    | 77.2 $\pm$ 0.88      | 75.3 $\pm$ 1.57  | 77.1 $\pm$ 0.86  | 76.0 $\pm$ 2.08   | 71.28          |
> | InfoGraph         | 67.5 $\pm$ 0.11   | 73.2 $\pm$ 0.43    | 63.7 $\pm$ 0.50      | 59.9 $\pm$ 0.30    | 76.5 $\pm$ 1.07      | 74.1 $\pm$ 0.74  | 75.1 $\pm$ 0.99  | 77.8 $\pm$ 0.88   | 70.96          |
> | MolCLR            | 66.6 $\pm$ 1.89   | 73.0 $\pm$ 0.16    | 62.9 $\pm$ 0.38      | 57.5 $\pm$ 1.77    | 86.1 $\pm$ 0.95      | 72.5 $\pm$ 2.38  | 76.2 $\pm$ 1.51  | 71.5 $\pm$ 3.17   | 70.79          |
> | 3D InfoMax        | 68.3 $\pm$ 1.12   | 76.1 $\pm$ 0.18    | 64.8 $\pm$ 0.25      | 60.6 $\pm$ 0.78    | 79.9 $\pm$ 3.49      | 74.4 $\pm$ 2.45  | 75.9 $\pm$ 0.59  | 79.7 $\pm$ 1.54   | 72.47          |
> | GraphMVP          | 69.4 $\pm$ 0.21   | 76.2 $\pm$ 0.38    | 64.5 $\pm$ 0.20      | 60.5 $\pm$ 0.25    | 86.5 $\pm$ 1.70      | 76.2 $\pm$ 2.28  | 76.2 $\pm$ 0.81  | 79.8 $\pm$ 0.74   | 73.66          |
> | MoleculeSDE (VE)  | 68.3 $\pm$ 0.25   | 76.9 $\pm$ 0.23    | 64.7 $\pm$ 0.06      | 60.2 $\pm$ 0.29    | 80.8 $\pm$ 2.53      | 76.8 $\pm$ 1.71  | 77.0 $\pm$ 1.68  | 79.9 $\pm$ 1.76   | 73.15          |
> | MoleculeSDE(VP)   | 70.1 $\pm$ 1.35   | 77.0 $\pm$ 0.12    | 64.0 $\pm$ 0.07      | 60.8 $\pm$ 1.04    | 82.6 $\pm$ 3.64      | 76.6 $\pm$ 3.25  | 77.3 $\pm$ 1.31  | 81.4 $\pm$ 0.66   | 73.73          |
> |  **Ours**          | **70.2 $\pm$ 2.23**   | **77.2 $\pm$ 0.39**    | **65.0 $\pm$ 0.48**     | **62.2 $\pm$ 0.974**    | **88.2 $\pm$ 1.57**      | **77.3 $\pm$ 1.17**  | **77.6 $\pm$ 0.5**1  | **82.1 $\pm$ 0.96**  | **74.85**

---

> ### Author Response · Authors · 2023-11-20
> **Reply to Reviewer 7f3c (2/3)**
>
> **Table 2: Results on 12 quantum mechanics prediction tasks from QM9. We take 110K for training, 10K for validation, and 11K for testing. The evaluation is mean absolute error~(MAE), and the best results are marked in bold.  The backbone is SchNet.**
> | Pretraining | Alpha ↓ | Gap ↓ | HOMO ↓ | LUMO ↓ | Mu ↓ | Cv ↓ | G298 ↓ | H298 ↓ | R2 ↓ | U298 ↓ | U0 ↓ | Zpve ↓ |
> |-------------|---------|-------|--------|--------|------|------|--------|--------|------|--------|------|---------|
> | Random init | 0.070 | 50.59 | 32.53 | 26.33 | 0.029 | 0.032 | 14.68 | 14.85 | 0.122 | 14.70 | 14.44 | 1.698 |
> | Supervised | 0.070 | 51.34 | 32.62 | 27.61 | 0.030 | 0.032 | 14.08 | 14.09 | 0.141 | 14.13 | 13.25 | 1.727 |
> | Type Prediction | 0.084 | 56.07 | 34.55 | 30.65 | 0.040 | 0.034 | 18.79 | 19.39 | 0.201 | 19.29 | 18.86 | 2.001 |
> | Angle Prediction | 0.084 | 57.01 | 37.51 | 30.92 | 0.037 | 0.034 | 15.81 | 15.89 | 0.149 | 16.41 | 15.76 | 1.850 |
> | 3D InfoGraph | 0.076 | 53.33 | 33.92 | 28.55 | 0.030 | 0.032 | 15.97 | 16.28 | 0.117 | 16.17 | 15.96 | 1.666 |
> | GeossL-RR | 0.073 | 52.57 | 34.44 | 28.41 | 0.033 | 0.038 | 15.74 | 16.11 | 0.194 | 15.58 | 14.76 | 1.804 |
> | GeossL-InfoNCE | 0.075 | 53.00 | 34.29 | 27.03 | 0.029 | 0.033 | 15.67 | 15.53 | 0.125 | 15.79 | 14.94 | 1.675 |
> | GeossL-EBM-NCE | 0.073 | 52.86 | 33.74 | 28.07 | 0.031 | 0.032 | 14.02 | 13.65 | 0.121 | 13.70 | 13.45 | 1.677 |
> | MoleculeSDE | 0.062 | 47.74 | 28.02 | 24.60 | 0.028 | 0.029 | 13.25 | 12.70 | 0.120 | 12.68 | 12.93 | 1.643 |
> | **Ours** | **0.054** | **44.88** | **25.45** | **23.75** | **0.027** | **0.028** | **12.03** | **11.46** | **0.110** | **11.32** | **11.25** | **1.568** |
>
> **Table 3: Results on eight force prediction tasks from MD17. We take 1K for training, 1K for validation, and 48K to 991K molecules for the test concerning different tasks. The evaluation is mean absolute error, and the best results are marked in bold.**
> | Pretraining | Aspirin ↓ | Benzene ↓ | Ethanol ↓ | Malonaldehyde ↓ | Naphthalene ↓ | Salicylic ↓ | Toluene ↓ | Uracil ↓ |
> |-------------|-----------|-----------|------------|-----------------|----------------|--------------|------------|-----------|
> | (random init) | 1.203 | 0.380 | 0.386 | 0.794 | 0.587 | 0.826 | 0.568 | 0.773 |
> | Type Prediction | 1.383 | 0.402 | 0.450 | 0.879 | 0.622 | 1.028 | 0.662 | 0.840 |
> | Distance Prediction | 1.427 | 0.396 | 0.434 | 0.818 | 0.793 | 0.952 | 0.509 | 1.567 |
> | Angle Prediction | 1.542 | 0.447 | 0.669 | 1.022 | 0.680 | 1.032 | 0.623 | 0.768 |
> | 3D InfoGraph | 1.610 | 0.415 | 0.560 | 0.900 | 0.788 | 1.278 | 0.768 | 1.110 |
> | RR | 1.215 | 0.393 | 0.514 | 1.092 | 0.596 | 0.847 | 0.570 | 0.711 |
> | InfoNCE | 1.132 | 0.395 | 0.466 | 0.888 | 0.542 | 0.831 | 0.554 | 0.664 |
> | EBM-NCE | 1.251 | 0.373 | 0.457 | 0.829 | 0.512 | 0.990 | 0.560 | 0.742 |
> | 3D InfoMax | 1.142 | 0.388 | 0.469 | 0.731 | 0.785 | 0.798 | 0.516 | 0.640 |
> | GraphMVP | 1.126 | 0.377 | 0.430 | 0.726 | 0.498 | 0.740 | 0.508 | 0.620 |
> | GeoSSL-1L | 1.364 | 0.391 | 0.432 | 0.830 | 0.599 | 0.817 | 0.628 | 0.607 |
> | GeoSSL | 1.107 | 0.360 | 0.357 | 0.737 | 0.568 | 0.902 | 0.484 | 0.502 |
> | MoleculeSDE (VE) | 1.112 | 0.304 | 0.282| 0.520| 0.455 | 0.725 | 0.515 | 0.447 |
> | MoleculeSDE (VP) | 1.244 | 0.315 | 0.338 | 0.488 | 0.432 | 0.712 | 0.478 | 0.468 |
> | **Ours** | **0.880** | **0.252** | 0.398 | 0.556 | **0.325** | **0.572** | **0.362** | **0.420** |

---

> ### Author Response · Authors · 2023-11-20
> **Reply to Reviewer 7f3c (3/3)**
>
> > **Q1:** I may missed it but didn't find it: what's the number of diffusion timesteps T for each experiment's config? This is important for audience to have a better understanding of the method's sampling efficiency.
>
> **A1:** Thanks for the question. In our experiments, the diffusion timestep T is set to 5000, and crucial hyperparameters are provided in Tables 5 and 6 of the Appendix. In addition, we include more results with different numbers of timesteps to further support our method. Please see the Table 4 below. Specifically, we include the experimental results of 200 steps and 500 steps and compare them with GeoDiff. The results suggest that our method significantly outperforms the baseline when the same sampling step is adopted, demonstrating the competitive sampling efficiency of our method.
>
>
> **Table 4: Results on the {GEOM-QM9} dataset. The threshold $\delta=0.5A$**
> | Steps  |  Models   | sampling methods|  COV-R (%)  Mean $\uparrow$  |  COV-R (%) Median $\uparrow$ | MAT-R (A)  Mean $\downarrow$   | MAT-R (A) Median  $\downarrow$ | COV-P (%)  Mean$\uparrow$   |  COV-P (%)  Median$\uparrow$ | MAT-P (A) Mean $\downarrow$    | MAT-P (A) Median $\downarrow$  |
> |-----------|-------|--------|-------|--------|--------|---------|-------|--------|--------|---------|
> |5000|  ConfGF | Langevin dynamics| 88.49 | 94.31 | 0.2673 | 0.2685 | 46.43 | 43.41 | 0.5224 | 0.5124
> |
> |5000 | GeoDiff   | DDPM         |   80.36 |   83.82 |   0.2820 |   0.2799 |   53.66 |   50.85 |   0.6673 |   0.4214 |
> |5000 | Ours   | DDPM         |   90.91$\uparrow$ |   95.59$\uparrow$ |   0.2460$\uparrow$ |   0.2351$\uparrow$ |   50.16 |   48.01 |   0.6114$\uparrow$ |   0.4791 |
> |500|GeoDiff |DDPM|80.20   |   83.59| 0.3617| 0.3412 | 45.49    |  45.45 | 1.1518  |0.5087
> |500|**ours**|DDPM|89.78$\uparrow_{9.58}$  |    94.17$\uparrow_{10.58}$ |  0.2417$\uparrow_{0.12}$| 0.2449$\uparrow_{0.0963}$|50.03$\uparrow_{4.54}$    |  48.31$\uparrow_{2.86}$| 0.5571$\uparrow_{0.5947}$|0.4921$\uparrow_{0.0166}$|
> 200| GeoDiff| DDPM  |      69.90    |   72.04| 0.4222 | 0.4272 | 36.71 |     33.51 | 0.8532 | 0.5554
> 200|  **ours**|  DDPM |    85.53$\uparrow_{15.63}$   |    88.99$\uparrow_{16.95}$ |  0.2994$\uparrow_{0.1228}$ | 0.3033$\uparrow_{0.1239}$ | 47.76$\uparrow_{11.05}$  |    45.89$\uparrow_{12.38}$  | 0.6971$\uparrow_{0.156}$ | 0.5118$\uparrow_{0.0436}$
> |
> |500|GeoDiff |Langevin dynamics|87.80 |     93.66 | 0.3179| 0.3216 | 46.25    |  45.02 | 0.6173  |0.5112
> |500|**ours**|Langevin dynamics|91.40$\uparrow_{3.6}$    |  95.39$\uparrow_{1.73}$ |   0.2543$\uparrow_{0.0636}$ | 0.2601$\uparrow_{0.0615}$|51.71$\uparrow_{5.46}$  |   48.50$\uparrow_{3.48}$| 0.5035$\uparrow_{0.1138}$ | 0.4734$\uparrow_{0.0378}$|
> |200| GeoDiff|Langevin dynamics |       86.60   |   93.09 | 0.3532 | 0.3574 | 42.98     |  42.60 | 0.5563 | 0.5367
> 200| **ours** |Langevin dynamics |       90.36$\uparrow_{3.76}$   |   95.93$\uparrow_{2.84}$ |  0.3064$\uparrow_{0.0468}$ | 0.3098$\uparrow_{0.0476}$ | 48.56$\uparrow_{5.580}$  |    46.46$\uparrow_{3.86}$  | 0.5540$\uparrow_{0.0023}$ | 0.5082$\uparrow_{0.0285}$

---

### Official Review · Reviewer_eamf · 2023-10-30

**Soundness:** 2 fair
**Presentation:** 2 fair
**Contribution:** 2 fair
**Rating:** 5
**Confidence:** 4

**Summary:**

The authors proposed MaskedDiff and SubgDiff, where molecular substructures are preserved during the diffusion process. With this additional inductive bias and tailored diffusion process (taking masking into consideration), the model shows improved performance over existing models on the GEOM dataset.

In general, I find the substructure-preserving diffusion model interesting, yet the results are not convincing enough to demonstrate the superiority of masked diffusion compared to vanilla diffusion models with a much simpler formulation. Please see my concerns below.

**Strengths:**

- Preserving molecular fragments/motifs during the diffusion process helps the model better capture molecular structures.
- The authors performed both molecular property prediction and conformation generation benchmarks, showing that the proposed method is beneficial for representation learning.

**Weaknesses:**

- The number of diffusion steps is the same as that in GeoDiff, i.e., 5000 steps, which is quite a lot. Given this large amount of diffusion steps, the alternating update scheme might work well. However, I am not sure if using more cost-effective sampling methods would still benefit from masked diffusion?
- Taking masking into consideration makes the diffusion process more complicated by design. Therefore, one should show that these extra modeling complexity is worthwhile by showing more promising performance on various downstream applications. The marginal performance gain on the GEOM-QM9 dataset (not GEOM-Drugs) is not convincing enough.

**Questions:**

- From Table 5 in Appendix, the proposed method is inferior to GeoDiff across all evaluation metrics. Minor performance gain on GEOM-QM9 may not be that significant since the target molecules are small in general. That being said, MaskedDiff cannot outperform SOTA models while introducing extra complication during modeling.

- Diffusion models are known for their slow sampling process. Here the large number of steps (i.e., 5000) would make the sampling process particularly inefficient. For instance, for protein conformation generation, the number of steps is typically 200 or 500. Upon reducing the number of diffusion steps, would the alternating update scheme still work?

- Does other sampling techniques, e.g., probability flow ODE, work for the proposed method?

- Incorporating a well-designed prior distribution could also reduce the challenging learning task, e.g., [EigenFold](https://arxiv.org/abs/2304.02198). The model should be able to figure out the correlation between data dimensions during training. Would this be more effective and easier to learn than introducing masks for the diffusion process?

- Minor: Conclusion, "European space", fix typo.

---

> ### Author Response · Authors · 2023-11-20
> **Reply to Reviewer eamf (1/5)**
>
> We thank the valuable comments from the reviewer! We provide point-wise responses below.
>
> > **Q1**：The number of diffusion steps is the same as that in GeoDiff, i.e., 5000 steps, which is quite a lot. Given this large amount of diffusion steps, the alternating update scheme might work well. However, I am not sure if using more cost-effective sampling methods would still benefit from masked diffusion.
>
>
> **A1:** Thanks for your insightful comments. We conduct experiments with fewer sampling steps (500,200) to evaluate our method and the results are shown in Table 1 below. The findings suggest that: **1)** With 500 steps, our method can outperform existing methods like ConfGF, which uses 5000 sampling steps; **2)** Our method can consistently outperform GeoDiff for both DDPM and LD sampling methods, particularly when the number of diffusion steps is small. This indicates that different sampling methods can benefit from our masked diffusion.
>
> **Table 1: Results on the {GEOM-QM9} dataset. The threshold $\delta=0.5A$**
> | Steps  |  Models   | Sampling methods|  COV-R (%)  Mean $\uparrow$  |  COV-R (%) Median $\uparrow$ | MAT-R (A)  Mean $\downarrow$   | MAT-R (A) Median  $\downarrow$ | COV-P (%)  Mean$\uparrow$   |  COV-P (%)  Median$\uparrow$ | MAT-P (A) Mean $\downarrow$    | MAT-P (A) Median $\downarrow$  |
> |-----------|-------|--------|-------|--------|--------|---------|-------|--------|--------|---------|
> |5000|  ConfGF | Langevin dynamics| 88.49 | 94.31 | 0.2673 | 0.2685 | 46.43 | 43.41 | 0.5224 | 0.5124
> |
> |500|GeoDiff |DDPM|80.20   |   83.59| 0.3617| 0.3412 | 45.49    |  45.45 | 1.1518  |0.5087
> |500|Ours|DDPM|89.78$\uparrow_{9.58}$  |    94.17$\uparrow_{10.58}$ |  0.2417$\uparrow_{0.12}$| 0.2449$\uparrow_{0.0963}$|50.03$\uparrow_{4.54}$    |  48.31$\uparrow_{2.86}$| 0.5571$\uparrow_{0.5947}$|0.4921$\uparrow_{0.0166}$|
> 200| GeoDiff| DDPM  |      69.90    |   72.04| 0.4222 | 0.4272 | 36.71 |     33.51 | 0.8532 | 0.5554
> 200|  Ours |  DDPM |    85.53$\uparrow_{15.63}$   |    88.99$\uparrow_{16.95}$ |  0.2994$\uparrow_{0.1228}$ | 0.3033$\uparrow_{0.1239}$ | 47.76$\uparrow_{11.05}$  |    45.89$\uparrow_{12.38}$  | 0.6971$\uparrow_{0.156}$ | 0.5118$\uparrow_{0.0436}$
> |
> |500|GeoDiff |Langevin dynamics|87.80 |     93.66 | 0.3179| 0.3216 | 46.25    |  45.02 | 0.6173  |0.5112
> |500|**Ours**|Langevin dynamics|91.40$\uparrow_{3.6}$    |  95.39$\uparrow_{1.73}$ |   0.2543$\uparrow_{0.0636}$ | 0.2601$\uparrow_{0.0615}$|51.71$\uparrow_{5.46}$  |   48.50$\uparrow_{3.48}$| 0.5035$\uparrow_{0.1138}$ | 0.4734$\uparrow_{0.0378}$|
> |200| GeoDiff|Langevin dynamics |       86.60   |   93.09 | 0.3532 | 0.3574 | 42.98     |  42.60 | 0.5563 | 0.5367
> 200| ours |Langevin dynamics |       90.36$\uparrow_{3.76}$   |   95.93$\uparrow_{2.84}$ |  0.3064$\uparrow_{0.0468}$ | 0.3098$\uparrow_{0.0476}$ | 48.56$\uparrow_{5.580}$  |    46.46$\uparrow_{3.86}$  | 0.5540$\uparrow_{0.0023}$ | 0.5082$\uparrow_{0.0285}$
>
>
> **Reference:**
>
> *[1] Yang Song and Stefano Ermon. Generative modeling by estimating gradients of the data distribution. In Advances in Neural Information Processing Systems, pp. 11918–11930, 2019.*
>
> ---
> >**Q2**: Taking masking into consideration makes the diffusion process more complicated by design. Therefore, one should show that this extra modeling complexity is worthwhile by showing more promising performance on various downstream applications. The marginal performance gain on the GEOM-QM9 dataset (not GEOM-Drugs) is not convincing enough.
>
> **A2:** Thanks a lot for the helpful advice. We conduct additional experiments on self-supervised learning tasks, evaluating our approach to various downstream tasks. These tasks include 2D molecular property prediction (MoleculeNet) and 3D molecular property predictions (QM9 and MD17). The results are presented in Tables 2, 3, and 4 below.
> In the experiments, we incorporate the score network used in MoleculeSDE[1] to assess the effectiveness of SubGDiff. The results indicate that our methods surpass the baseline models in the realm of self-supervised learning. This implies that our technique can serve as a valuable tool for enhancing pretrained models in capturing substructure information during training.
>
> Beyond representation learning, we highlight that our method exhibits superior sampling efficiency for conformation generation (refer to Table 1 above).
>
> **Reference:**
>
> *[1] Liu S, Du W, Ma Z M, et al. A group symmetric stochastic differential equation model for molecule multi-modal pretraining//International Conference on Machine Learning. PMLR, 2023: 21497-21526.*

---

> ### Author Response · Authors · 2023-11-20
> **Reply to Reviewer eamf (2/5)**
>
> **Table 2: Results for molecular property prediction tasks (with 2D topology only)**
> | Pre-training      | BBBP $\uparrow$ | Tox21 $\uparrow$ | ToxCast $\uparrow$ | Sider $\uparrow$ | ClinTox $\uparrow$ | MUV $\uparrow$ | HIV $\uparrow$ | Bace $\uparrow$ | Avg $\uparrow$ |
> |-------------------|-----------------|------------------|--------------------|------------------|--------------------|----------------|----------------|-----------------|----------------|
> | -- (random init)  | 68.1 $\pm$ 0.59   | 75.3 $\pm$ 0.22    | 62.1 $\pm$ 0.19      | 57.0 $\pm$ 1.33    | 83.7 $\pm$ 2.93      | 74.6 $\pm$ 2.35  | 75.2 $\pm$ 0.70  | 76.7  $\pm$  2.51   | 71.60          |
> | AttrMask          | 65.0 $\pm$ 2.36   | 74.8 $\pm$ 0.25    | 62.9 $\pm$ 0.11      | 61.2 $\pm$ 0.12    | 87.7 $\pm$ 1.19      | 73.4 $\pm$ 2.02  | 76.8 $\pm$ 0.53  | 79.7 $\pm$ 0.33   | 72.68          |
> | ContextPred       | 65.7 $\pm$ 0.62   | 74.2 $\pm$ 0.06    | 62.5 $\pm$ 0.31      | 62.2 $\pm$ 0.59    | 77.2 $\pm$ 0.88      | 75.3 $\pm$ 1.57  | 77.1 $\pm$ 0.86  | 76.0 $\pm$ 2.08   | 71.28          |
> | InfoGraph         | 67.5 $\pm$ 0.11   | 73.2 $\pm$ 0.43    | 63.7 $\pm$ 0.50      | 59.9 $\pm$ 0.30    | 76.5 $\pm$ 1.07      | 74.1 $\pm$ 0.74  | 75.1 $\pm$ 0.99  | 77.8 $\pm$ 0.88   | 70.96          |
> | MolCLR            | 66.6 $\pm$ 1.89   | 73.0 $\pm$ 0.16    | 62.9 $\pm$ 0.38      | 57.5 $\pm$ 1.77    | 86.1 $\pm$ 0.95      | 72.5 $\pm$ 2.38  | 76.2 $\pm$ 1.51  | 71.5 $\pm$ 3.17   | 70.79          |
> | 3D InfoMax        | 68.3 $\pm$ 1.12   | 76.1 $\pm$ 0.18    | 64.8 $\pm$ 0.25      | 60.6 $\pm$ 0.78    | 79.9 $\pm$ 3.49      | 74.4 $\pm$ 2.45  | 75.9 $\pm$ 0.59  | 79.7 $\pm$ 1.54   | 72.47          |
> | GraphMVP          | 69.4 $\pm$ 0.21   | 76.2 $\pm$ 0.38    | 64.5 $\pm$ 0.20      | 60.5 $\pm$ 0.25    | 86.5 $\pm$ 1.70      | 76.2 $\pm$ 2.28  | 76.2 $\pm$ 0.81  | 79.8 $\pm$ 0.74   | 73.66          |
> | MoleculeSDE(VE)  | 68.3 $\pm$ 0.25   | 76.9 $\pm$ 0.23    | 64.7 $\pm$ 0.06      | 60.2 $\pm$ 0.29    | 80.8 $\pm$ 2.53      | 76.8 $\pm$ 1.71  | 77.0 $\pm$ 1.68  | 79.9 $\pm$ 1.76   | 73.15          |
> | MoleculeSDE(VP)   | 70.1 $\pm$ 1.35   | 77.0 $\pm$ 0.12    | 64.0 $\pm$ 0.07      | 60.8 $\pm$ 1.04    | 82.6 $\pm$ 3.64      | 76.6 $\pm$ 3.25  | 77.3 $\pm$ 1.31  | 81.4 $\pm$ 0.66   | 73.73          |
> |  **Ours**          | **70.2 $\pm$ 2.23**   | **77.2 $\pm$ 0.39**    | **65.0 $\pm$ 0.48**     | **62.2 $\pm$ 0.974**    | **88.2 $\pm$ 1.57**      | **77.3 $\pm$ 1.17**  | **77.6 $\pm$ 0.5**1  | **82.1 $\pm$ 0.96**  | **74.85**
>
> **Table 3: Results on 12 quantum mechanics prediction tasks from QM9. We take 110K for training, 10K for validation, and 11K for testing. The evaluation is mean absolute error~(MAE), and the best results are marked in bold.  The backbone is SchNet.**
> | Pretraining | Alpha ↓ | Gap ↓ | HOMO ↓ | LUMO ↓ | Mu ↓ | Cv ↓ | G298 ↓ | H298 ↓ | R2 ↓ | U298 ↓ | U0 ↓ | Zpve ↓ |
> |-------------|---------|-------|--------|--------|------|------|--------|--------|------|--------|------|---------|
> | Random init | 0.070 | 50.59 | 32.53 | 26.33 | 0.029 | 0.032 | 14.68 | 14.85 | 0.122 | 14.70 | 14.44 | 1.698 |
> | Supervised | 0.070 | 51.34 | 32.62 | 27.61 | 0.030 | 0.032 | 14.08 | 14.09 | 0.141 | 14.13 | 13.25 | 1.727 |
> | Type Prediction | 0.084 | 56.07 | 34.55 | 30.65 | 0.040 | 0.034 | 18.79 | 19.39 | 0.201 | 19.29 | 18.86 | 2.001 |
> | Angle Prediction | 0.084 | 57.01 | 37.51 | 30.92 | 0.037 | 0.034 | 15.81 | 15.89 | 0.149 | 16.41 | 15.76 | 1.850 |
> | 3D InfoGraph | 0.076 | 53.33 | 33.92 | 28.55 | 0.030 | 0.032 | 15.97 | 16.28 | 0.117 | 16.17 | 15.96 | 1.666 |
> | GeossL-RR | 0.073 | 52.57 | 34.44 | 28.41 | 0.033 | 0.038 | 15.74 | 16.11 | 0.194 | 15.58 | 14.76 | 1.804 |
> | GeossL-InfoNCE | 0.075 | 53.00 | 34.29 | 27.03 | 0.029 | 0.033 | 15.67 | 15.53 | 0.125 | 15.79 | 14.94 | 1.675 |
> | GeossL-EBM-NCE | 0.073 | 52.86 | 33.74 | 28.07 | 0.031 | 0.032 | 14.02 | 13.65 | 0.121 | 13.70 | 13.45 | 1.677 |
> | MoleculeSDE | 0.062 | 47.74 | 28.02 | 24.60 | 0.028 | 0.029 | 13.25 | 12.70 | 0.120 | 12.68 | 12.93 | 1.643 |
> | **Ours** | **0.054** | **44.88** | **25.45** | **23.75** | **0.027** | **0.028** | **12.03** | **11.46** | **0.110** | **11.32** | **11.25** | **1.568** |

---

> ### Author Response · Authors · 2023-11-20
> **Reply to Reviewer eamf (3/5)**
>
> **Table 4: Results on eight force prediction tasks from MD17. We take 1K for training, 1K for validation, and 48K to 991K molecules for the test concerning different tasks. The evaluation is mean absolute error, and the best results are marked in bold.**
> | Pretraining | Aspirin ↓ | Benzene ↓ | Ethanol ↓ | Malonaldehyde ↓ | Naphthalene ↓ | Salicylic ↓ | Toluene ↓ | Uracil ↓ |
> |-------------|-----------|-----------|------------|------------|----------------|-------------|------------|-----------|
> | (random init) | 1.203 | 0.380 | 0.386 | 0.794 | 0.587 | 0.826 | 0.568 | 0.773 |
> | Type Prediction | 1.383 | 0.402 | 0.450 | 0.879 | 0.622 | 1.028 | 0.662 | 0.840 |
> | Distance Prediction | 1.427 | 0.396 | 0.434 | 0.818 | 0.793 | 0.952 | 0.509 | 1.567 |
> | Angle Prediction | 1.542 | 0.447 | 0.669 | 1.022 | 0.680 | 1.032 | 0.623 | 0.768 |
> | 3D InfoGraph | 1.610 | 0.415 | 0.560 | 0.900 | 0.788 | 1.278 | 0.768 | 1.110 |
> | RR | 1.215 | 0.393 | 0.514 | 1.092 | 0.596 | 0.847 | 0.570 | 0.711 |
> | InfoNCE | 1.132 | 0.395 | 0.466 | 0.888 | 0.542 | 0.831 | 0.554 | 0.664 |
> | EBM-NCE | 1.251 | 0.373 | 0.457 | 0.829 | 0.512 | 0.990 | 0.560 | 0.742 |
> | 3D InfoMax | 1.142 | 0.388 | 0.469 | 0.731 | 0.785 | 0.798 | 0.516 | 0.640 |
> | GraphMVP | 1.126 | 0.377 | 0.430 | 0.726 | 0.498 | 0.740 | 0.508 | 0.620 |
> | GeoSSL-1L | 1.364 | 0.391 | 0.432 | 0.830 | 0.599 | 0.817 | 0.628 | 0.607 |
> | GeoSSL | 1.107 | 0.360 | 0.357 | 0.737 | 0.568 | 0.902 | 0.484 | 0.502 |
> | MoleculeSDE (VE) | 1.112 | 0.304 | 0.282 | 0.520| 0.455 | 0.725 | 0.515 | 0.447 |
> | MoleculeSDE (VP) | 1.244 | 0.315 | 0.338 | 0.488 | 0.432 | 0.712 | 0.478 | 0.468 |
> | **Ours** | **0.880** | **0.252** | 0.398 | 0.556 | **0.325** | **0.572** | **0.362** | **0.420** |
>
>
> ---
> >**Q3**: From Table 5 in the Appendix, the proposed method is inferior to GeoDiff across all evaluation metrics. Minor performance gain on GEOM-QM9 may not be that significant since the target molecules are small in general. That being said, MaskedDiff cannot outperform SOTA models while introducing extra complications during modeling.
> >
> **A3:** Thanks for your careful reviews! It is important to clarify that our method isn't inferior to GeoDiff across all evaluation metrics. We should consider different sampling methods when making comparisons with GeoDiff from Table 5 in the Appendix. **1)** Our method outperforms GeoDiff among all the Recall metrics when we use DDPM sampling(ancestral sampling) methods, demonstrating that our method can effectively improve the diversity of the generative samples compared to GeoDiff. The results are shown in Table 5 below, where the values are obtained by evaluating from the released [checkpoint](https://github.com/MinkaiXu/GeoDiff/tree/main). **2)** For a more comprehensive comparison, we include results using 500 diffusion steps. The findings suggest that our method significantly outperforms GeoDiff when reducing the number of diffusion steps, indicating an improvement in sampling efficiency.   On the other hand, **3)** it is true that our methods cannot outperform GeoDiff when we use the same 5000-step Langevin dynamics(LD) sampling(Table 6). This is attributed to the LD sampling method's inability to consider mask prediction when using our model trained by SubGDiff, which could be a potential problem for generating relatively larger molecular graphs in the Drugs dataset with a large number of steps (e.g., 5000). This challenge can be alleviated when reducing the number of steps (e.g. 500), as demonstrated in Table 6.
>
> We will add more descriptions to avoid this misunderstanding. Thanks again for pointing it out!
>
> **Table 5: Results on the {GEOM-Drugs} dataset. The threshold $\delta=1.25A$ and the sampling method is DDPM(ancestral sampling)**
> |Steps| Models    |  Sampling methods|   CCOV-R (%)  Mean $\uparrow$  |  COV-R (%) Median $\uparrow$ | MAT-R (A)  Mean $\downarrow$   | MAT-R (A) Median  $\downarrow$ |
> |---|------|---------|-------|--------|--------|----|
> |5000| GeoDiff |  DDPM(ancestral sampling) |   56.31 |   57.73 |   1.2159 |   1.2053 |
> |5000| **Ours**   | DDPM(ours)  |   **87.86** |   **96.71** |   **0.9038** |   **0.8984** |
> |
> | 500  | GeoDiff    | DDPM   | 50.25  | 48.18  | 1.3101  | 1.2967  |
> | 500  | **Ours**     | DDPM (ours)      | **76.16**↑ | **86.43**↑ | **1.0463**↑ | **1.0264**↑ |
>
> **Table 6: Results on the {GEOM-Drugs} dataset. The threshold $\delta=1.25A$. Both models adopt the same Langevin dynamics sampling method.**
> | Steps | Models    |  sampling methods|  CCOV-R (%)  Mean $\uparrow$  |  COV-R (%) Median $\uparrow$ | MAT-R (A)  Mean $\downarrow$   | MAT-R (A) Median  $\downarrow$ |
> |---|--|-----|---|--------|----|-----|
> |5000| GeoDiff | Langevin dynamics | 89.13| 97.88 | 0.8629 | 0.8529
> |5000| Ours | Langevin dynamics |   87.67 |   97.30 |   0.8978 |   0.8938 |
> |
> | 500 | GeoDiff    | Langevin dynamics   | 64.12  | 75.56  | 1.1444  | 1.1246  |
> | 500| **Ours**     | Langevin dynamics  | **74.30**↑ | **77.87**↑ | **1.0003**↑ | **0.9905**↑ |

---

> ### Author Response · Authors · 2023-11-20
> **Reply to Reviewer eamf (4/5)**
>
> >**Q4**: Diffusion models are known for their slow sampling process. Here the large number of steps (i.e., 5000) would make the sampling process particularly inefficient. For instance, for protein conformation generation, the number of steps is typically 200 or 500. Upon reducing the number of diffusion steps, would the alternating update scheme still work?
>
> **A4:** Thanks for your valuable question. The proposed method still **demonstrates effectiveness** when the number of diffusion steps is reduced. Following your instruction, we performed experiments for a diffusion step sensitivity analysis, and the results can be found in Table 5 below. As observed in Table 7, our method significantly outperforms GeoDiff across all evaluation metrics when the number of steps is reduced to 500 and 200. Furthermore, our method even achieves competitive performance against 5000-step ConfGF, highlighting its ability to enhance the denoising efficiency of the denoising network through the introduction of the mask.
>
> **Table 7: Results on the {GEOM-QM9} dataset. The threshold $\delta=0.5A$**
> | Steps  |  Models   | Sampling methods|  COV-R (%)  Mean $\uparrow$  |  COV-R (%) Median $\uparrow$ | MAT-R (A)  Mean $\downarrow$   | MAT-R (A) Median  $\downarrow$ | COV-P (%)  Mean$\uparrow$   |  COV-P (%)  Median$\uparrow$ | MAT-P (A) Mean $\downarrow$    | MAT-P (A) Median $\downarrow$  |
> |-----------|-------|--------|-------|--------|--------|---------|-------|--------|--------|---------|
> |5000|  ConfGF | Langevin dynamics| 88.49 | 94.31 | 0.2673 | 0.2685 | 46.43 | 43.41 | 0.5224 | 0.5124
> |
> |500|GeoDiff |DDPM|80.20   |   83.59| 0.3617| 0.3412 | 45.49    |  45.45 | 1.1518  |0.5087
> |500|Ours|DDPM|89.78$\uparrow_{9.58}$  |    94.17$\uparrow_{10.58}$ |  0.2417$\uparrow_{0.12}$| 0.2449$\uparrow_{0.0963}$|50.03$\uparrow_{4.54}$    |  48.31$\uparrow_{2.86}$| 0.5571$\uparrow_{0.5947}$|0.4921$\uparrow_{0.0166}$|
> 200| GeoDiff| DDPM  |      69.90    |   72.04| 0.4222 | 0.4272 | 36.71 |     33.51 | 0.8532 | 0.5554
> 200|  Ours |  DDPM |    85.53$\uparrow_{15.63}$   |    88.99$\uparrow_{16.95}$ |  0.2994$\uparrow_{0.1228}$ | 0.3033$\uparrow_{0.1239}$ | 47.76$\uparrow_{11.05}$  |    45.89$\uparrow_{12.38}$  | 0.6971$\uparrow_{0.156}$ | 0.5118$\uparrow_{0.0436}$
> |
> |500|GeoDiff |Langevin dynamics|87.80 |     93.66 | 0.3179| 0.3216 | 46.25    |  45.02 | 0.6173  |0.5112
> |500|**Ours**|Langevin dynamics|91.40$\uparrow_{3.6}$    |  95.39$\uparrow_{1.73}$ |   0.2543$\uparrow_{0.0636}$ | 0.2601$\uparrow_{0.0615}$|51.71$\uparrow_{5.46}$  |   48.50$\uparrow_{3.48}$| 0.5035$\uparrow_{0.1138}$ | 0.4734$\uparrow_{0.0378}$|
> |200| GeoDiff|Langevin dynamics |       86.60   |   93.09 | 0.3532 | 0.3574 | 42.98     |  42.60 | 0.5563 | 0.5367
> 200| ours |Langevin dynamics |       90.36$\uparrow_{3.76}$   |   95.93$\uparrow_{2.84}$ |  0.3064$\uparrow_{0.0468}$ | 0.3098$\uparrow_{0.0476}$ | 48.56$\uparrow_{5.580}$  |    46.46$\uparrow_{3.86}$  | 0.5540$\uparrow_{0.0023}$ | 0.5082$\uparrow_{0.0285}$
>
> > **Q5**: Does other sampling techniques, e.g., probability flow ODE, work for the proposed method?
>
> **A5:** Thanks for your insightful question! **1)** Theoretically, ODE-based sampling methods cannot be directly applied to our approach. This is due to our method combining a discrete distribution (mask distribution $p(S)$) into the Gaussian distribution. If we transform the discrete Diffusion process to its corresponding continue version (i.e., increasing the number of noise scales $N\to \infty$, $\Delta t := 1/N \to 0$), the discrete mask distribution wouldn't be converted into a continuous function, resulting in that the Markov chain $\{R^i\}_{i=1}^{N}$ cannot converge to a continuous stochastic process[1] when $\Delta t := 1/N \to 0$. Therefore, obtaining ODE-based sampling methods from rigorous mathematical derivations is not possible. **2)**  In practice, it is possible to apply ODE-based sampling methods by exclusively utilizing the noise predictor for sampling and disregarding the mask predictor. This is feasible since the noise predictor can be well-trained among all nodes through uniform subgraph sampling.
>
>
> *[1] Song, Yang, et al. "Score-Based Generative Modeling through Stochastic Differential Equations." International Conference on Learning Representations. 2021.*

---

> ### Author Response · Authors · 2023-11-20
> **Reply to Reviewer eamf (5/5)**
>
> > **Q6**: Incorporating a well-designed prior distribution could also reduce the challenging learning task, e.g., EigenFold. The model should be able to figure out the correlation between data dimensions during training. Would this be more effective and easier to learn than introducing masks for the diffusion process?
>
> **A6:** Thanks for your valuable advice! **1)** A well-designed prior distribution is a potentially effective method to mitigate the independence of Gaussian noise injected into atomic coordinates and simultaneously enhance the score network's ability to perceive substructure. **2)** However, the harmonic prior provided in EigenFold cannot be directly used to tackle this learning task. Specifically, sampling from the harmonic prior restricts the diversity of the noisy conformations input to the score networks. Consequently, when using this diffusion method as a pretraining task, it may simplify the learning process for the generation task but fail to enhance the model's capability to perceive substructure. As a result, downstream tasks may not derive significant benefits from the pretraining. **3)** Although the harmonic prior contains graph structure information, it does not contribute to enhancing the denoising network $s_\theta(G,R^t,t)$ in capturing substructure. This is because the input $R^t$ already contains sufficient prior information (via adding noise) to help $s_\theta$ predict the noise related to the harmonic prior. In contrast, our method introduces masks, allowing the model to learn prior knowledge by predicting the mask.
>
> Thanks again for your useful insights, which help us a lot to improve our work.

---

> > ### Comment · Reviewer_eamf · 2023-11-21
> > **Response to author rebuttal**
> >
> > Dear authors,
> >
> > Thanks for your response and comprehensive additional experimental results. I went through other reviews and have updated my evaluation. I am still inclined towards weak rejection due to the following reasons: [1] With reduced sampling steps, model performance drops significantly (although better than GeoDiff in this case). [2] I resonate with other reviewers regarding the minor performance gain over SOTA model. [3] Lastly, the contribution of this work is not significant enough to be accepted by ICLR.

---

> > > ### Author Response · Authors · 2023-11-22
> > > **Further response to Reviewer eamf**
> > >
> > > > I resonate with other reviewers regarding the minor performance gain over SOTA model.
> > >
> > > We sincerely appreciate your efforts in reevaluating our work and improving the score. We have taken into consideration the comments provided by you and other reviewers. In our revised version, we have made significant enhancements (highlighted in blue and orange), particularly in the self-supervised learning task, where we now significantly outperform the state-of-the-art models. Thank you for your thoughtful review and valuable insights.

---

### Official Review · Reviewer_yz4M · 2023-11-01

**Soundness:** 2 fair
**Presentation:** 2 fair
**Contribution:** 2 fair
**Rating:** 3
**Confidence:** 5

**Summary:**

The paper studies 3D molecular graphs, and performs diffusion on a subgraph sampled by a learnable mask vector. Then the major work is to fulfill the training and sampling of the diffusion model based on the mask strategy. The developed methods are applied to both molecular generation tasks and self-supervised learning tasks.

**Strengths:**

A technical contribution in this paper could be - fulfilling the training and sampling of diffusion models on a subset of a graph with reasonably rigorous math. Even though the subgraph (I would call subset) in the paper is not convincing, the technical implementation would be at least inspirational.

**Weaknesses:**

1. My major concern is that, the overall motivation is not convincing or valid. The authors think diffusing individual atoms in a molecule may constrain the capacity of the models, as the connections among these atoms can be important. However, the methods tend to be a mask to choose a subset of atoms, which can be anywhere of the original graph and disconnected at all. As a comparison, there exist a bunch of studies [1,2] that consider REAL subgraphs (like motifs, functional groups etc) in generation by leveraging domain knowledge. The motivation of these existing works is more convincing and valid.

2. The title and scope of the paper are a little weird. The work is actually not regular 3D molecular representation learning, but for molecular generation and self-supervised based representation learning. I suggest the authors consider this in the revision.

3. Experimental results on GEOM-QM9 are very marginal, especially on the COV-P and MAT-P metrics where GeoDiff is much better. I also have concerns about the experiment setting that both generation and representation learning are conducted. However, for each one, the setup (like datasets)  is not sufficient or the results are not strong enough compared to baseline methods. A better solution is to only study one problem (like generation) with sufficient and strong results. This is also related to the title and scope of this work.

4. To me, the implementation of the training and sampling of diffusion models on a subset of a graph is not trivial, but there exist several studies for similar purposes, like [1,2,3]. The authors may want to clearly state the unique **technical** contribution compared with these studies.




Ref

[1] Haitao Lin et al, Functional-Group-Based Diffusion for Pocket-Specific Molecule Generation and Elaboration, NeurIPS 2023.

[2] Yangtian Zhang et al, DiffPack: A Torsional Diffusion Model for Autoregressive Protein Side-Chain Packing, NeurIPS 2023.

[3] Lingkai Kong, Autoregressive Diffusion Model for Graph Generation, ICML 2023.

**Questions:**

See Weakness. A major consideration during revision is a valid and convincing motivation.

---

> ### Author Response · Authors · 2023-11-20
> **Reply to Reviewer yz4M (1/4)**
>
> We thank the reviewer for the constructive comments. We provide pointwise responses below.
>
> >**Q1**. My major concern is that, the overall motivation is not convincing or valid. The authors think diffusing individual atoms in a molecule may constrain the capacity of the models, as the connections among these atoms can be important. However, the methods tend to be a mask to choose a subset of atoms, which can be anywhere of the original graph and disconnected at all. As a comparison, there exist a bunch of studies [1,2] that consider REAL subgraphs (like motifs, functional groups, etc) in generation by leveraging domain knowledge. The motivation of these existing works is more convincing and valid.
>
> **A1:** We sincerely thank the reviewer's comment!
> **1)** The reviewer rightly emphasizes the importance of connections among subgraph atoms, and we agree that randomly selecting a subset of atoms could result in disconnected subgraphs. Our proposed method is a versatile framework, allowing us to design specific mask distributions to ensure the connected property of the selected subgraph. We will provide detailed instructions on constraining the mask distribution when introducing maskedDiff.
> **2)** In our setting, we employ the torsional-based decomposition method (refer to sec 5.1 in the main text), in which the torsional edge in a molecule will be randomly selected and cut off to make the molecule into two components. This effectively guarantees the connectivity of the subgraphs.
> **3)** Using the REAl subgraphs proposed by [1,2] in our framework is also a potentially effective approach. One key difference between our work and [1,2] is that in our work the atoms of a subgraph are denoised in the same step, and no 3D information of the subgraph (3D local structure) is directly input to the model. Our model aims to learn these prior domain knowledge so as to enhance the capability of 3D representation. While in [1,2], the 3D structure of the REAL subgraphs is given to the model as a prior and their aim is to make the generation task easier.
>
>
> ---
> > **Q2**. The title and scope of the paper are a little weird. The work is actually not regular 3D molecular representation learning, but for molecular generation and self-supervised based representation learning. I suggest the authors consider this in the revision.
> >
> **A2:** Thanks for the suggestion. One of our motivations is that a generation task can be used for self-supervised representation learning. To better incorporate the 3D information, we choose the 3D molecular conformation generation as our representation learning task and introduce subgraphs. 3D molecular representation learning is our focus. We will try to make this clearer in the paper and maybe use a more appropriate title.

---

> ### Author Response · Authors · 2023-11-20
> **Reply to Reviewer yz4M (2/4)**
>
> > **Q3**. Experimental results on GEOM-0M9 are very marginal, especially on the Co-P and MATP metrics where GeoDifis much better.I also have concerns about the experiment setting that both generation and representation learning are conducted. However, for each one, the setup (like datasets) is not sufficient or the results are not strong enough compared to baseline methods. A better solution is to only study one problem (like generation) with sufficient and strong results. This is also related to the title and scope of this work.
>
> **A3:** Thank you for your valuable suggestions. **1)** We have updated the results for conformation generation in the revised version. **2)** Following your advice, we've shifted our focus to self-supervised representation learning here. Additional experiments were conducted, and the results are presented in Tables 1, 2, and 3 below.
> In these experiments, we assessed the efficacy of SubGDiff by incorporating the score network from MoleculeSDE[1], a pretraining framework. The results demonstrate that our method outperforms baseline models in the realm of self-supervised learning. This underscores the utility of our technique in enhancing pre-trained models to capture substructure information during the pre-training process.
>
>
> *[1] Liu S, Du W, Ma Z M, et al. A group symmetric stochastic differential equation model for molecule multi-modal pretraining[C]//International Conference on Machine Learning. PMLR, 2023: 21497-21526.*
>
>
>
> **Table 1: Results for molecular property prediction tasks (with 2D topology only)**
> | Pre-training      | BBBP $\uparrow$ | Tox21 $\uparrow$ | ToxCast $\uparrow$ | Sider $\uparrow$ | ClinTox $\uparrow$ | MUV $\uparrow$ | HIV $\uparrow$ | Bace $\uparrow$ | Avg $\uparrow$ |
> |-------------------|-----------------|------------------|--------------------|------------------|--------------------|----------------|----------------|-----------------|----------------|
> | -- (random init)  | 68.1 $\pm$ 0.59   | 75.3 $\pm$ 0.22    | 62.1 $\pm$ 0.19      | 57.0 $\pm$ 1.33    | 83.7 $\pm$ 2.93      | 74.6 $\pm$ 2.35  | 75.2 $\pm$ 0.70  | 76.7  $\pm$  2.51   | 71.60          |
> | AttrMask          | 65.0 $\pm$ 2.36   | 74.8 $\pm$ 0.25    | 62.9 $\pm$ 0.11      | 61.2 $\pm$ 0.12    | 87.7 $\pm$ 1.19      | 73.4 $\pm$ 2.02  | 76.8 $\pm$ 0.53  | 79.7 $\pm$ 0.33   | 72.68          |
> | ContextPred       | 65.7 $\pm$ 0.62   | 74.2 $\pm$ 0.06    | 62.5 $\pm$ 0.31      | 62.2 $\pm$ 0.59    | 77.2 $\pm$ 0.88      | 75.3 $\pm$ 1.57  | 77.1 $\pm$ 0.86  | 76.0 $\pm$ 2.08   | 71.28          |
> | InfoGraph         | 67.5 $\pm$ 0.11   | 73.2 $\pm$ 0.43    | 63.7 $\pm$ 0.50      | 59.9 $\pm$ 0.30    | 76.5 $\pm$ 1.07      | 74.1 $\pm$ 0.74  | 75.1 $\pm$ 0.99  | 77.8 $\pm$ 0.88   | 70.96          |
> | MolCLR            | 66.6 $\pm$ 1.89   | 73.0 $\pm$ 0.16    | 62.9 $\pm$ 0.38      | 57.5 $\pm$ 1.77    | 86.1 $\pm$ 0.95      | 72.5 $\pm$ 2.38  | 76.2 $\pm$ 1.51  | 71.5 $\pm$ 3.17   | 70.79          |
> | 3D InfoMax        | 68.3 $\pm$ 1.12   | 76.1 $\pm$ 0.18    | 64.8 $\pm$ 0.25      | 60.6 $\pm$ 0.78    | 79.9 $\pm$ 3.49      | 74.4 $\pm$ 2.45  | 75.9 $\pm$ 0.59  | 79.7 $\pm$ 1.54   | 72.47          |
> | GraphMVP          | 69.4 $\pm$ 0.21   | 76.2 $\pm$ 0.38    | 64.5 $\pm$ 0.20      | 60.5 $\pm$ 0.25    | 86.5 $\pm$ 1.70      | 76.2 $\pm$ 2.28  | 76.2 $\pm$ 0.81  | 79.8 $\pm$ 0.74   | 73.66          |
> | MoleculeSDE(VE)  | 68.3 $\pm$ 0.25   | 76.9 $\pm$ 0.23    | 64.7 $\pm$ 0.06      | 60.2 $\pm$ 0.29    | 80.8 $\pm$ 2.53      | 76.8 $\pm$ 1.71  | 77.0 $\pm$ 1.68  | 79.9 $\pm$ 1.76   | 73.15          |
> | MoleculeSDE(VP)   | 70.1 $\pm$ 1.35   | 77.0 $\pm$ 0.12    | 64.0 $\pm$ 0.07      | 60.8 $\pm$ 1.04    | 82.6 $\pm$ 3.64      | 76.6 $\pm$ 3.25  | 77.3 $\pm$ 1.31  | 81.4 $\pm$ 0.66   | 73.73          |
> |  **Ours**          | **70.2 $\pm$ 2.23**   | **77.2 $\pm$ 0.39**    | **65.0 $\pm$ 0.48**     | **62.2 $\pm$ 0.974**   | **88.2 $\pm$ 1.57**      | **77.3 $\pm$ 1.17**  | **77.6 $\pm$ 0.5**1  | **82.1 $\pm$ 0.96**  | **74.85**

---

> ### Author Response · Authors · 2023-11-20
> **Reply to Reviewer yz4M (3/4)**
>
> **Table 2: Results on 12 quantum mechanics prediction tasks from QM9. We take 110K for training, 10K for validation, and 11K for testing. The evaluation is mean absolute error~(MAE), and the best results are marked in bold.  The backbone is SchNet.**
> | Pretraining | Alpha ↓ | Gap ↓ | HOMO ↓ | LUMO ↓ | Mu ↓ | Cv ↓ | G298 ↓ | H298 ↓ | R2 ↓ | U298 ↓ | U0 ↓ | Zpve ↓ |
> |-------------|---------|-------|--------|--------|------|------|--------|--------|------|--------|------|---------|
> | Random init | 0.070 | 50.59 | 32.53 | 26.33 | 0.029 | 0.032 | 14.68 | 14.85 | 0.122 | 14.70 | 14.44 | 1.698 |
> | Supervised | 0.070 | 51.34 | 32.62 | 27.61 | 0.030 | 0.032 | 14.08 | 14.09 | 0.141 | 14.13 | 13.25 | 1.727 |
> | Type Prediction | 0.084 | 56.07 | 34.55 | 30.65 | 0.040 | 0.034 | 18.79 | 19.39 | 0.201 | 19.29 | 18.86 | 2.001 |
> | Angle Prediction | 0.084 | 57.01 | 37.51 | 30.92 | 0.037 | 0.034 | 15.81 | 15.89 | 0.149 | 16.41 | 15.76 | 1.850 |
> | 3D InfoGraph | 0.076 | 53.33 | 33.92 | 28.55 | 0.030 | 0.032 | 15.97 | 16.28 | 0.117 | 16.17 | 15.96 | 1.666 |
> | GeossL-RR | 0.073 | 52.57 | 34.44 | 28.41 | 0.033 | 0.038 | 15.74 | 16.11 | 0.194 | 15.58 | 14.76 | 1.804 |
> | GeossL-InfoNCE | 0.075 | 53.00 | 34.29 | 27.03 | 0.029 | 0.033 | 15.67 | 15.53 | 0.125 | 15.79 | 14.94 | 1.675 |
> | GeossL-EBM-NCE | 0.073 | 52.86 | 33.74 | 28.07 | 0.031 | 0.032 | 14.02 | 13.65 | 0.121 | 13.70 | 13.45 | 1.677 |
> | MoleculeSDE | 0.062 | 47.74 | 28.02 | 24.60 | 0.028 | 0.029 | 13.25 | 12.70 | 0.120 | 12.68 | 12.93 | 1.643 |
> | **Ours** | **0.054** | **44.88** | **25.45** | **23.75** | **0.027** | **0.028** | **12.03** | **11.46** | **0.110** | **11.32** | **11.25** | **1.568** |
>
> **Table 3: Results on eight force prediction tasks from MD17. We take 1K for training, 1K for validation, and 48K to 991K molecules for the test concerning different tasks. The evaluation is mean absolute error, and the best results are marked in bold.**
> | Pretraining | Aspirin ↓ | Benzene ↓ | Ethanol ↓ | Malonaldehyde ↓ | Naphthalene ↓ | Salicylic ↓ | Toluene ↓ | Uracil ↓ |
> |-------------|-----------|-----------|------------|-----------------|----------------|--------------|------------|-----------|
> | (random init) | 1.203 | 0.380 | 0.386 | 0.794 | 0.587 | 0.826 | 0.568 | 0.773 |
> | Type Prediction | 1.383 | 0.402 | 0.450 | 0.879 | 0.622 | 1.028 | 0.662 | 0.840 |
> | Distance Prediction | 1.427 | 0.396 | 0.434 | 0.818 | 0.793 | 0.952 | 0.509 | 1.567 |
> | Angle Prediction | 1.542 | 0.447 | 0.669 | 1.022 | 0.680 | 1.032 | 0.623 | 0.768 |
> | 3D InfoGraph | 1.610 | 0.415 | 0.560 | 0.900 | 0.788 | 1.278 | 0.768 | 1.110 |
> | RR | 1.215 | 0.393 | 0.514 | 1.092 | 0.596 | 0.847 | 0.570 | 0.711 |
> | InfoNCE | 1.132 | 0.395 | 0.466 | 0.888 | 0.542 | 0.831 | 0.554 | 0.664 |
> | EBM-NCE | 1.251 | 0.373 | 0.457 | 0.829 | 0.512 | 0.990 | 0.560 | 0.742 |
> | 3D InfoMax | 1.142 | 0.388 | 0.469 | 0.731 | 0.785 | 0.798 | 0.516 | 0.640 |
> | GraphMVP | 1.126 | 0.377 | 0.430 | 0.726 | 0.498 | 0.740 | 0.508 | 0.620 |
> | GeoSSL-1L | 1.364 | 0.391 | 0.432 | 0.830 | 0.599 | 0.817 | 0.628 | 0.607 |
> | GeoSSL | 1.107 | 0.360 | 0.357 | 0.737 | 0.568 | 0.902 | 0.484 | 0.502 |
> | MoleculeSDE (VE) | 1.112 | 0.304 | 0.282 | 0.520| 0.455 | 0.725 | 0.515 | 0.447 |
> | MoleculeSDE (VP) | 1.244 | 0.315 | 0.338 | 0.488 | 0.432 | 0.712 | 0.478 | 0.468 |
> | **Ours** | **0.880** | **0.252** | 0.398 | 0.556 | **0.325** | **0.572** | **0.362** | **0.420** |

---

> ### Author Response · Authors · 2023-11-20
> **Reply to Reviewer yz4M (4/4)**
>
> > **Q4**: To me, the implementation of the training and sampling of diffusion models on a subset of a graph is not trivial, but there exist several studies for similar purposes, like [1,2,3]. The authors may want to clearly state the unique technical contribution compared with these studies.
>
> **A4:** Thanks a lot for suggesting these studies. We highlight the differences between our method with [1,2,3] as follows.
>  - Compare to [1]: D3FG[1] adopts three different diffusion models (D3PM,DDPM,and SO(3) Diffusion) to generate three different components of molecules(linkerr types, center atom position, and functional group orientations), respectively. In general, these three components can also be considered as three subgraphs(subset). Essentially, D3FG firstly selects the subgraph and only injects noise into the fixed subgraph during the *entire* diffusion process, **while** our method has the flexibility to select different subgraphs from a mask distribution at *each time step* during the forward process. Further, our model is actually a single diffusion model that can enhance the denoising network to perceive substructure information by incorporating a mask variable in it.
>
>  - Compare to [2]: DIffPAck[2] is an Autoregressive generative method predicting the torsional angle $\chi_i (i=1,2,..,4)$ of protein side-chains conditioned on $\chi_{1,...,i-1}$, where $\chi_i$ is a predefined subset of atoms. It uses a torsional-based diffusion model to approximate the distribution $p(\chi_i| \chi_{1,...,i-1})$. Each $\chi_i$ requires a separate score network for estimation.  Essentially, it can be viewed as selecting a subset initially and then adding noise exclusively to the fixed subset throughout the *entire* diffusion process.  **In contrast**, our method takes a different approach by randomly sampling a subset from a mask distribution $p(S)$ at each time step during the forward process. This approach offers greater flexibility and cost-effectiveness, requiring only a score network and a subgraph predictor.
>
>  - Compare to [3]: [3] introduces an autoregressive diffusion model GraphARM, which absorbs *one node* in each time-step by masking it along with its connecting edges during the forward process.   **Unlike GraphARM[3]**, instead of masking individual nodes, selects a *subgraph* in each time-step to inject the Gaussian noise. This is equivalent to masking several nodes during the forward process. Moreover, GraphARM mandates that the number of steps must be the same as the number of nodes due to the usage of the absorbing state. **In contrast**, our method can set any number of diffusion steps theoretically since we use the real-value Gaussian noise.
>
>
> We will add those comparisons to our revised version. Thanks for your constructive suggestions.
>
> **References:**
>
> [1] Haitao Lin et al, Functional-Group-Based Diffusion for Pocket-Specific Molecule Generation and Elaboration, NeurIPS 2023.
>
> [2] Yangtian Zhang et al, DiffPack: A Torsional Diffusion Model for Autoregressive Protein Side-Chain Packing,
> NeurIPS 2023.
>
> [3] Lingkai Kong, Autoregressive Diffusion Model for Graph Generation, ICML 2023.

---

> ### Comment · Reviewer_yz4M · 2023-11-21
>
> I thank the authors for their efforts in adding more experiments on SSL. However, the motivation of diffusing a subset is still not convincing to me; the scope of this work has not been changed; and empirical studies on generation are still weak. Given these considerations, I feel a significant revision is needed so that the paper can be resubmitted to another venue. I  will maintain my score.

---

> ### Author Response · Authors · 2023-11-22
> **Further response to Reviewer yz4M (1/2)**
>
> Thank you so much for your response! We would like to provide the following clarification.
> >Summary:
> The paper studies 3D molecular graphs, and performs diffusion on a subgraph sampled by a **learnable mask vector**.  Then the major work is to fulfill the training and sampling of the diffusion model based on the mask strategy.
>
> **Reponse:**  We clarify that our method does not adpot a learnable mask vector during the diffusion process. Actually, a mask vector corresponds to a subgraph, and the mask vector is sampled from a predefined mask distribution. Thus our selected subgraph is actually **a REAL subgraph**.
> In our method and experiments, we pre-define the mask distribution to be a discrete distribution with a sample space $\chi =\\{G ^i_{sub}\\} _{i=1}^{N} $, and $p _{s _t} (\mathcal{S} = G _{sub} ^i )$ , $ = 1/N,t>1 $, where $G ^i _{sub}$ is the subgraph split by the torsional-based decomposition methods [1]. The decomposition approach will cut off one torsional edge in a 3D molecule to make the molecule into two **connected components**, each containing at least two atoms. The torisional edge here acutally corresponds to the Bridge(or Cut Edge) in Graph Theory. The two components are represented as two complementary mask vectors  (i.e. $\mathbf{s}'+\mathbf{s}=\mathbf{1}$). Thus $n$ torsional edges in $G _{3D}$ will generate $2n$ subgraphs. Finally, for each atom $v$, the $s _{t_v} \sim \text{Bern}(0.5)$, ensuring that every atom can be added noise.
>
> We have revised our PDF (highlighted in orange) for clarification.
>
> **Reference**:
>
> *[1] Jing B, Corso G, Chang J, et al. Torsional diffusion for molecular conformer generation[J]. Advances in Neural Information Processing Systems, 2022, 35: 24240-24253.*
>
> ---
>
> > **Further comment 1**: The motivation of diffusing a subset is still not convincing to me.
>
> **Response 1**: The motivation for using subgraph (subset) sampling during diffusion is to enhance the perception of 3D molecular structures in the denoising network. The subgraph (subset) is predefined, not a random subset. The mask distribution is equivalent to the subgraph distribution, defined over a predefined sample space $\chi =\\{G ^i_{sub}\\} _{i=1}^{N}$, where each sample is a connected subgraph extracted from $G$. The mask predictor $s _{\vartheta}$ learns the substructure information by predicting the mask, which serve as an additonal task to denoising. Thus, the denoising network $\epsilon _\theta$ can improve its ability to capture the substructure through sharing the GNN encoder with the mask predictor $s _{\vartheta}$.
>
> The whole training process can be summarized in the following algorithm:
>
>  for $i$ in $\{1,2,...,K\}$; do
>
> 1. Sample $G_{3D} \sim q(\mathrm X)$ // Sample a 3D molecule graph
> 2. Sample $t \sim \mathcal{U}(1, ..., T)$, $\epsilon \sim \mathcal{N}(\mathbf{0}, \mathbf{I}))$
> 3. Create sample space $\chi = \\{ G _{sub} ^i \\} _{i=1} ^{N}$;
>  $G ^i _{sub} \in $ Torsional_decomposition$(G _{3D})$ / / For the mask distribution $p _{s _t}(\mathcal{S})$
> 4. Sample $\mathbf{s}^t \sim p_{s_t}(\mathcal{S})$ //Sample a mask vector (subgraph)
> 5. $R^t \gets q(R^t | R^{0})$  // Eq. 14
> 6. $\mathcal{L} = \\|\text{diag}(\mathbf{s} _t) (\epsilon-\epsilon _{\theta}(\mathcal{G}, R^t, t))\\|^2 +\lambda \mathrm{BCE}(\mathbf{s} _t, s _\vartheta(\mathcal{G}, R^t, t))$
> 7. $\text{optimizer.step}(\mathcal{L})$
>
> end

---

> > ### Author Response · Authors · 2023-11-22
> > **Further response to Reviewer yz4M (2/2)**
> >
> > >**Further comment 2:** Empirical studies on generation are still weak.
> >
> > **Response 2:** Self-supervised representation learning is our main focused task, and we have revised our paper to make it clearer. Please check the PDF (highlighted in orange and blue). We use conformation generation as our representation learning task, and our aim is not beating other conformation generation methods. Instead, we simply want to verify that our method can be used for generation and have certain advantages over the Geodiff baseline, such as better generalization or efficiency.
> >
> > ---
> > > **Further comment 3:** The scope of this work has not been changed;
> >
> > **Response 3:**
> > Thanks for you valuable comments!
> > We would like to further clarify the scope of our work and explain why we use the representaion learning in our title.
> >
> > - Generative Tasks: Generative tasks involve creating or generating new data points that are similar to the training data. Examples include generating images, text, or other types of data. Generative models, such as Generative Adversarial Networks (GANs), Variational Autoencoders (VAEs), or Diffusion Probability Models (DPMs) are commonly used for generative tasks.
> >
> > - Representation Learning: Representation learning is a broader concept that focuses on learning efficient representations or features from the raw data. The idea is to automatically learn a meaningful and compact representation of the input data that captures its essential characteristics. These learned representations can be useful for various tasks, including both generative and discriminative tasks. For example, in image recognition, a good learned representation might make it easier to perform tasks like classification or object detection.
> >
> > - While generative tasks often involve representation learning (learning to represent the underlying structure of the data), representation learning is not limited to generative tasks. It can be applied to various tasks, including classification, regression, clustering, and more.
> >
> > In summary, generative tasks can be seen as one application of representation learning. Representation learning is a broader concept that encompasses the idea of learning meaningful and useful representations from data, and these learned representations can be applied to a variety of tasks, not just generative ones.
> >
> > Therefore, in our SubGDiff, the training prodcure of conformation generation can be regarded as represention learning, and the representions are used for classification and regression (Molecular property prediction).

---

### Author Response · Authors · 2023-11-20
**General Response**

Dear Reviewers,

We sincerely thank all the reviewers for their insightful comments and appreciate their acknowledgment that the paper is clearly presented (7f3c), well theoretically motivated (aJi2,7f3c), and provides good mathematical details and analysis (7f3c,yz4M); proposed method is novel (aJi2), interesting (7f3c), technically sound (yz4M), and well explained (7f3c); experimental results are good (aJi2) and well-explained (7f3c); tackled problem is interesting (7f3c), promising (aJi2), and attractive (aJi2). We have carefully considered the comments and will take them into account to further enhance the quality of our work.

For the time being, we have updated the PDF accordingly, and the changes are listed as follows:


- We include the experimental results for 2D molecular property prediction tasks in Table 2 in the main text.

- We update the results for 3D molecular property prediction tasks. Please refer to Table 1. In addition, we provide the experimental results for MD17, as shown in Table 12 in the Appendix.

- We conduct experiments on conformation generation under different numbers of diffusion time steps. For detailed results, please check Table 4 in the main text, Table 7 and Table 9 in the Appendix.

- Important experimental details, such as denoising network architecture, are added to Appendix F for further clarity.

- We include some necessary proofs and derivations in Appendix B and C.


Please also find below our responses to specific concerns of each reviewer. We remain committed to addressing any further questions or concerns from the reviewers promptly.

Best regards,

The Authors

---

### Meta-Review · Area_Chair_A2hu · 2023-12-04

**Metareview:**

The paper introduces a new 3D molecular generation framework, starting with the MaskedDiff approach for enhanced representation learning in molecular graphs. Subsequently, the authors propose SubgDiff model to extend the model's capabilities for molecule generation. The experiments conducted showcase the method's effectiveness in both generation and property prediction. The overall idea is both interesting and promising. The paper is also well-organized and impressed by providing good mathematical details and analysis for the proposed method.  However, the experiment section exhibits a weakness as the results do not convincingly demonstrate the method's utility. The rebuttal, unfortunately, does not adequately address the majority of concerns raised by the reviewers, particularly regarding unclear motivation, limited performance improvement, and generalization ability. Consequently, the consensus among reviewers tends towards rejecting the paper. The AC aligns with the reviewers' opinions, indicating that the paper is not currently ready for publication. Nevertheless, the AC acknowledges the promise of the proposed method and encourages the authors to enhance their paper by incorporating the suggestions provided by the reviewers. The AC recommends resubmitting the revised paper for consideration at the next venue.

**Justification For Why Not Higher Score:**

The paper is below the bar of the publication standards of the conference. The authors have not adequately addressed the significant concerns raised by the reviewers. Following discussions, all reviewers reached a unanimous consensus to reject the paper.

**Justification For Why Not Lower Score:**

NA

---

### Decision · Program_Chairs · 2024-01-16

Reject